# Fibroblast hierarchy dynamics during mammary gland morphogenesis and tumorigenesis

Rosa Pascual [1,2], Jinming Cheng [1,2,3], Amelia H De Smet[1,2], Bianca D Capaldo [1,2], Minhsuang Tsai[1], Somayeh Kordafshari[1,2], François Vaillant [1,2], Xiaoyu Song [1,2], Göknur Giner[3], Michael J G Milevskiy[1,2], Felicity C Jackling[1], Bhupinder Pal[4], Toby Dite[2,5], Jumana Yousef [2,5], Laura F Dagley[2,5], Gordon K Smyth [3,6], Naiyang Fu[1,2], Geoffrey J Lindeman [1,7,8], Yunshun Chen[1,2,3] & Jane E Visvader [1,2]✉

## Abstract

Fibroblasts form a major component of the stroma in normal mammary tissue and breast tumors. Here, we have applied longitudinal single-cell transcriptome profiling of >45,000 fibroblasts in the mouse mammary gland across five different developmental stages and during oncogenesis. In the normal gland, diverse stromal populations were resolved, including lobular-like fibroblasts, committed preadipocytes and adipogenesis-regulatory, as well as cycling fibroblasts in puberty and pregnancy. These specialized cell types appear to emerge from CD34[high] mesenchymal progenitor cells, accompanied by elevated Hedgehog signaling. During late tumorigenesis, heterogeneous cancer-associated fibroblasts (CAFs) were identified in mouse models of breast cancer, including a population of CD34[-] myofibroblastic CAFs (myCAFs) that were transcriptionally and phenotypically similar to senescent CAFs. Moreover, Wnt9a was demonstrated to be a regulator of senescence in CD34[-] myCAFs. These findings reflect a diverse and hierarchically organized stromal compartment in the normal mammary gland that provides a framework to better understand fibroblasts in normal and cancerous states.

**Keywords** Fibroblasts; Mammary Gland Development; Cancer-Associated Fibroblasts (CAFs); Senescence; Wnt9a
**Subject Categories** Cancer; Methods & Resources

## Introduction

Fibroblasts encompass a heterogeneous population of mesenchymal stromal cells that play key roles in tissue homeostasis and disease. Their main function is to provide structural support to tissue by depositing and remodeling the extracellular matrix (ECM), but they are also responsible for building cellular niches through the generation of mechanical forces and the secretion of cytokines and growth factors (Plikus et al, 2021; Younesi et al, 2024). Fibroblasts are known for their remarkable plasticity as they harbor multilineage potential and can fluctuate between resting and activated states (Kalluri, 2016; Younesi et al, 2024). The inherent plasticity within this cell population has hampered the identification of specific fibroblast markers and our understanding of fibroblast function (Sahai et al, 2020).

Recent single-cell RNA sequencing (scRNA-seq) studies of fibroblasts in their steady-state have identified mesenchymal progenitors and specialized fibroblasts that maintain tissue integrity (Muhl et al, 2020; Buechler et al, 2021). In the mouse mammary gland, molecular profiling has confirmed the presence of these subtypes and illuminated two primary subsets of fibroblasts distinguished on the basis of CD26 expression (Yoshitake et al, 2022; Houthuijzen et al, 2023). Moreover, CD26 was previously shown to delineate two spatially distinct populations of fibroblasts in the normal human breast: lobular fibroblasts in the more collagenous stroma surrounding the terminal duct lobular unit (TDLU) and interlobular fibroblasts in the adipose tissue outside the TDLU (Morsing et al, 2016; Kumar et al, 2023). With aging, the proportion of mammary fibroblasts and their expression of extracellular matrix (ECM)-related genes appears to decrease (Li et al, 2020; Pal et al, 2021), likely reflecting the influence of the reproductive state on ECM composition (Schedin and Keely, 2011). However, heterogeneity within the fibroblast compartment during morphogenesis and remodeling of the post-natal mammary gland has yet to be explored.

In breast tumors, the profiling of cancer-associated fibroblasts (CAFs) by scRNA-seq has identified distinct CAF populations with either high expression of ECM genes or enriched for immunomodulatory genes, termed myCAFs (myofibroblast-like) and iCAFs (inflammatory), respectively. This broad CAF classification was

[1]ACRF Cancer Biology and Stem Cells Division, The Walter and Eliza Hall Institute of Medical Research, Parkville, VIC 3052, Australia. [2]Department of Medical Biology, The University of Melbourne, Parkville, VIC 3010, Australia. [3]Bioinformatics Division, The Walter and Eliza Hall Institute of Medical Research, Parkville, VIC 3052, Australia. [4]Translational Breast Cancer Program, Olivia Newton-John Cancer Research Institute and School for Cancer Medicine La Trobe University, Heidelberg, VIC 3084, Australia. [5]Advanced Technology and Biology Division, The Walter and Eliza Hall Institute of Medical Research, Parkville, VIC 3052, Australia. [6]School of Mathematics and Statistics, The University of Melbourne, Parkville, VIC 3010, Australia. [7]Department of Medicine, Royal Melbourne Hospital, The University of Melbourne, Parkville, VIC 3010, Australia. [8]Parkville Familial Cancer Centre and Department of Medical Oncology, The Royal Melbourne Hospital and Peter MacCallum Cancer Centre, Parkville, VIC 3050, Australia. ✉E-mail: visvader@wehi.edu.au

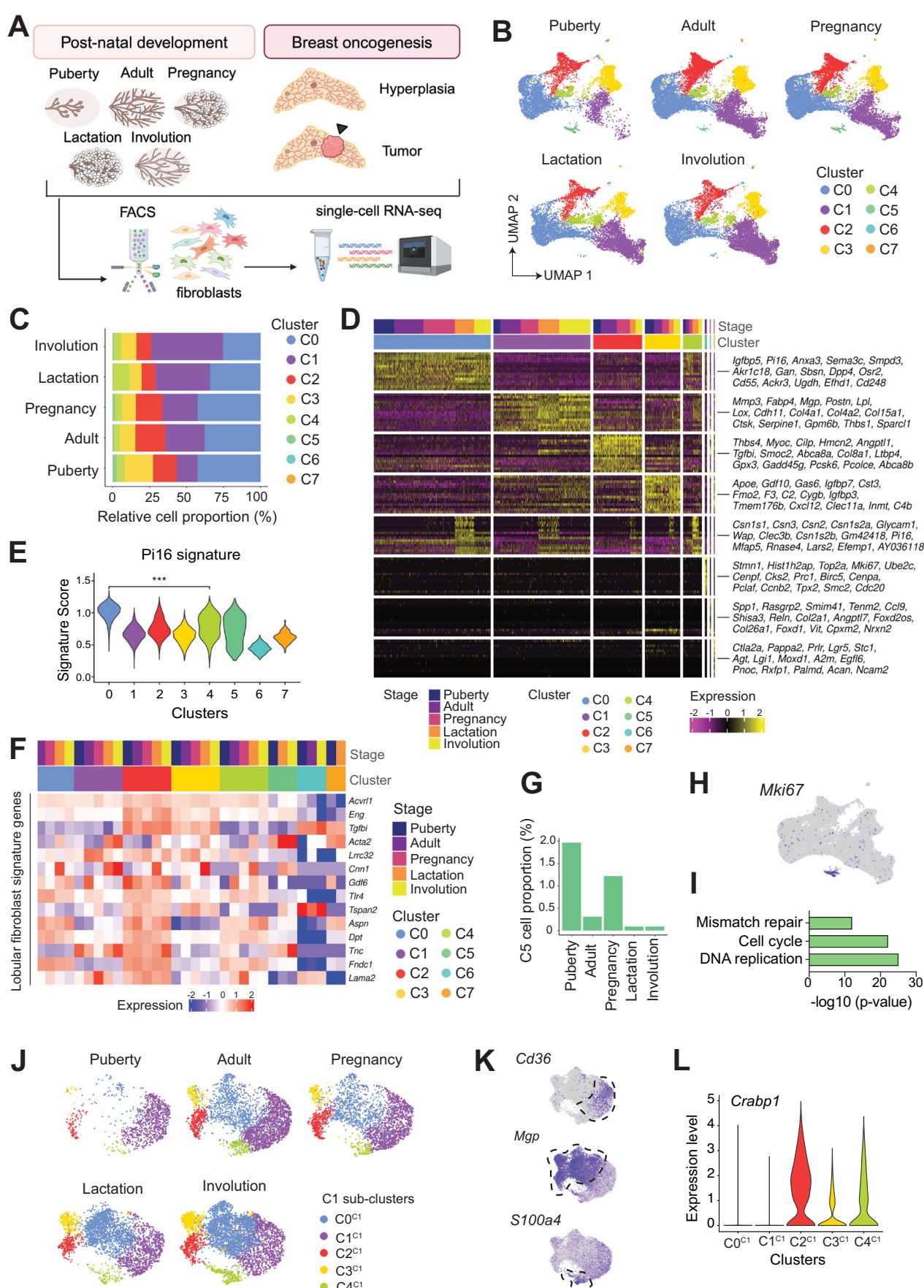

**Figure 1.   Single-cell atlas of fibroblasts across post-natal developmental stages.**

(A) Workflow for generation of a single-cell RNA sequencing (scRNA-seq, 10x Chromium platform) atlas of mouse mammary fibroblasts. Lineage-negative CD24-negative cells from C57BL/6 mice were isolated by FACS from five developmental stages: puberty (4.5-week old), adult (9-week-old virgin), pregnancy (14.5 days), lactation (10 days) and involution (4 days). CD45, CD31, and TER-119 were used as lineage markers. For breast oncogenesis, Lin⁻GFP⁺ cells from *MMTV-Wnt1 Pdgfra*-GFP tumors or hyperplastic mammary glands from the same mouse were isolated by FACS before scRNA-seq ($n = 2$). Created with BioRender.com. (B) UMAP plots of the integration analysis across the different developmental stages colored by cluster identity (C0–C7), including 6924 cells for puberty, 11,024 cells for adult, 10,470 cells for pregnancy, 8054 cells for lactation and 9302 cells for involution. (C) Relative cell proportion (%) of each cluster at each developmental stage. (D) Heatmap for gene expression showing the top 15 marker genes for each cluster at five developmental stages. (E) Violin plot for the enrichment of the Pi16 signature (Buechler et al, 2021) in each cluster. ***$p < 0.001$, Wilcoxon rank-sum test. (F) Heatmap of pseudo-bulk samples showing gene expression for marker genes of lobular fibroblasts (Morsing et al, 2016) in each cluster at five developmental stages. (G) Relative cell proportion of cycling fibroblasts (C5) in each developmental stage. The same data as in Fig. 1C, showing C5 alone for clarity. (H) UMAP plot colored by *Mki67* expression. (I) Bar plot of top KEGG upregulated pathways in C5 vs the rest of clusters. Up or down-regulated genes were obtained by pseudo-bulk differential gene expression analysis. (J) UMAP plots of the sub-clustering of C1 colored by cluster identity (C0^{C1}–C4^{C1}) at the different developmental stages. (K) UMAP plots of the sub-clustering of C1 colored by expression of selected genes. (L) Violin plot for *Crabp1* expression in the C1 subclusters (C0^{C1}–C4^{C1}).

**Table 1.   Marker genes of different cell types.**

| Cell_type | Marker_genes |
| --- | --- |
| Fibroblast | *Dcn, Lum, Pdgfra, Col1a2, Igfbp5* |
| Epithelial | *Epcam, Itga6, Krt8, Krt18, Krt14, Acta2* |
| Endothelial | *Pecam1, Vwf, Cd34* |
| Pericyte | *Rgs5, Mcam* |
| Schwann | *Mpz, Mbp, Sox10, Col20a1* |
| Macrophage | *C1qa, Cd68, Fcgr1, Fcgr2b, Cd163* |
| Keratinocyte | *Krtdap, Krt10* |
| Muscle | *Des, Myf5, Myog, Acta1* |

first defined for pancreatic tumors (Öhlund et al, 2017), while further subpopulations and spatial contexts have been subsequently identified in different cancer types (Bartoschek et al, 2018; Foster et al, 2022; Wu et al, 2020; Houthuijzen and Jonkers, 2018; Kieffer et al, 2020; Friedman et al, 2020; Cords et al, 2024; Elyada et al, 2019). Recently, senescent CAFs (senCAFs) were revealed in breast and pancreatic tumors as a subset of myCAFs that exhibited potent tumor-promoting features (Ye et al, 2024; Belle et al, 2024). CAF populations appear to originate from normal tissue-resident fibroblasts in the mammary gland (Houthuijzen and Jonkers, 2018), but the precise relationship between normal cells and CAFs is an evolving area of investigation.

In this study, we sought to dissect the molecular and cellular heterogeneity within the mouse mammary fibroblast compartment during the different stages of post-natal morphogenesis and neoplastic progression. Through single-cell profiling of >45,000 fibroblasts across puberty, adulthood, pregnancy, lactation and involution, together with functional studies, we identified dynamic changes in novel subsets of fibroblasts during morphogenesis and resolved a putative differentiation hierarchy emanating from CD34^{hi} mesenchymal progenitor cells. Perturbation of the differentiation hierarchy occurred during mammary oncogenesis, with tumors harboring a large population of CD34-negative myCAFs that possessed traits of senCAFs and were regulated by Wnt9a. This atlas provides insights into the hierarchical organization of fibroblasts during normal tissue development and oncogenesis.

## Results

### Mammary fibroblast subpopulations across post-natal development

To construct a single-cell atlas of mammary fibroblasts at different stages of post-natal development, we profiled the Lineage-negative (depleted for CD31-, CD45-, and TER-119-positive cells) and CD24-negative (Lin⁻ CD24⁻) stromal fractions from mice during puberty, adulthood, pregnancy, lactation, and early involution. Following the sorting of stromal cells, scRNA-seq was performed using the 10x Chromium platform (Fig. 1A, see Methods). After quality control and removal of potential doublets, at least 6500 single cells per sample were subjected to unsupervised clustering analysis. Minor contaminants, such as endothelial and epithelial cells, were identified based on known marker genes (Fig. EV1A; Table 1) and removed prior to the integration of fibroblast transcriptomes across five developmental stages.

Unsupervised clustering revealed eight fibroblast clusters (C0–C7) that occurred in differing proportions across post-natal development, visualized in the unifold manifold approximation and projection (UMAP) plots (Fig. 1B,C). Top marker genes were obtained for each cluster, revealing fibroblast subpopulations with potentially distinct functions in the mammary gland (Fig. 1D; Dataset EV1). C0 expressed features of a pan-tissue mesenchymal progenitor population (Buechler et al, 2021), enriched for the Pi16 signature that includes *Dpp4* (encoding CD26) and general stem cell markers such as *Cd34* and *Ly6a* (Fig. 1D,E). Cluster C1 was marked by the matrix metallopeptidase *Mmp3* and the fatty acid binding protein *Fabp4* and was most prominent in involution (Fig. 1C,D). Interestingly, C2 appeared to constitute a novel population of mammary fibroblasts that expressed *Bmp5* and *Thbs4*, and exhibited a transcriptional profile distinct from steady-state fibroblast populations in other tissues (Buechler et al, 2021) (Figs. 1D and EV1B,C). Strikingly, C2 shared expression features with lobular fibroblasts located within the more fibrous human TDLU (Morsing et al, 2016) (Fig. 1F). This signature includes Endoglin (*Eng*, CD105) and Fibronectin type III domain containing I (*Fndc1*) as well as TGFβ-associated genes (*Cilp, Tgfbi*) (Figs. 1D,F and EV1D). Thus, C2 was annotated as lobular-like fibroblasts, and these were found to be less abundant in lactation and involution (Fig. 1C). Cluster C3 was identified as comprising

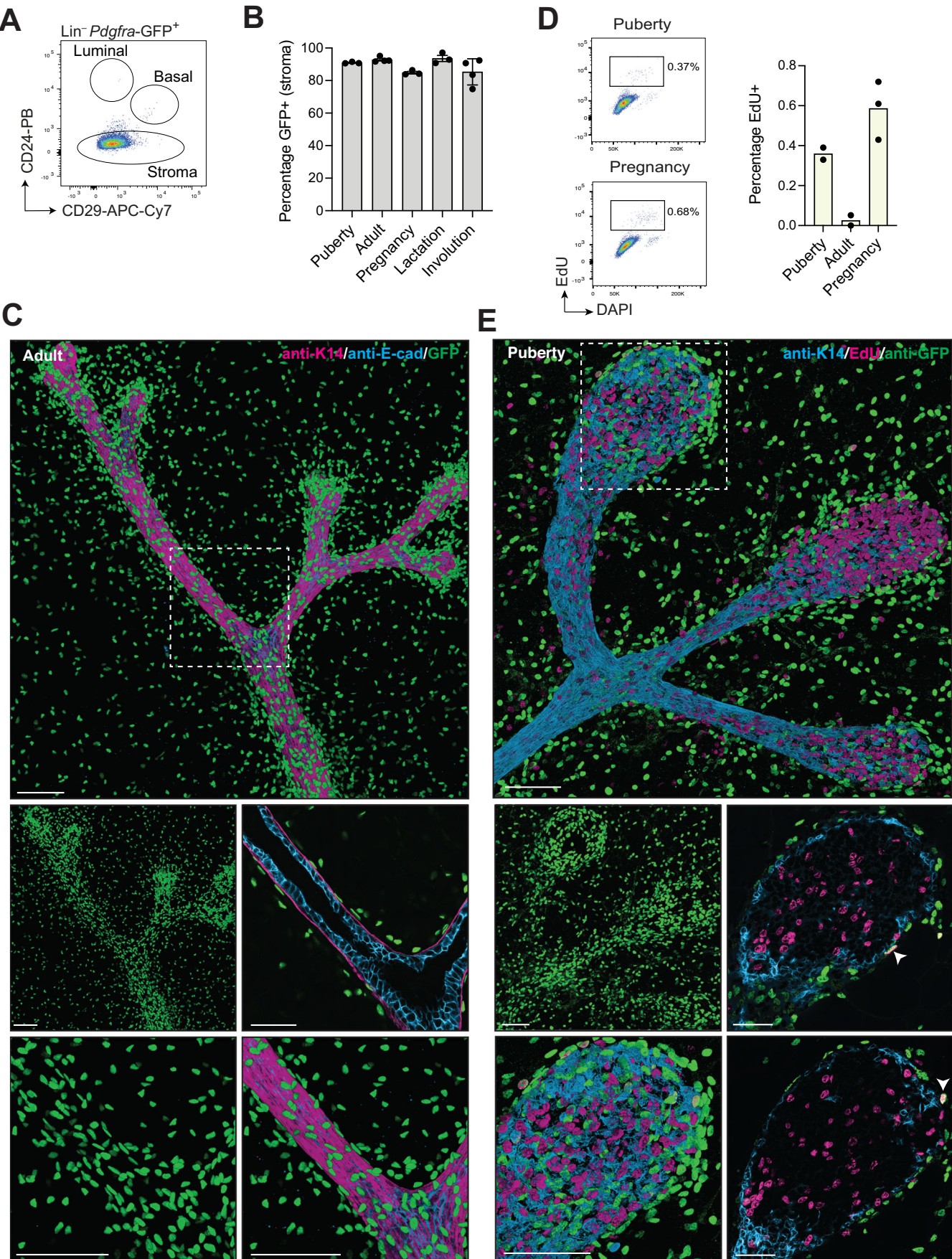

**Figure 2. Spatial distribution of Pdgfrα⁺ fibroblasts in the mammary gland.**

(A) Representative flow cytometry plots for CD29 and CD24 expression in the Lin⁻Pdgfra-GFP⁺ population in adult mammary glands (n = 15; 9–11-week old). (B) Percentage of Pdgfra-GFP⁺ cells in the stroma at different stages: puberty (n = 3, 4.5-week-old), adult (n = 4, 9–11-week-old virgin), pregnancy (n = 3, 14.5 days), lactation (n = 3, 10 days) and involution (n = 4, 4 days). Error bars, mean ± s.e.m. (C) Representative 3D confocal image and optical sections from an adult Pdgfra^H2B-GFP mammary gland (n = 3). Keratin 14 (K14), E-cadherin, and GFP are shown in magenta, cyan, and green, respectively. Scale bar, 100 μm for wholemounts and 50 μm for optical sections. (D) Representative flow cytometry plots for nuclear content (DAPI) and EdU incorporation in the stromal compartment (left) in puberty and pregnancy. Bar plot for the percentage of EdU-positive cells in the stroma (right). Each dot represents an individual mouse (n = 2 for adult and puberty, n = 3 for pregnancy 14.5 days). (E) Representative 3D confocal image and optical sections from pubertal Pdgfra^H2B-GFP mice injected with EdU 2 h prior to collection (6-week old, n = 2). Keratin 14 (K14), EdU and GFP shown in cyan, magenta and green, respectively. Scale bar, 100 μm for wholemounts and 50 μm for optical sections. Arrowheads indicate EdU⁺GFP⁺ double-positive cells. Source data are available online for this figure.

adipogenesis-regulatory cells on the basis of expression of *Clec11a*, *Gdf10*, and *Fmo2*, amongst other genes (Schwalie et al, 2018; Yoshitake et al, 2022), and was enriched in puberty compared to the other stages, suggesting that adipogenesis and ductal morphogenesis are coordinated processes (Figs. 1C,D and EV1E). Cluster C4 displayed commonalities with both C0 and C1 but also included cluster-specific markers such as the cell adhesion-related gene *Glycam1* and the glycoprotein *Tnxb* (Fig. 1D). Rare cycling fibroblasts were confined to cluster C5 and were detected in both puberty and pregnancy (Fig. 1G–I).

As C1 showed the most notable changes during ductal morphogenesis (Fig. 1B,C), we further interrogated heterogeneity within this cluster (Figs. 1J and EV1F). Reclustering of C1 revealed five subclusters (C0^C1–C4^C1), where subcluster C1^C1 was highly enriched for *Fabp4⁺* *Cd36⁺* committed preadipocytes (Merrick et al, 2019) and represented almost the entire C1 cluster at puberty (Figs. 1J,K and EV1G). Accordingly, C1 showed upregulation of PPARγ signaling compared to the other clusters (Fig. EV1D). Although the early adipocyte precursor gene *Dlk1* (Pref1) (Hudak et al, 2014) was not expressed in C1, expression was observed in the minor clusters C6 and C7 (Fig. EV1H), suggesting that several adipocyte progenitor populations may co-exist in the mammary gland. Interestingly, new subpopulations of fibroblasts within C1 (C0^C1, C2^C1, C3^C1, and C4^C1) emerged in the adult, including fibroblasts expressing the vitamin K-dependent matrix protein Mgp, which has been linked to mechanotransduction and fibrosis in breast cancer (Foster et al, 2022) (Fig. 1J,K). C1 Mgp⁺ fibroblasts displayed specific upregulation of *Lox*, encoding lysyl oxidase, which mediates collagen cross-linking and ECM stiffness (Levental et al, 2009), and these were expanded during involution (Figs. 1J,K and EV1F). Within the C1 cluster, we also observed a small population of *S100a4* (Fsp1)⁺ fibroblasts (C4^C1) expressing the non-fibrillar collagen *Col14a1* that also increased in involution (Figs. 1J,K and EV1F). Cumulatively, these data suggest that C1 fibroblasts, which are more abundant in involution, may favor a stiffer ECM. Indeed, the stroma of tissues with high mammographic density is stiffer and comprises more collagen and higher levels of collagen cross-linking enzymes (Northey et al, 2024). Interestingly, the C1 transcriptional signature but not the lobular-like C2 signature was upregulated in human fibroblasts from high versus low-density breast tissue (Kumar et al, 2023) (Fig. 1EVI). Our data further revealed a subpopulation of fibroblasts (C2^C1), which is markedly decreased in puberty, and expresses the retinoic acid binding protein *Crabp1* and the retinol transporter *Stra6* (Figs. 1J,L and EV1F). CRABP1 has been associated with high regenerative competence in skin fibroblasts (Sinha et al, 2022; Guerrero-Juarez et al, 2019) and is expressed in a subset of CAFs in breast tumors enriched for genes related to ECM deposition and remodeling (Bartoschek et al, 2018). This finding was validated in an independent scRNA-seq dataset of pubertal and adult mammary fibroblasts (Fig. EV1J–M).

## Heterogeneity within the Pdgfrα⁺ fibroblast compartment

*Pdgfra* was homogeneously expressed across the clusters (Fig. EV2A), in accordance with Pdgfrα representing a pan-fibroblast marker expressed by both mesenchymal progenitors and specialized fibroblasts (Buechler et al, 2021). We therefore utilized the *Pdgfra*^H2B-GFP reporter mouse to analyze fibroblasts in the mammary gland, thus confirming that *Pdgfra*-GFP expression was restricted to the stroma (Figs. 2A and EV2B,C). The proportion of Pdgfrα⁺ cells within the stromal fraction appeared to be relatively constant across the different stages of post-natal morphogenesis (Fig. 2B). High-resolution 3D confocal imaging (Rios et al, 2019) showed an abundance of *Pdgfra*-GFP⁺ cells in the stroma of the adult gland, with a substantially higher density surrounding the ducts. Indeed, a subset of fibroblasts appeared to line the myoepithelial layer of the ducts, likely providing structural support to the ductal system (Fig. 2C). In puberty, we observed Pdgfrα⁺ cells close to the terminal end buds (TEBs) that are surrounded by a thinner basement membrane compared to the subtending ducts (Fig. EV2D,E) (Gjorevski and Nelson, 2010; Schedin and Keely, 2011). In addition, we confirmed the existence of cycling fibroblasts (C5) at the puberty and pregnancy stages through FACS analysis of mammary glands isolated from mice labeled with EdU in vivo 2 h before harvesting and determined their location by 3D confocal imaging (Figs. 2D,E and EV2F). Rare EdU⁺ Pdgfrα⁺ fibroblasts were seen in close proximity to the highly proliferative TEBs (Fig. 2E) as well as abutting myoepithelial cells that envelop the alveolar structures in pregnancy (Fig. EV2F).

## Changes in mammary fibroblast subpopulations upon acute hormonal stimulation

To determine the influence of ovarian hormones on normal fibroblast subpopulations, we profiled Lin⁻CD24⁻ cells by scRNA-seq analysis after administering a potent mitogenic stimulus comprising estrogen (E) and the progestin medroxyprogesterone acetate (MPA) to adult mice. As anticipated (Fu et al, 2017), hormonal stimulation led to an increased basal-to-luminal cell ratio compared to that in control mice (Fig. 3A). Similar to the integrated analysis of stromal cells through mammary gland morphogenesis (Fig. 1B), unsupervised clustering of stromal cells from control and hormone-stimulated mice showed five main clusters: *Pi16⁺* mesenchymal progenitors (C0^S), *Crabp1⁺* 'ECM remodeling' fibroblasts (C1^S), *Fabp4⁺* committed preadipocytes (C2^S), *Gdf10⁺* adipogenesis-regulatory cells (C3^S), and *Bmp5⁺* lobular-like fibroblasts (C4^S) (Figs. 3B,C and EV3A). A minor cluster of cycling fibroblasts (C5^S) was also identified (Figs. 3C and EV3A). Both cycling and *Crabp1⁺* 'ECM remodeling' fibroblasts increased with hormonal stimulation, suggesting that hormones influence the stiffness of the

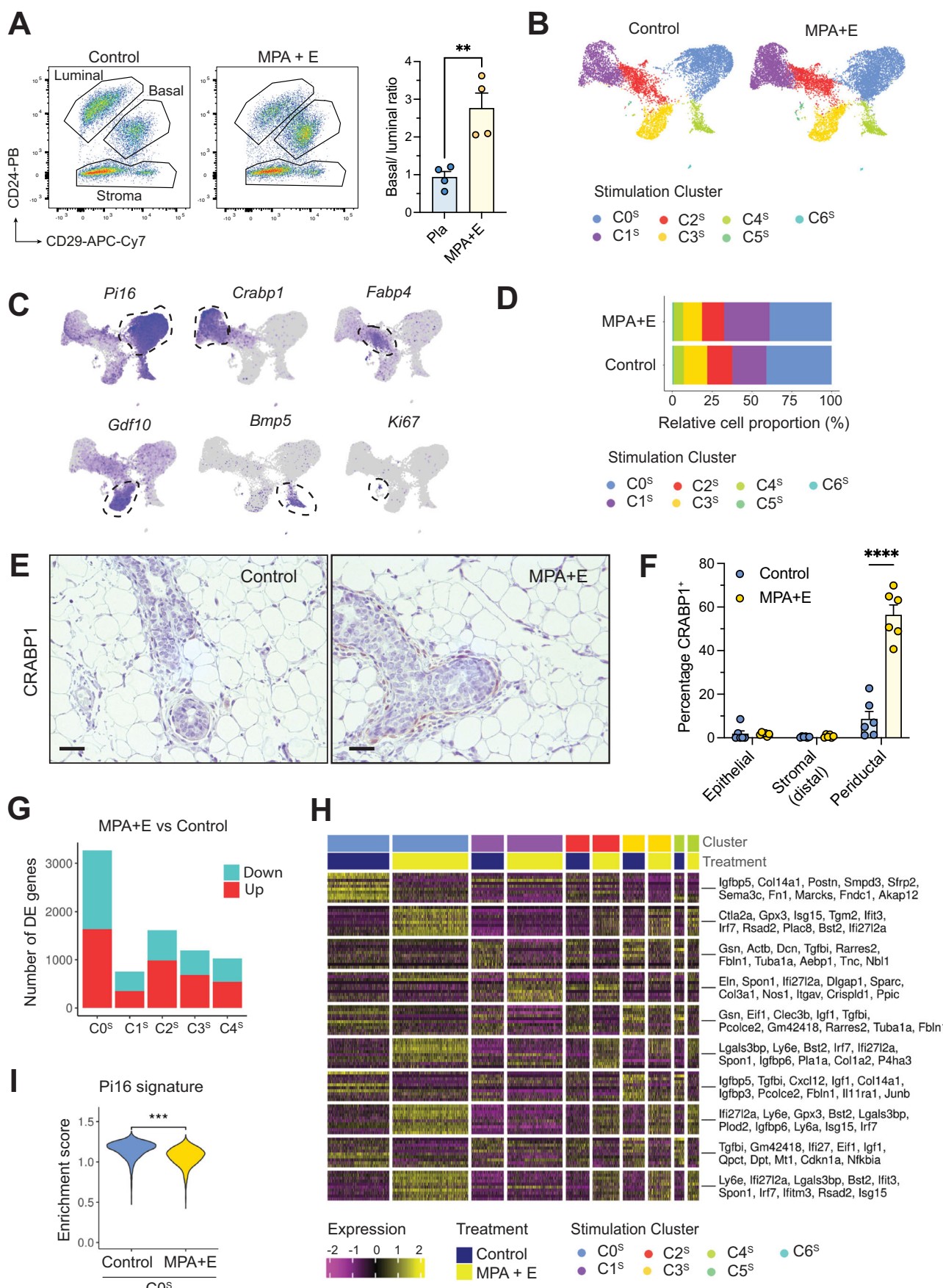

**Figure 3. Changes in the fibroblast compartment upon acute hormonal stimulation.**

(A) Representative flow cytometry plots for CD24 and CD29 expression in the Lineage-negative compartment of placebo control or MPA + E treated adult mice (left) (9-week old). Bar plot showing the ratio of the percentage of basal vs luminal cells in the epithelial compartment (right). Each dot represents an individual mouse ($n = 4$). Error bars, mean ± s.e.m. **$p < 0.01$, unpaired $t$-test. MPA medroxyprogesterone acetate, E estrogen. (B) UMAP plots of the Seurat integration analysis for cells from control (9495 cells) or MPA + E (12,216 cells) treated mice. Seven clusters (C0$^S$–C6$^S$) are indicated by color, $^S$ depicts clusters from the integration of the acute hormonal stimulation experiment (control and MPA + E). (C) UMAP plots of integrated data colored by expression of selected marker genes. (D) Relative cell proportion (%) of each cluster (C0$^S$–C6$^S$) for each treatment condition. (E) Representative immunostained images for CRABP1 in control (placebo) or MPA + E treated mammary glands ($n = 6$) (9-week old). Scale bar, 20 μm. (F) Quantification of the percentage of CRABP1$^+$ cells in different mammary gland locations (epithelial, distal stromal, or periductal) using QuPath in control (placebo) or hormonal stimulation (MPA + E) ($n = 6$). Cells from one inguinal mammary gland section were quantified per mouse. Error bars indicate mean ± s.e.m., ****$P < 0.0001$, two-way ANOVA. (G) A number of upregulated and downregulated differentially expressed (DE) genes by pseudo-bulk analysis in the five main clusters for MPA + E vs control (placebo). (H) Expression heatmap showing the top ten marker genes in each cluster for either MPA + E or control conditions. (I) Violin plot showing enrichment of the Pi16 signature (Buechler et al, 2021) in C0$^S$ (mesenchymal progenitors) under control vs MPA + E conditions. ***$p < 0.001$, Wilcoxon rank-sum test. Source data are available online for this figure.

stroma as reported for human tissue (Boyd et al, 2006) (Fig. 3D). CRABP1 immunostaining of tissue from control and hormone-treated mice confirmed an increase in CRABP1$^+$ fibroblasts upon hormonal stimulation (Fig. 3E,F). Remarkably, CRABP1$^+$ cells were specifically confined to stromal cells close to the epithelial ducts, as quantified using QuPath (Figs. 3E,F and EV3B).

The most striking transcriptional changes were evident for mesenchymal progenitors, where C0$^S$ showed the highest number of differentially expressed genes in MPA + E vs control mice compared to other clusters, including downregulation of *Igfbp5*, which is part of the *Pi16* signature (Figs. 3G,H and EV3C). Indeed, the *Pi16* signature was significantly decreased after hormonal stimulation relative to control mice, suggesting enhanced differentiation of progenitor cells (Fig. 3I). As previously reported (Kanaya et al, 2019), fibroblasts expressed the estrogen and progesterone receptors (Fig. EV3D,E). Therefore, the observed responses to acute hormonal treatment could be directly mediated via steroid receptors on stromal cells.

## Interrogation of the predicted mammary fibroblast hierarchy in normal tissue and during oncogenesis

To investigate potential relationships amongst the different fibroblast clusters across development, we performed a pseudo-time trajectory analysis selecting the mesenchymal progenitor population as a starting point (Fig. 4A). These progenitor cells were marked by high levels of CD34 (Fig. EV3F), and were identified on the basis of the transcriptional signature of CD34$^{hi}$ vs CD34$^{lo}$ fibroblasts at a population level (Fig. 4B–E). In accordance with a previous study (Houthuijzen et al, 2023), CD34$^{hi}$ cells expressed higher levels of *Dpp4* but lower levels of *Col15a1* and *Col18a1* (Dataset EV2). Three differentiation sub-trajectories (STs) were predicted to generate the three major fibroblast clusters: C2 (ST1), C3 (ST2), and C4 (ST3) (Fig. 4F). We next examined gene expression of key determinants of cell differentiation along each ST, calculating a readout of pathway activity by aggregating the expression levels of manually curated lists of target genes for each pathway (Rowton et al, 2022). Upregulation of the Hedgehog (Hh) pathway correlated with increasing differentiation for all groups, while Notch activity decreased in ST1 and Wnt pathway activity increased in ST3 (Fig. 4G). Strikingly, CD34$^{lo}$ fibroblasts upregulated Hh-related genes and ECM-related pathways (Figs. 4H and EV3G). Mass spectrometry-based proteomics analysis of the secretome of cultured CD34$^{lo}$ fibroblasts revealed an enrichment of proteins related to ECM organization and remodeling (e.g., TIMP-3 and Collagen XIV), while CD34$^{hi}$ fibroblasts contained an increased abundance of secreted

factors that were related to protein degradation and included ANXA3 and OGN that are part of the Pi16 mesenchymal progenitor signature (Fig. EV3H–J).

To further examine the potential inverse relationship between CD34 and Hh activity in fibroblasts, we generated a *Gli1*-GFP reporter mouse model as Gli1 is a key transcriptional effector of Hedgehog signaling (Cassandras et al, 2020) (Fig. EV4A). The specificity of the reporter was confirmed by the enrichment of *Gli1* and other key Hh-target genes (*Gli2*, *Ptch1*, and *Ptch2*) in GFP$^+$ vs GFP$^-$ sorted fibroblasts (Fig. EV4B,C). In the mammary gland, *Gli1*-GFP expression was restricted to CD34$^{lo}$ cells in the stroma, where it accounted for ~12% of cells, indicating that *Gli1* demarcates a subset of specialized fibroblasts (Figs. 4I,J and EV4D). Moreover, *Gli1*-GFP$^+$ fibroblasts were Pdgfrα$^+$ and comprised a higher percentage of Pdgfrβ$^+$ cells and a lower percentage of Podoplanin (Pdpn)$^+$ cells compared to *Gli1*-GFP$^-$ fibroblasts (Figs. 4J and EV4E). Conversely, CD34$^{hi}$ mesenchymal progenitors were Pdpn$^{hi}$ in pubertal and adult glands (Fig. EV4F,G). When used as feeder layers in co-culture assays with basal cells (see Methods, Fig. EV4H), *Gli1*-GFP$^+$ cells behaved similarly to CD34$^{lo}$ specialized fibroblasts (Fig. EV4I,J). Although fibroblasts are known to change upon culture (Salminen, 2023), CD34$^{lo}$ specialized fibroblasts retained lower expression of *Cd34* and higher expression of *Col15a1* compared to CD34$^{hi}$ mesenchymal progenitors as per freshly sorted samples (Fig. EV4K). Together our data indicate a hierarchical organization of mammary fibroblasts and suggest that upregulation of the Hh pathway may represent a potential readout of differentiation status.

To determine how the normal fibroblast differentiation hierarchy changes through oncogenesis, we crossed *Pdgfra*-GFP or *Gli1*-GFP reporter mice with the *MMTV-Wnt1* model of breast cancer (raGFP-Wnt1 or Gli1-Wnt1, respectively). Interestingly, we observed a major population of CD34$^-$ cancer-associated fibroblasts (CAFs) in raGFP-Wnt1 tumors that was not apparent at the hyperplastic stage (MG$^{hyper}$) (Fig. 4K). Parallel findings were made in an independent mammary tumor model (MMTV-cre$^{T/+}$ *Trp53*$^{fl/+}$ *Brca2*$^{fl/fl}$), suggesting that perturbation of normal differentiation primarily occurs in the late stages of tumor development (Fig. 4L,M). Moreover, CD34 expression distinguished inflammatory (iCAF, CD34$^+$) and myofibroblastic CAFs (myCAF, CD34$^-$) based on their transcriptomes at the population level (Fig. 4N). In line with CAFs assuming specialized phenotypes in tumors, Pdgfrα$^+$ *Gli1*-GFP$^+$ fibroblasts increased in tumors compared to hyperplastic tissue (Fig. 4O). However, in contrast to normal tissue, no association between CD34 expression and Hh activity was evident within tumors (Fig. EV4I; Dataset EV3).

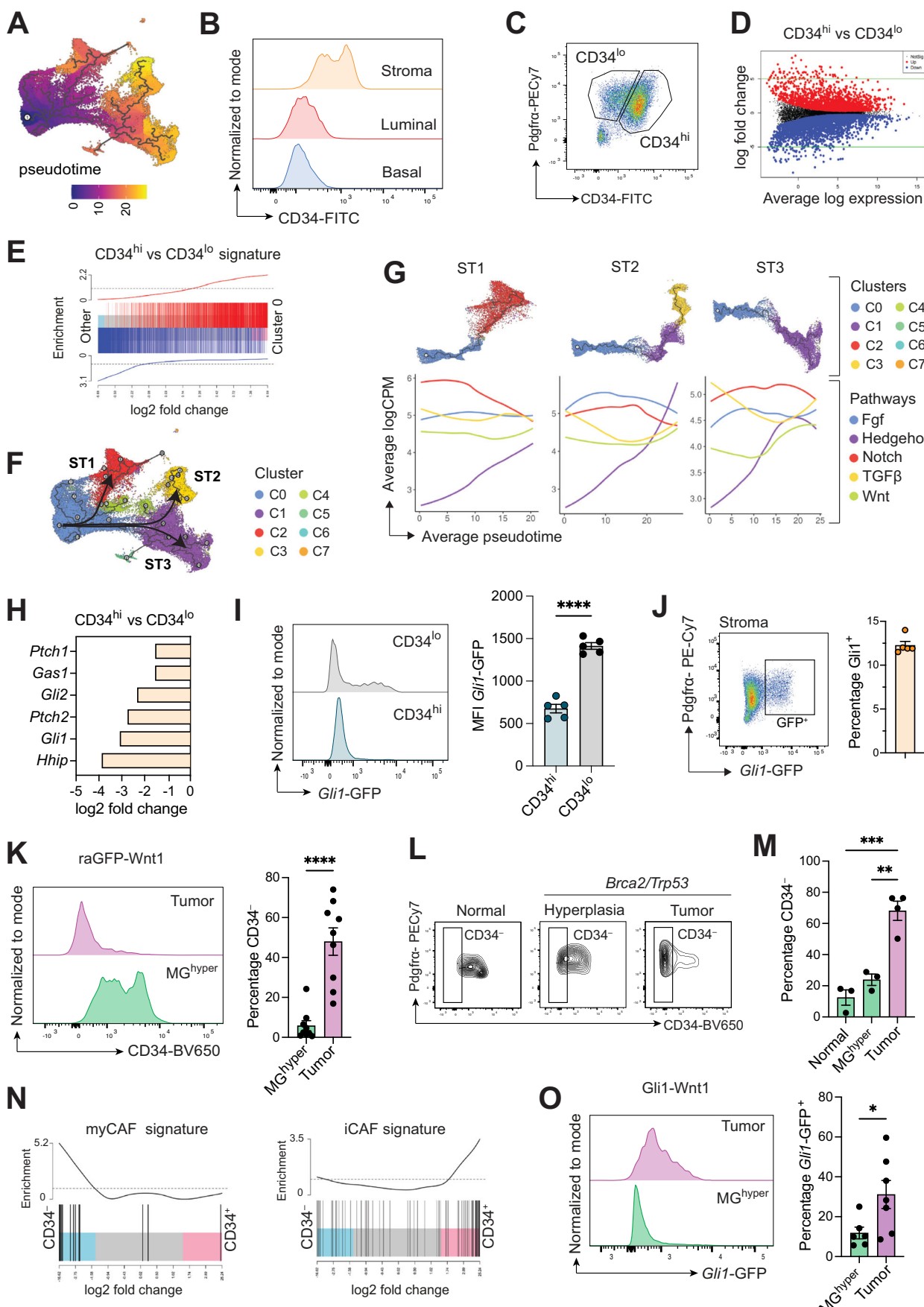

Figure 4.   Predicted fibroblast hierarchy from CD34$^{hi}$ progenitor cells in homeostasis and tumors.

(A) UMAP plot colored according to the pseudo-time trajectory analysis with the starting node in C0 (mesenchymal progenitors). (B) Representative histogram of CD34 expression in stromal (CD24$^-$Pdgfrα$^+$), luminal (CD29$^{lo}$CD24$^{hi}$), and basal (CD29$^{hi}$CD24$^{lo}$) cells in adult FVB/NJ mice (9–10-week old) by flow cytometry (n = 10). (C) Representative flow cytometry plots for CD34 and Pdgfrα expression in the stromal (Lin$^-$ CD24$^-$) compartment of pubertal 6-week-old FVB/NJ mice (n = 8). (D) Mean-difference plots showing differentially expressed genes between CD34$^{hi}$ and CD34$^{lo}$ fibroblasts from pubertal 6-week-old FVB/NJ mammary glands analyzed by RNA-seq (n = 3). Significantly upregulated and downregulated differentially expressed genes are shown as red and blue dots, respectively. (E) Barcode plot showing enrichment scores of CD34$^{hi}$ vs CD34$^{lo}$ transcriptional signature (bulk RNA-seq, n = 3) in C0 (mesenchymal progenitors) compared to other clusters. Red and blue bars indicate upregulated genes in CD34$^{hi}$ or CD34$^{lo}$ fibroblasts, respectively. (F) UMAP plot showing sub-trajectory analysis in the pseudo-time colored by cluster identity (C0–C7). Three sub-trajectories are indicated by arrows. ST, sub-trajectory. (G) Independent sub-trajectories (ST1-3) colored by cluster identity (above). Aggregated gene expression along the pseudo-time for specific signaling pathways indicated by color (below). (H) Log$_2$ fold change for significantly upregulated Hedgehog-related genes in CD34$^{lo}$ vs CD34$^{hi}$ fibroblasts assessed by RNA-seq analysis (n = 3). (I) Representative histogram for Gli1-GFP expression in Pdgfrα$^+$CD34$^{hi/lo}$ fibroblasts by flow cytometry (left). Bar plot for MFI of Gli1-GFP expression in Pdgfrα$^+$CD34$^{hi/lo}$ (right). Each dot represents an individual mouse (n = 5). Error bars, mean ± s.e.m., ****p < 0.0001, unpaired t-test. MFI mean fluorescence intensity. (J) Representative flow cytometry plot of the stromal compartment for Pdgfrα and Gli1-GFP expression (left) and quantification of the percentage of Gli1-GFP$^+$ cells in adult mammary glands (9-week old, right). Each dot represents an individual mouse (n = 5). Error bars, mean ± s.e.m. (K) Representative histograms of CD34 expression in Pdgfra-GFP$^+$ cells from tumors or paired hyperplastic glands (MG$^{hyper}$) from tumor-bearing raGFP-Wnt1 mice (left) and quantification of the percentage of CD34-negative cells in the Pdgfra-GFP$^+$ compartment (right) assessed by flow cytometry. Each dot represents an individual mouse (n = 9). Error bars, mean ± s.e.m., ****p < 0.0001, unpaired t-test. (L) Representative flow cytometry plots of CD34 and Pdgfrα expression in the Pdgfrα$^+$ population in hyperplastic mammary glands and tumors in MMTV-cre$^{T/+}$ Trp53 $^{fl/+}$ Brca2$^{fl/fl}$ mice or littermate controls (normal, MMTV-cre$^{+/+}$ Trp53 $^{fl/+}$ Brca2$^{fl/fl}$). n = 3 for normal and hyperplasia, n = 4 for tumors. (M) Quantification of the percentage of CD34$^-$ fibroblasts in the Pdgfrα$^+$ compartment in hyperplastic mammary glands or tumors in MMTV-cre$^{T/+}$ Trp53 $^{fl/+}$ Brca2$^{fl/fl}$ mice or littermate controls (normal, MMTV-cre$^{+/+}$ Trp53 $^{fl/+}$ Brca2$^{fl/fl}$). n = 3 for normal and hyperplasia, n = 4 for tumors. Error bars, mean ± s.e.m., **p < 0.01, ***p < 0.001, ordinary one-way ANOVA. (N) Barcode plot showing enrichment scores of myCAF and iCAF transcriptional signatures (Elyada et al, 2019) in raGFP-Wnt1 CD34$^{+/-}$ CAFs analyzed by bulk RNA-seq (n = 3). (O) Representative histograms of Gli1-GFP expression in Pdgfrα$^+$ cells from tumors or hyperplastic glands (MG$^{hyper}$) derived from tumor-bearing Gli1-Wnt1 mice (left) and quantification of the percentage of Gli1-GFP$^+$ cells in the Pdgfrα$^+$ compartment (right) assessed by flow cytometry. Each dot represents an individual mouse for MG$^{hyper}$ (n = 6) or tumor (n = 7). Error bars, mean ± s.e.m., *p < 0.05, unpaired t-test.

## Evolving fibroblast populations during hyperplasia and mammary tumor development

Changes in fibroblasts in the hyperplastic period preceding mammary tumor development remain poorly characterized. To assess the impact of hyperplasia-associated fibroblasts (HAFs) versus CAFs on the growth of normal epithelial cells, we optimized a 3D organoid co-culture system using growth factor-reduced conditions. In this assay, epithelial cells cultured alone did not form organoids (Fig. EV5A). Notably, only freshly sorted CAFs but not HAFs enabled the growth of organoids from primary normal basal or luminal progenitor (LP) cells (Fig. 5A,B). The proliferation of HAFs and CAFs appeared similar in vitro (Fig. EV5B). Interestingly, CAF-driven LP-derived organoids lost their luminal cystic appearance and were morphologically similar to organoids generated from LP cells from hyperplastic *MMTV-Wnt1* tissue (Figs. 5A and EV5C). These results highlight intrinsic differences between the stromal compartments of hyperplastic and cancerous tissue and indicate that CAFs create a unique environment, which can influence both the proliferation and differentiation of mammary epithelial cells.

Although fibroblasts form a major component of the tumor microenvironment, the stromal fraction is markedly reduced in tumors versus hyperplastic tissue, thus limiting the power of scRNA-seq studies on total tissue (Fig. EV5D). To enrich for fibroblasts, we sorted *Pdgfra*-GFP$^+$ cells from raGFP-Wnt1 hyperplastic mammary glands and tumors for scRNA-seq analysis. Integration of the Wnt1 scRNA-seq data revealed tumor-specific fibroblast clusters (C4$^W$, C7$^W$, C9$^W$, and C10$^W$) that were marked by expression of *Acta2* and *Tnc*, and included a cycling cluster (C10$^W$) (Fig. 5C–E). Analysis of fibroblast populations from hyperplastic tissue revealed *Crabp1$^+$* "ECM remodeling" fibroblasts as the most abundant HAF population, similar to our findings after acute hormonal stimulation (Figs. 3B–D and 5F,G). Indeed, immunostaining revealed increased CRABP1 expression in hyperplastic compared to age-matched control tissue and these cells localized to

a periductal niche (Fig. 5H). All CRABP1$^+$ cells were *Pdgfra*-GFP$^+$, indicating their fibroblastic nature (Fig. 5I). Other HAF clusters included mesenchymal progenitors (C1$^H$), committed preadipocytes (C2$^H$), adipogenesis-regulatory cells (C3$^H$) and *Bmp5$^+$* lobular-like fibroblasts (C5$^H$), with the latter markedly reduced compared to normal tissue (Figs. 5F,G and EV5E; Dataset EV4). Comparable findings were made for hyperplastic glands from *Brca2/Trp53*-deficient mice (Fig. EV5F,G).

## The CD34-negative CAF population comprises senescent myCAFs

In mammary tumors, CAFs grouped into seven main clusters (C0$^T$-C6$^T$) that were classified as myCAFs (C0$^T$, C1$^T$, and C4$^T$), iCAFs (C2$^T$, C3$^T$, and C5$^T$), or cycling CAFs (C6$^T$) according to their gene expression profile (Elyada et al, 2019) (Fig. 6A,B; Dataset EV5). Antigen-presenting apCAFs were not identified in our dataset, consistent with other studies (Houthuijzen et al, 2023; Wu et al, 2020). Interestingly, C2$^T$, C3$^T$, and C5$^T$ shared expression profiles with *Pi16$^+$* mesenchymal progenitors, *Crabp1$^+$* "ECM remodeling" fibroblasts and *Gdf10$^+$* adipogenesis-regulatory cells, respectively (Fig. EV5H; Dataset EV5). *Bmp5$^+$* lobular-like fibroblasts and *Cd36$^+$* committed preadipocytes were not present within tumors, while *Acta2* (α-SMA)$^+$ myCAFs presented new tumor clusters not found in hyperplastic glands (Fig. EV5H,I). The myCAF transcriptional signature (Elyada et al, 2019) significantly overlapped with our CD34$^-$ CAF transcriptional signature based on bulk RNA-seq data, which was specifically enriched in cluster C0$^T$ (Fig. 6C,D). Tumor fibroblasts expressing CD34, which marks the majority of stroma in normal tissue, appeared transcriptionally similar to iCAFs (Fig. 6B,C). Accordingly, the transcriptional signatures of myCAF clusters (C0$^T$, C1$^T$, and C4$^T$) but not iCAFs (C2$^T$, C3$^T$, and C5$^T$) were enriched in human breast cancer fibroblasts compared to normal fibroblasts, independent of breast cancer subtype (Fig. 6E). The cycling CAF signature was mostly upregulated in CAFs from triple-negative breast tumors, while the CD34$^-$ transcriptional signature was enriched in ER$^+$ and HER2$^+$ CAFs (Fig. EV5J,K).

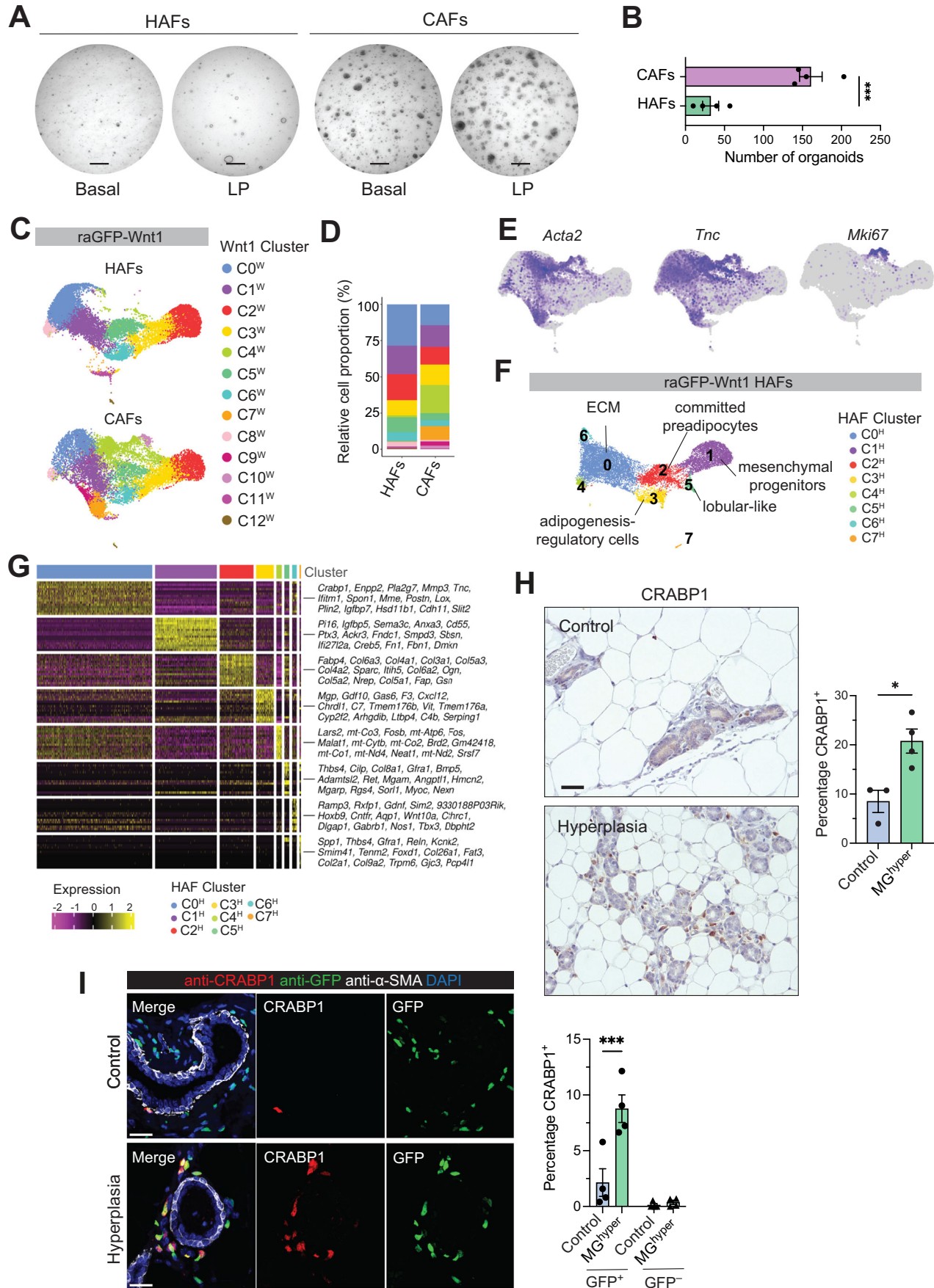

**Figure 5. Fibroblast changes through mammary oncogenesis.**

(A) Representative brightfield images of organoids generated from normal basal or luminal progenitor (LP) cells grown with freshly sorted hyperplasia-associated fibroblasts (HAFs) or cancer-associated fibroblasts (CAFs) from the Wnt1 mouse model ($n = 4$). Scale bars, 200 μm. (B) Quantification of organoids generated from normal basal cells co-cultured with CAFs or HAFs ($n = 4$ independent sets of paired Wnt1 HAFs/CAFs). Error bars, mean ± s.e.m., ***$p < 0.001$, unpaired $t$-test. (C) UMAP plots of the Seurat integration analysis for GFP+ cells from raGFP-Wnt1 mice colored by cluster identity (C0$^W$–C12$^W$), including 12,981 cells for hyperplasia (HAFs) and 12,041 cells for tumors (CAFs). (D) Relative cell proportion (%) of each cluster (C0$^W$–C12$^W$) in hyperplastic tissue (HAFs) or tumors (CAFs) from raGFP-Wnt1 mice. (E) UMAP plots of the integrated Wnt1 data (raGFP-Wnt1 hyperplasia and tumors) colored by expression of selected markers. (F) UMAP plot of raGFP-Wnt1 hyperplasia-associated fibroblasts (HAFs) showing eight clusters (C0$^H$–C7$^H$) indicated by color. Clusters were annotated according to their expression profiles. (G) Heatmap of gene expression showing the top 15 marker genes for each cluster in raGFP-Wnt1 hyperplastic glands (C0$^H$–C7$^H$). (H) Representative immunostained images (left) and quantification (right) of the percentage of CRABP1-positive stromal cells in the periductal niche in raGFP-Wnt1 hyperplastic ($n = 4$, Pdgfra-GFP$^{KI/+}$ MMTV-Wnt1$^{T/+}$) or age-matched littermate ($n = 3$, Pdgfra-GFP$^{KI/+}$ MMTV-Wnt1$^{+/+}$) control tissues. Cells from one entire inguinal mammary gland section were quantified per mouse. Scale bar, 20 μm. Error bars indicate mean ± s.e.m., *$P < 0.05$, unpaired $t$-test. (I) Representative confocal images (left) of raGFP-Wnt1 hyperplastic (Pdgfra-GFP$^{KI/+}$ MMTV-Wnt1$^{T/+}$) or age-matched littermate (Pdgfra-GFP$^{KI/+}$ MMTV-Wnt1$^{+/+}$) control mammary glands ($n = 4$) stained with anti-CRABP1 (red), anti-GFP (fibroblasts, green), anti-α-SMA (myoepithelial marker, white) and DAPI (nuclei, blue). Scale bar, 20 μm. Quantification of CRABP1+ cells within GFP+ or GFP− cells. Error bars indicate mean ± s.e.m., ***$P < 0.001$, ordinary one-way ANOVA (right). Source data are available online for this figure.

To further probe heterogeneity within the myCAF subsets (C0$^T$, C1$^T$, and C4$^T$), we performed pseudo-bulk gene expression and KEGG pathway analysis (Fig. 6F). This analysis revealed that C1$^T$ myCAFs upregulated Glycolysis and Ribosome pathways and that C4$^T$ was enriched for regulators of ECM production and secreted ligands (e.g., *Pdgfrl*, *Tgfb3*, and *Scube2*) (Figs. 6F and EV5l). Furthermore, the ecm-myCAF transcriptional signature associated with immunosuppression and immunotherapy resistance (Kieffer et al, 2020) was found to be more prominent in C4$^T$ (Fig. EV6A). Interestingly, C0$^T$ CD34− myCAFs showed enrichment for the cellular senescence pathway and the Fridman gene set of cellular senescence (Fridman and Tainsky, 2008), and included *Cdkn1a* and *Cdkn2a* as top marker genes (Figs. 6F,G and EV5l). *Cdkn2a* (encoding p16) was exclusively expressed in the CD34− myCAF cluster (C0$^T$), while bulk RNA-seq analysis of CD34− vs CD34+ CAFs also showed upregulation of several senescence-associated genes (such as *Cdkn2a, Cdkn1a* and *Cxcl14*) (Fig. 6H–J). Furthermore, CD34− CAFs expressed higher levels of *Lrrc15* based on scRNA-seq data from the *Wnt1*-driven and *Brca2/Trp53*-deficient mouse models and bulk transcriptomic analysis of *Wnt1* tumors (Figs. 6I and EV6B–D). This feature is reminiscent of the recently described senescent CAF (senCAF) population with a tumor-promoting role in pancreatic and mammary tumors (Ye et al, 2024; Belle et al, 2024). To determine whether CD34− CAFs comprised senCAFs, we isolated CD34− and CD34+ CAFs from the raGFP-Wnt1 model, exposed them to the DNA-damaging agent etoposide and read-out cellular senescence using the β-galactosidase (β-gal) assay. Remarkably, untreated CD34− CAFs showed higher levels of β-gal+ cells, with significantly increased levels observed upon damage compared to CD34+ CAFs (Fig. 6K). Concurrently, we observed decreased proliferation of CD34− compared to CD34+ CAFs in untreated conditions (Fig. EV6E). Thus, CD34− myCAFs appear to be transcriptionally and phenotypically similar to senCAFs.

### Wnt9a regulates the senescence phenotype of CD34− myCAFs

To explore potential mechanisms that drive the senescent phenotype of CAFs, we combined pseudo-bulk gene expression analysis of single-cell data with CD34− vs CD34+ CAFs bulk RNA-seq data. We generated three pseudo-bulk samples for the total fibroblast population across three states: normal tissue, hyperplastic tissue, and tumors. The multidimensional scaling (MDS) plot revealed tumor-associated fibroblasts as the most transcriptionally distinct (Fig. EV6F). We next compared top upregulated genes in CAFs vs HAFs and in CD34− vs CD34+ CAFs, yielding 419 common genes (Figs. 6I, 7A and EV6G). Panther analysis of these gene sets uncovered a myriad of significantly enriched pathways that included the Wnt and Trp53 pathways (Fig. 7A), the latter of which is a major effector of cell senescence (Boutelle and Attardi, 2021). The relationship between Wnt signaling and senescence is yet to be determined but appears to be influenced by both context and cell type (Adams and Enders, 2008). Interestingly, we observed upregulation of *Wnt9a* (also known as *Wnt14*) in CAFs from *Brca2/Trp53*-deficient and *Wnt1*-driven tumors, where it was exclusively expressed in myCAFs (Figs. 7B and EV6H,I). To examine a potential role for Wnt9a in senCAFs, we employed CRISPR/Cas9 editing in primary CD34− myCAFs using two independent single guide RNAs targeting the *Wnt9a* locus (sgWnt9a) (Fig. EV6J) and evaluated the capacity of these cells to become senescent upon etoposide treatment. Downregulation of Wnt9a did not significantly impact cell growth in basal conditions (Fig. EV6K). Notably, Wnt9a loss diminished cellular senescence in myCAFs, as reflected in the decreased number of β-gal+ cells relative to those transduced with the non-targeting control guide (Figs. 7C,D and EV6l). Concomitantly, upon induction of senescence, upregulation of the senescent markers p21 and p16 was diminished in Wnt9a-KO fibroblasts (Figs. 7E and EV6M). These data indicate a functional link between Wnt9a and the acquisition of a senCAF phenotype.

## Discussion

Fibroblasts are the principal constituent of connective tissue where they govern tissue homeostasis through the synthesis and remodeling of ECM. To investigate heterogeneity amongst mammary fibroblasts in the post-natal mouse mammary gland and gain insight into how they change during oncogenic progression, we profiled mammary fibroblasts at the single-cell level across five developmental stages as well as in hyperplastic tissue and tumors. Profound heterogeneity amongst fibroblasts was uncovered in the normal mammary gland, while new populations within this compartment were found to appear late in the oncogenic process rather than at the hyperplastic stage (Fig. 7F).

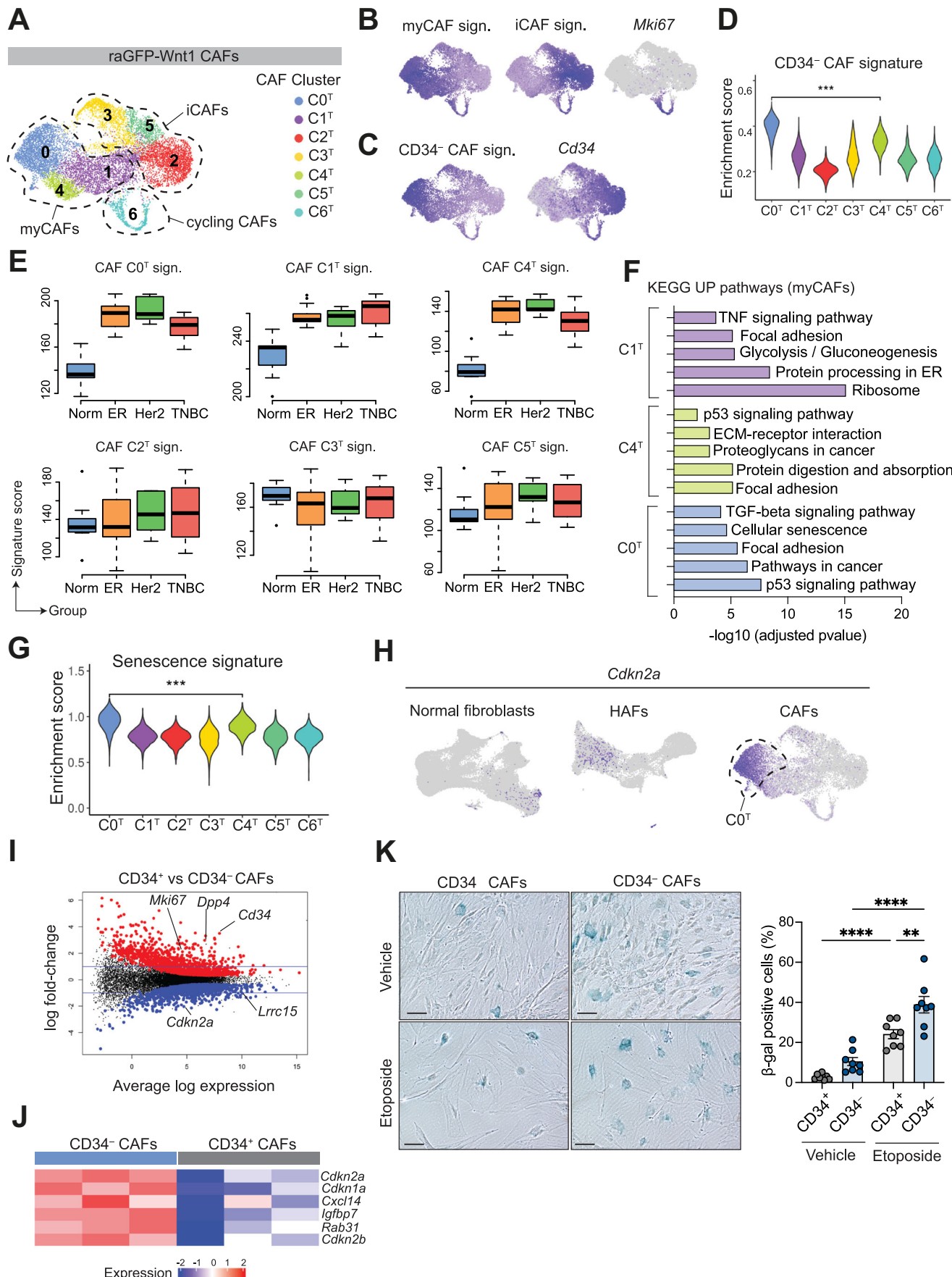

**Figure 6. CD34⁻ myCAFs emerge in mammary tumors.**

(A) UMAP plot of raGFP-Wnt1 tumor fibroblasts showing seven CAF clusters ($C0^T$-$C6^T$) indicated by color. Clusters were annotated according to their expression profiles. (B) UMAP plots of raGFP-Wnt1 CAFs colored by enrichment for myCAF and iCAF signatures (Elyada et al, 2019) or *Mki67* expression. Sign., transcriptional signature. (C) UMAP plots of raGFP-Wnt1 CAFs colored by the raGFP-Wnt1 CD34⁻ vs CD34⁺ CAF signature generated by bulk differential gene expression analysis ($n = 3$) or *Cd34* expression. Sign., transcriptional signature. (D) Violin plot for enrichment of CD34⁻ CAF signature generated by bulk differential gene expression analysis ($n = 3$) in each raGFP-Wnt1 CAF cluster ($C0^T$–$C6^T$). ***$p < 0.001$, Wilcoxon rank-sum test. (E) Box plots for the enrichment of the mouse CAF transcriptional signatures for clusters $C0^T$–$C5^T$ in normal human fibroblasts ($n = 13$) or in CAFs from different human breast cancer subtypes. ER estrogen receptor ($n = 13$). HER2 human epidermal growth factor receptor 2 ($n = 6$). TNBC triple-negative breast cancer ($n = 8$). Box plots show quartiles, minimum and maximum. Sign., transcriptional signature. (F) Bar plot of top KEGG upregulated pathways in each specific myCAF cluster ($C0^T$, $C1^T$, and $C4^T$) vs all other CAF clusters. Down- or up-regulated genes for each cluster were obtained by pseudo-bulk differential gene expression analysis. (G) Violin plots for the enrichment of the senescence signature (Fridman and Tainsky, 2008) in each tumor cluster ($C0^T$–$C6^T$). ***$p < 0.001$, Wilcoxon rank-sum test. (H) UMAP plots of normal fibroblasts, raGFP-Wnt1 hyperplasia-associated fibroblasts (HAFs), and raGFP-Wnt1 cancer-associated fibroblasts (CAFs) colored by *Cdkn2a* expression. (I) Mean-difference plot showing differentially expressed genes between raGFP-Wnt1 CD34⁺ and CD34⁻ CAFs analyzed by bulk RNA-seq ($n = 3$). Significantly upregulated and downregulated genes are shown as red and blue dots, respectively. (J) Heatmap of expression of senescence-associated genes in raGFP-Wnt1 CD34⁻ vs CD34⁺ CAFs (bulk RNA-seq, $n = 3$). (K) Representative brightfield images of β-galactosidase staining of raGFP-Wnt1 CD34⁺ or CD34⁻ CAFs treated with vehicle or etoposide. Scale bars, 100 μm. (left). Quantification of the percentage of β-galactosidase (β-gal) positive cells (right). Each dot represents a technical duplicate, $n = 4$ independent experiments with two independent sets of primary CAFs. Error bars, mean ± s.e.m., **$p < 0.01$,****$p < 0.0001$, ordinary one-way ANOVA. Source data are available online for this figure.

Interestingly, increased Hh signaling was evident along the normal differentiation hierarchy, suggesting that previously reported roles of stromal Hh in mammary gland morphogenesis (Monkkonen et al, 2017; Zhao et al, 2017) are likely to be driven by specialized fibroblasts.

Prior studies on the adult mammary gland have identified two primary fibroblast subsets based on CD26 (*Dpp4*) expression (Houthuijzen et al, 2023; Morsing et al, 2016; Yoshitake et al, 2022), where high expression of *Dpp4* distinguishes the pan-tissue *Pi16*⁺ mesenchymal progenitor cells (Buechler et al, 2021). The positive correlation noted between *Cd34* and *Dpp4* expression through development suggests that these markers may be interchangeable, and we further demonstrate the utility of CD34 as a marker to enable the isolation of mesenchymal progenitor cells. Interestingly, mesenchymal progenitor cells were identified to be the most responsive to a potent hormonal stimulus, consistent with transcriptional changes occurring in the fibroblast compartment after hormone treatment (Kanaya et al, 2019). Fibroblasts might also respond to endogenous hormonal fluctuations during the estrous cycle and breast epithelial-associated fibroblasts have been observed to change with menopausal status (Pal et al, 2021).

The integrative analysis of mouse mammary fibroblasts across post-natal development revealed four specialized fibroblast subtypes in addition to mesenchymal progenitor cells. These fibroblast clusters were evident at most stages of mammary gland morphogenesis, however, striking changes were seen in the transition from puberty to the adult. While the adipose-associated stromal populations, *Cd36*⁺ committed preadipocytes and *Gdf10*⁺ adipogenesis-regulatory cells, were abundant in puberty, *Mgp*⁺ and *Crabp1*⁺ 'ECM remodeling' fibroblasts were lacking. During involution, we observed a subpopulation of *S100a4*⁺ fibroblasts expressing *Col14a*, that may contribute to the collagen-rich and immunosuppressive pro-tumorigenic microenvironment of the involuting mammary gland (Guo et al, 2017). The stromal compartment during pregnancy was similar to that in the adult, except for the presence of a rare cycling population, but may conceivably change upon consecutive pregnancies. In hyperplastic mammary glands, we identified two altered fibroblast subtypes: *Bmp5*⁺ lobular-like and *Crabp1*⁺ 'ECM remodeling' fibroblasts. *Bmp5*⁺ fibroblasts represent a transcriptionally unique stromal population reminiscent of human breast lobular fibroblasts (Morsing et al, 2016) that decline in their relative proportion

during mammary tumorigenesis. On the other hand, *Crabp1*⁺ 'ECM remodeling' fibroblasts have been previously detected in tumors (Sebastian et al, 2020) but not in normal mammary tissue. Our data indicate a periductal location for CRABP1⁺ fibroblasts and show that these cells profoundly increase after acute hormonal stimulation but are most prevalent in hyperplastic mammary glands. *Crabp1*⁺ fibroblasts have been identified in skin wound regeneration (Guerrero-Juarez et al, 2019; Sinha et al, 2022; Abbasi et al, 2020) and CRABP1 regulation of retinoic acid signaling is implicated in fibroblast to myofibroblast differentiation (Wang et al, 2020). However, αSMA⁺ fibroblasts were not evident in the normal homeostatic mammary gland nor in hyperplastic tissue and were only observed in tumors.

Dynamic changes within the fibroblast compartment occur in late mammary tumorigenesis. In mouse mammary tumors, we observed subpopulations of myCAFs, iCAFs, and cycling CAFs, similar to those in human breast tumors (Liu et al, 2022; Wu et al, 2020). Notably, CD34 expression was sufficient to distinguish the myCAF and iCAF populations, where CD34⁺ cells were transcriptionally similar to iCAFs, and CD34⁻ cells were similar to myCAFs, compatible with recent findings (Houthuijzen et al, 2023). Moreover, transcriptional signatures for CD34⁺ and CD34⁻ CAF clusters were enriched in human normal fibroblasts and breast tumors, respectively. Thus, a gradient of CD34 expression can resolve different fibroblast subpopulations in both normal (CD34hi vs CD34lo) and cancerous tissue (CD34⁺ vs CD34⁻). CD34⁻ myCAFs express high levels of *Lrrc15*⁺, a marker of immunosuppressive fibroblasts that mediates resistance to immune checkpoint blockade in pancreatic cancer, and *Col12a1*, identified as a potential driver of invasion in breast cancer (Krishnamurty et al, 2022; Dominguez et al, 2020; Papanicolaou et al, 2022; Buechler et al, 2021).

Interestingly, a subset of CD34⁻ myCAFs was found to be transcriptionally and phenotypically analogous to the recently discovered senCAFs (Ye et al, 2024). Notably, we identified Wnt9a, a poorly understood member of the Wnt family of ligands, as a regulator of senescence in these cells. Wnt9a can promote renal fibrosis by accelerating cellular senescence (Luo et al, 2018), but a role for Wnt9a in modulating senCAF-secreted extracellular matrix production is yet to be determined. Whether CD34⁻ senCAFs can revert their senescent phenotype in vivo remains unclear, as the activation of such cells appears to be an inefficient process (Ramponi et al, 2025). Collectively, this single-cell atlas of

mammary fibroblasts profiled at different stages of normal development and neoplasia sheds light on the fibroblast hierarchy and potential functional markers that could be used for the reprogramming of CAFs to aid in breast cancer treatment.

# Methods

### Reagents and tools table

| Reagent/resource | Reference or source | Identifier or catalog number |
|---|---|---|
| **Experimental models** | | |
| C57BL/6J (*M.musculus*) | The Jackson Laboratory | Cat# JAX:000664 |
| FVB/NJ (*M.musculus*) | The Jackson Laboratory | Cat# JAX:001800 |
| *Pdgfra*^H2B-GFP (*M.musculus*) | The Jackson Laboratory | Cat# JAX:007669 |
| MMTV-Wnt1 (*M.musculus*) | The Jackson Laboratory | Cat# JAX:002870 |
| *Gli1*-rtTA-GFP (*M.musculus*) | This study | This study |
| **Recombinant DNA** | | |
| U6-gRNA-PGKpuro2-BFP | Addgene | Cat#: 50946 |
| **Antibodies** | | |
| Armenian Hamster IgG FITC anti-mouse CD29 (clone HMβ1-1) | Biolegend | Cat# 102206 |
| Armenian Hamster IgG APC-Cy7 anti-mouse CD29 (clone HMβ1-1) | Biolegend | Cat# 102226 |
| Armenian hamster Pacific blue anti-mouse CD24 (clone M1/69) | Biolegend | Cat# 101820 |
| Rat IgG2a PE-Cy7 anti-mouse CD14 (clone Sa14-2) | Biolegend | Cat# 123316 |
| Rat IgG2a APC anti-mouse CD31 (clone 390) | Biolegend | Cat# 102410 |
| Rat IgG2a Biotin anti-mouse CD31 (Clone MEC13.3) | Biolegend | Cat# 102504 |
| Rat IgG2b APC anti-mouse CD45 (clone 30-F11) | Biolegend | Cat# 103112 |
| Rat IgG2b Biotin anti-mouse CD45 (Clone 30-F11) | Biolegend | Cat# 103104 |
| Rat IgG2b APC anti-mouse TER-119 (clone TER-119) | Biolegend | Cat# 116212 |
| Rat IgG2b Biotin TER-119 (clone TER-119) | Biolegend | Cat# 116204 |
| Rat IgG2a PE-Cy7 anti-mouse CD140a (clone APA5) | Biolegend | Cat# 135911 |
| Rat IgG2a PE anti-mouse CD140b (clone APB5) | Biolegend | Cat# 136006 |
| Syrian Hamster IgG APC anti-mouse Podoplanin (clone 8.1.1) | Biolegend | Cat# 127410 |
| Streptavidin BV650 | BD Biosciences | Cat# 563855 |
| Mouse anti-alpha-SMA (clone 1A4) | Abcam | Cat# ab7817 |
| Rabbit anti-Collagen IV (polyclonal) | Abcam | Cat# ab19808 |
| Rabbit anti-CRABP1 (clone D7F9T) | Cell Signalling Technologies | Cat# 13163 |

| Reagent/resource | Reference or source | Identifier or catalog number |
|---|---|---|
| Rabbit anti-Keratin 14 (polyclonal) | Thermo Fisher Scientific | Cat# LBVRB-9020-P0 |
| Sheep anti-human progesterone receptor (polyclonal) | R&D Systems | Cat# AF5415 |
| Chicken anti-GFP (polyclonal) | Abcam | Cat# ab13970 |
| Rat IgG2a monoclonal anti-E-cadherin (clone ECCD-2) | Thermo Fisher Scientific | Cat# 13-1900 |
| Goat anti-chicken IgY (H + L) Alexa Fluor 488 | Thermo Fisher Scientific | Cat# A-11039 |
| Donkey anti-rabbit IgG (H + L) Alexa Fluor 647 | Thermo Fisher Scientific | Cat# A-31573 |
| Donkey anti-sheep IgG (H + L) Alexa Fluor 555 | Thermo Fisher Scientific | Cat# A-21436 |
| Goat anti-rat IgG (H + L) Alexa Fluor 555 | Thermo Fisher Scientific | Cat# A-21434 |
| Goat anti-mouse IgG (H + L) Alexa Fluor 555 | Thermo Fisher Scientific | Cat# A-21422 |
| Rabbit anti-p21 (polyclonal) | Cell Signaling Technologies | Cat# 64016 |
| Rabbit anti-p16 (polyclonal) | Proteintech | Cat# 10883-1-AP |
| Mouse anti-alpha-tubulin (clone DM1A) | Merck | Cat# T6199 |
| Mouse anti-Vinculin (clone hVIN-1) | Merck | Cat# V9131 |
| **Oligonucleotides and other sequence-based reagents** | | |
| PCR primers | This study | Table 2 |
| qPCR primers | This study | Table 2 |
| CRISPR single guide RNAs | Sanger Arrayed Whole Genome Lentiviral CRISPR Library | Table 2 |
| **Chemicals, Enzymes and other reagents** | | |
| Trypsin (2.5%) | Thermo Fisher Scientific | Cat# 15090-046 |
| Dispase II (neutral protease, grade II) | Sigma Aldrich | Cat# 4942078001; CAS Number: 9001-92-7 |
| Gibco DMEM/F12, GlutaMAX Supplement | Thermo Fisher Scientific | Cat# 10565018 |
| Gibco Penicillin-Streptomycin | Thermo Fisher Scientific | Cat# 15140122 |
| Insulin (Roche) | Sigma Aldrich | Cat# 11376497001; CAS Number: 11061-68-0 |
| Cyclodextrin-encapsulated hydrocortisone | Sigma Aldrich | Cat# H0396; PubChem Substance ID: 24895401 |
| Epidermal growth factor (EGF) | Sigma Aldrich | Cat# E9644; CAS Number: 62229-50-9 |
| Cholera Toxin | Sigma Aldrich | Cat# C-8052; CAS Number: 9012-63-9 |

| Reagent/resource | Reference or source | Identifier or catalog number |
|---|---|---|
| Trypan Blue 0.4% | Thermo Fisher Scientific | Cat# T10282; CAS Number: 72-57-1 |
| Deoxyribonuclease I (DNAse I) | Worthington Biochemical Corp | Cat# LS002140; CAS Number: 9003-98-9 |
| Clostridiopeptidase A (Collagenase) | Sigma Aldrich | Cat# C9891; CAS Number: 9001-12-1 |
| Hyaluronate 4-glycanohydrolase (Hyaluronidase) | Sigma Aldrich | Cat# H3506; CAS Number: 37326-33-3 |
| EGTA | Sigma Aldrich | Cat# E0396; CAS Number: 67-42-5 |
| 7-Aminoactinomycin D (7-AAD) | Sigma Aldrich | Cat# A9400; CAS Number: 7240-37-1 |
| DPBS, no calcium, no magnesium | Gibco | Cat# 14190-144 |
| UltraPure™ DNase/RNase-Free Distilled Water | Invitrogen | Cat# 10977015 |
| HEPES pH 7.5 | Sigma Aldrich | Cat# H3375 |
| Triton X-100 | Sigma Aldrich | Cat# T9284; CAS Number: 9036-19-5 |
| Tween 20 | Sigma Aldrich | Cat# P7949; CAS Number: 9005-64-5 |
| Bovine serum albumin | Sigma Aldrich | Cat# A7906; CAS Number: 9048-46-8 |
| Trizma (Tris) base (2-Amino-2-(hydroxymethyl)-1,3-propanediol) | Sigma Aldrich | Cat# T1699; CAS Number: 77-86-1 |
| Ethylenediaminetetraacetic acid, disodium salt dihydrate (EDTA) | Sigma Aldrich | Cat# 03690; CAS Number: 6381-92-6 |
| Paraformaldehyde (PFA) powder | Sigma Aldrich | Cat# P6148; CAS Number: 30525-89-4 |
| 4′,6-Diamidino-2-phenylindole dihydrochloride (DAPI) | Thermo Fisher Scientific | Cat# 62248; CAS Number: 28718-90-3 |
| Growth factor-reduced Matrigel | Merck | Cat# 3470 |
| **Software** | | |
| R | R Project for Statistical Computing | RRID: SCR_001905 |
| R packages Rsubread, edgeR, limma, Glimma, plyranges, Iranges, rtracklayer, BSgenome.Mmusculus.UCSC.mm10 | Bioconductor | RRID: SCR_006442 |
| R packages gplots, ComplexUpset, Pheatmap, viridisLite, dplyr, readr, writexl, purrr, ggplot2, tools, nplyr, magrittr, stringr, tidyverse | CRAN | RRID: SCR_003005 |
| FlowJo (Version 10.10) | BD Biosciences | RRID: SCR_008520 |
| Prism 10 | GraphPad | RRID:SCR_002798 |

| Reagent/resource | Reference or source | Identifier or catalog number |
|---|---|---|
| FIJI/ImageJ | https://imagej.nih.gov/ij/ | RRID:SCR_003070 |
| ZEISS ZEN Microscopy Software | Zeiss | RRID: SCR_013672 |
| Leica Application Suite X | Leica | RRID: SCR_013673 |
| Imaris 8.2, 8.4 and 9.5 with XT | Oxford Instruments | RRID: SCR_007370 |
| **Other** | | |
| Illumina NextSeq 2000 | Illumina | WEHI |
| Illumina NextSeq 500 | Illumina | WEHI |
| BD FACSAria Fusion | BD Biosciences | WEHI |
| BD LSRFortessa X-20 Cell Analyzer | BD Biosciences | WEHI |
| Zeiss LSM 880 Fast Airyscan Confocal microscope | Zeiss | WEHI |
| Zeiss LSM 980 Fast Airyscan Confocal microscope | Zeiss | WEHI |
| Leica Stellaris Confocal microscope | Leica | WEHI |

## Mice

Vera Ramaciotti Laboratory (Kew) provided FVB/NJ and C57BL/6 mice. *Pdgfra*[H2B-GFP] mice (C57BL/6) were kindly provided by Dr. C Biben (WEHI). The *Gli1*-rtTA-GFP mouse strain (C57BL/6 background) was created at the Australian Regenerative Medicine Institute (ARMI, Monash University), where the start codon of the *Gli1* gene was replaced by the rtTA-IRES-GFP-polyA cassette. Heterozygous KI/+ reporter mice were used for all experiments in both mouse lines. *Pdgfra*[H2B-GFP] and *Gli1*-rtTA-GFP mice were crossed to the MMTV-*Wnt1* transgenic strain (FVB/NJ, Tsukamoto et al, 1988) to generate raGFP-Wnt1 and Gli1-Wnt1 mouse lines, respectively. MMTV-cre *Trp53*[fl/+] *Brca2*[fl/fl] mice have been previously described (Joyce et al, 2024).

All mice were bred and maintained in the WEHI animal facility according to institutional guidelines. All experiments were approved by the WEHI Animal Ethics Committee (2020.005, 2022.061, and 2022.070). Littermate controls were used, when possible, as stated in the legends. Pubertal mice were between 4.5 and 6 weeks old, as stated in the legends. For timed pregnancies, adult female mice were mated, scored by the presence of vaginal plugs, and confirmed by examination of embryos at the time of mammary gland collection. All pregnancies analyzed were first pregnancies. For lactation experiments, adult female mice were mated, and six offspring for each litter were maintained. For early involution experiments, six pups were allowed to feed for 10 days before separating the lactating female for 4 days prior to collection. EdU (0.2 mg per 10 g body weight, Thermo Fisher Scientific Invitrogen #A10044) was injected 2 h before mammary gland collection. For hormonal treatment, the synthetic progesterone analog medroxyprogesterone acetate (MPA) (15 mg, 90-day release; Innovative Research of America, # NP-161) and custom-made estradiol (E) 0.5 mg pellets or Placebo (control) were implanted subcutaneously into 8-week-old C57BL/6 mice. Mammary glands from control or hormone-treated mice were collected 7 days later.

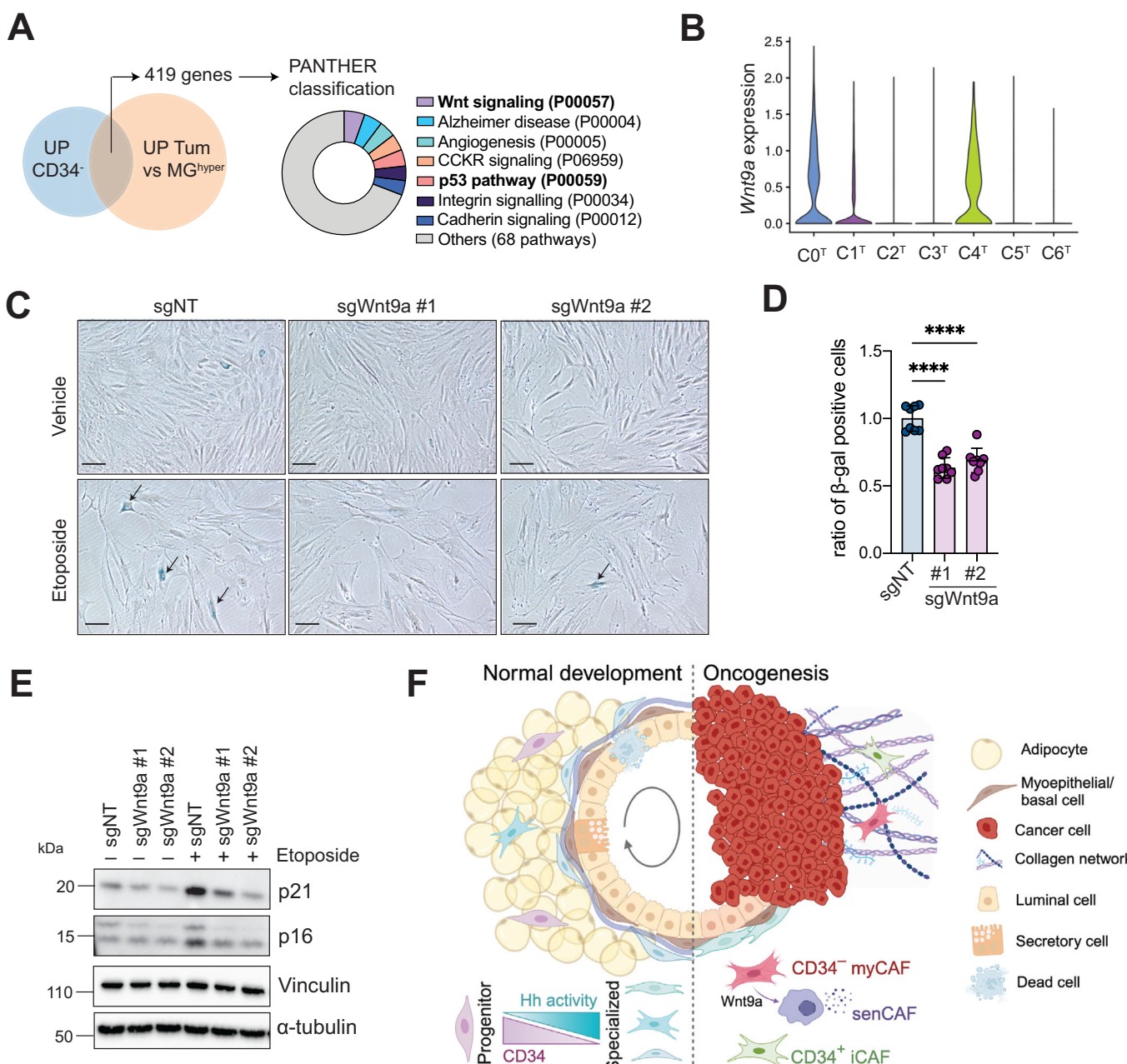

**Figure 7. Wnt9a contributes to the senescent phenotype of CD34⁻ myCAFs.**

(A) Venn diagram showing the overlap between significantly upregulated genes in raGFP-Wnt1 CD34⁻ vs CD34⁺ CAFs (bulk RNA-seq, $n = 3$) and in raGFP-Wnt1 tumors vs hyperplastic glands (all fibroblasts, by pseudo-bulk scRNA-seq, $n = 2$) (left). PANTHER pathway analysis showing the seven most significantly enriched pathways in a donut plot (right). (B) Violin plot showing *Wnt9a* expression in each raGFP-Wnt1 CAF cluster (C0ᵀ–C6ᵀ). (C) Representative brightfield images of β-galactosidase staining of raGFP-Wnt1 CD34⁻ myCAFs CRISPR-edited for Wnt9a (sgWnt9a#1 and #2) or control guide (sgNT, non-target) treated with either vehicle or etoposide. Arrows depict β-gal⁺ cells. Scale bars, 100 μm. (D) Quantification of the percentage of β-galactosidase-positive raGFP-Wnt1 CD34⁻ myCAFs, either Wnt9a-KO or control (sgNT, non-target) treated with etoposide. Values were normalized to the average of the control (etoposide-treated sgNT) for each experiment. Each dot represents a technical duplicate, $n = 4$ experiments with two independent sets of primary CAFs. Error bars, mean ± s.e.m., ****$p < 0.0001$, ordinary one-way ANOVA. (E) Western blot analysis showing expression of p21 and p16 in control (sgNT) or Wnt9a-KO raGFP-Wnt1 CD34⁻ myCAFs treated with vehicle or etoposide ($n = 3$). Vinculin and α-tubulin were used as a loading control. (F) Model of the evolving mammary fibroblast hierarchy during normal post-natal development and mammary oncogenesis. Created with BioRender.com. Hh Hedgehog, myCAF myofibroblastic CAF, senCAF senescent CAF, iCAF inflammatory CAF. Source data are available online for this figure.

## Mammary cell preparation and flow cytometry

Mammary glands or tumors were collected from female mice, and single-cell suspensions were prepared as previously described (Shackleton et al, 2006). Cell suspensions were subjected to cell labeling with fluorophore-conjugated primary antibodies or sequential labeling with biotin-conjugated primary and then Streptavidin-Brilliant Violet 650 (1:400, #563855) from BD Bioscience. Primary FACS antibodies from Biolegend included TER-119-APC (1:80, #116212), CD31-APC (1:40, #102510), CD45-APC (1:100, #103112), CD24-PB (1:200, #101820), CD29-FITC (1:200, #102206), CD29-APC-Cy7 (1:200, #102226), CD14-PE-Cy7 (1:200, #123316), CD140a-PE-Cy7 (1:60, #135911), CD140b-PE (1:60, #136006) and Pdpn-APC (1:200, #127410). Primary FACS antibodies, CD34-FITC (1:200, #11-0341-82) and CD34-biotin (1:200, #13-0341-82), were purchased from Thermo Fisher. For panels with lineage-biotinylated antibodies, TER-119-biotin (1:80, #553672), CD31-biotin (1:40, #553371) and CD45-biotin (1:100, #553078) were from BD Bioscience. To exclude dead cells, cells were resuspended in 0.2 µg/ml 7-AAD (Sigma, #A9400) before analysis. Cell sorting was performed on a FACS Aria Fusion Flow Cytometer or analyzed using an LSRFortessa™ X-20 (Becton Dickinson, BD). Downstream data analysis was performed using FlowJo 10.8 (BD).

## scRNA-seq sample preparation

For scRNA-seq, freshly sorted stromal fractions (Lineage⁻CD24⁻) were obtained from five post-natal developmental stages (2 C57BL/6 female mice per condition), control or hormone-stimulated (placebo control or MPA + E, $n = 2$, two C57BL/6 female mice per replicate) as well as freshly sorted GFP⁺ cells from raGFP-Wnt1 mammary tumors and paired hyperplastic mammary glands ($n = 2$). Developmental timepoints were puberty (4.5-week old), adult (virgin 9-week old), pregnancy (14.5 days), lactation (10 days), and involution (4 days). Total cells from $MMTV\text{-}cre^{T/+}$ $Trp53^{fl/+}$ $Brca2^{fl/fl}$ tumors and hyperplastic tissue from the same tumor-bearing mice were captured for scRNA-seq ($n = 2$). The 10x Genomics Chromium kit 3' (v3.1) was used for single-cell capture and cDNA preparation according to the 10x Single Cell 3' Protocol. Freshly sorted cells were manually counted, and equal numbers per sample (1000 cells/µl) were loaded for capture. Sequencing was carried out on an Illumina NextSeq2000 with four libraries per run.

## scRNA-seq analysis

The 10x Genomics Chromium sequencing data were pre-processed using Cell Ranger (v6.1.2). The filtered count matrix from Cell Ranger was used for downstream analysis. For quality control, cells with a low number of genes (<500) or a very high number of genes as well as a high mitochondria read percentage (>10%) were removed. The Seurat (v4.1.1) (Hao et al, 2021) standard pipeline was then performed on each sample. The raw count data was first normalized using the default log normalization method in the NormalizeData function. Top 1000 highly variable genes (HVGs) were selected by FindVariableFeatures function. The normalized data of the HVGs were then scaled by the ScaleData function for dimension reduction analysis. Principal component analysis (PCA) was performed to reduce the dimensions. Uniform manifold

approximation and projection (UMAP) was performed on the first 30 principal components to further project the data onto two dimensions. The principal components were also used to predict the potential doublets using scDblFinder (v1.10.0) (Germain et al, 2021). FindNeighbors and FindClusters functions were used to identify clusters with an appropriate resolution value. Fibroblast and other cell types were annotated using known marker genes (Table 1). Predicted doublets and contamination cells in each sample were removed, and only fibroblasts were kept for integration analysis. The anchor-based integration method in Seurat was used to integrate normal, hormone-treated, hyperplasia-, and cancer-associated fibroblast samples individually. FindAllMarkers function was used to identify the marker genes of each cluster. The average expression of each signature gene list of interest was used to indicate the signature score. Heatmaps were generated using the DoHeatmap function in Seurat, and the term "Expression" refers to row-scaled z-scores of log-normalized counts.

Pseudo-bulk samples were constructed by aggregating the counts of cells in each cluster of each sample. Differential analysis between each cluster and an average of other clusters was performed on the pseudo-bulk samples using the quasi-likelihood pipeline in edgeR (v3.38.1) (Chen et al, 2016) for normal fibroblasts and using the limma-voom pipeline (Law et al, 2014) for cancer-associated fibroblasts. Upregulated genes in each cluster of normal fibroblasts were used as signature genes for that cluster. The Kumar human fibroblast data were downloaded (Kumar et al, 2023), and average expression of normal cluster signatures was calculated and probed against the expression profile of fibroblasts from high and low breast density. Differential analysis between normal, hyperplasia-, and cancer-associated fibroblasts was performed on the pseudo-bulk samples using the limma-voom pipeline. All fibroblasts were aggregated into a pseudo-bulk sample for each of the normal (adult and puberty), hyperplasia, and cancer-associated fibroblast samples. Gene ontology (GO) analysis and KEGG pathway analysis were performed using the goana function and kegga function in limma (v3.55.5) (Ritchie et al, 2015), respectively. The barcode plot function in limma was used to demonstrate the enrichment of a gene set in a specific cluster.

Trajectory analysis using monocle3 (v1.2.9) (Cao et al, 2019) was performed on integrated normal fibroblasts. A start node was chosen to order cells along the trajectory and obtain pseudo-time. The start node was also used to split the trajectory into three sub-trajectories using the choose_graph_segments function in monocle3. Pseudo-time was split into bins with an interval length of 1. Cells in each bin were aggregated into a pseudo-bulk sample, and the average pseudo-time of all cells in the bin was used to represent the pseudo-time for that pseudo-bulk sample. Pseudo-bulk time course analysis (Cheng et al, 2023) was performed on the pseudo-bulk samples for each sub-trajectory to identify the DE genes along the time course. The average expression of selected genes of pathways of interest was calculated to investigate whether the pathways are changed along the pseudo-time course.

Human breast tissue single-cell RNA-seq profiles of normal donors and patients diagnosed with ER+, HER2+ or triple-negative tumors were obtained from a previous study (Pal et al, 2021). Fibroblast and CAF cell populations were extracted from the normal and the tumor samples, respectively, and pseudo-bulk gene expression profiles were generated by aggregating the gene

expression of all cells from the same patient. Signature scores for each pseudo-bulk sample were calculated using the average log-CPM values of human homolog marker genes for the relevant cell clusters and visualized in box plots. When calculating CD34⁻ vs CD34⁺ gene signatures in the pseudo-bulk samples, signature scores were computed using the weighted average log-CPM, with log-FC values of differentially expressed genes serving as the weights.

## Bulk RNA sequencing and data analysis

FACS-sorted Lin⁻CD24⁻Pdgfrα⁺ CD34ʰⁱ and CD34ˡᵒ fibroblasts from three pools of two 6-week-old FVB/NJ mice each or CD34⁻ and CD34⁺ GFP⁺ CAFs from three raGFP-Wnt1 tumors were snap frozen, and later resuspended in QIAzol for RNA-extraction following the manufacturer's protocol for the miRNeasy micro kit (Qiagen #217084), including the on-column DNase digestion (Qiagen #79256). A minimum of 10 ng of total RNA was used for sequencing libraries following Illumina's TruSeq RNA v2 protocol (Illumina #RS-122-2001). Libraries were sequenced on an Illumina NextSeq 500. Raw RNA-seq fastq files were aligned against to the Mouse (GRCm38/mm10) reference genome and the reads were counted using Rsubread v2.2.6 and BioConductor v3.11.1 (Liao et al, 2019). Genes with no annotation and low counts across all libraries were removed using the filterByExpr function from edgeR (Chen et al, 2016). Normalized count data were obtained using the trimmed mean of M-values (TMM) normalization method implemented in edgeR to eliminate composition biases between libraries (Robinson and Oshlack, 2010). To estimate the negative binomial (NB) common dispersion, the estimateDisp function in edgeR was employed (Robinson et al, 2010). Differential expression analysis was performed using the voom transformation followed by linear modeling and empirical Bayes moderation for statistical tests (limma v3.44.3) (Law et al, 2014). To control for multiple testing, resulting $p$ values were adjusted using the false discovery rate (FDR) method of Benjamini and Hochberg (Benjamini and Hochberg, 1995). Heatmaps were generated using the heatmap.2 function in gplots package v3.1.0, and the term "Expression" refers to row-scaled z-scores of log-normalized counts. Pathway enrichment analysis was conducted using the camera gene set test in edgeR with gene sets sourced from MsigDB (Mm.c2.all.v7.1.entrez.rds cancer-related gene sets downloaded from https://bioinf.wehi.edu.au/), KEGG and GO databases (Kanehisa et al, 2023; Kanehisa and Goto, 2000; Kanehisa, 2019; Ashburner et al, 2000; Gene Ontology Consortium et al, 2023). Barcode plots were produced using the limma package to visualize enrichment results of top gene sets identified from pathway analysis.

## Three-dimensional confocal imaging of whole-mount tissue

Imaging was performed as previously described (Rios et al, 2019). Paraformaldehyde fixed tissue was incubated with primary antibodies overnight, then incubated with secondary antibodies overnight before washing and clearing with FUnGI. During secondary staining, the tissues were stained with DAPI (Thermo Fisher, #62248) and/or phalloidin conjugated to Alexa Fluor 555 (Invitrogen, #A34055) or Alexa Fluor 647 (Invitrogen, #A22287). For EdU labeling, the tissues were further incubated with Click-it Imaging 647 Kit (Invitrogen,

#C10340) following secondary staining and before clearing. Cleared tissue was dissected under a Leica M205A fluorescence stereomicroscope. Primary antibodies: rat monoclonal anti-E-Cadherin (Thermo Fisher #13-1900, clone ECCD-2), rabbit polyclonal anti-Keratin 14, (Thermo Fisher, #LBVRB-9020-P0), chicken polyclonal anti-GFP (Abcam, #ab13970), sheep anti-progesterone receptor (R&D, #AF5415). Secondary antibodies (1:500): donkey anti-rat Alexa Fluor 555 (Invitrogen, #A78945), donkey anti-rat Alexa 647 (Jackson Immuno, #712-605-153, donkey anti-chicken Alexa 488 (Jackson Immuno, #703-545-155), donkey anti-rabbit Alexa 555 (Invitrogen, #A31572), donkey anti-rabbit Alexa 647 (Invitrogen, #A-31573). For confocal microscopy image acquisition, tile scans of Z-stacks were acquired at a minimum optical section resolution of $1024 \times 1024$ using software-optimized Z-resolution and a pinhole size of 1 AU. Samples were imaged on a laser scanning confocal microscope: Leica SP8, Zeiss 880/980. Zeiss microscopes were equipped with 25x/0.8 multi-immersion, 40x/1.20 Oil DIC or 63x/1.40 Oil DIC objectives and the Leica SP8 equipped with 20x/0.75 multi-immersion, 40x/1.3 Oil DIC or 63x/1.40 Oil DIC objectives and PMT/HyD detectors. Images were stitched in LASX (Leica) or ZEN2010 (Zeiss) software, and image processing was followed by visualization in Imarisv9.9 (Bitplane).

## 2D immunostaining

Mammary glands were fixed in 4% (w/v) paraformaldehyde (PFA) for 24 h at 4 °C prior to paraffin-embedding and preparation of 4 μm sections. Mammary glands from pubertal 6-week-old mice were micro-dissected to enrich for terminal end buds (TEBS) before fixation. After antigen retrieval using citrate buffer (pH = 6.0), tissue sections were stained with the following primary antibodies: anti-α-SMA (Abcam, #ab7817), anti-collagen IV (Abcam, #ab19808), anti-GFP (Abcam, #ab13970) and anti-CRABP1 (CST, #13163). For secondary staining, biotinylated anti-rabbit IgG antibody (vector Laboratories) or Alexa Fluor antibodies (Invitrogen) together with DAPI (Thermo Fisher, #62248) were used for immunohistochemistry or immunofluorescence, respectively. CRABP1 immunostained sections were scanned at 20x/0.8 magnification using the Olympus VS200 Slide Scanner. The percentage of positive stromal cells was quantified using QuPath (Bankhead et al, 2017). After running cell detection, epithelial, distal stromal and periductal niches were manually annotated then subsequently classified using the Train Object Classifier option. Immunofluorescent confocal images were acquired with a Leica Stellaris microscope using 20x/0.75 or 40x/1.3 objectives and processed using FIJI (ImageJ2).

## 3D air-liquid interface (ALI) co-cultures

Freshly sorted 5,000 basal (CD24ˡᵒCD29ʰⁱ) or luminal progenitor (CD24ʰⁱCD29ˡᵒCD14⁺) cells were plated together with 15,000 freshly sorted or expanded fibroblasts (detailed in figure legends) on a 1:1 mix of mammary growth media (5% v/v FCS) and growth factor-reduced Matrigel (BD Pharmingen) on top of polyester membranes in 24-well plates (Merck #3470) at 37 °C, 5% $O_2$, 5% $CO_2$. After 24 h, FCS was reduced to 1% (v/v) and incubated for a further 6 days. Brightfield images were obtained using a ZEISS Axio Observer microscope with a 5x/0.16 objective. Organoids were counted using an automated macro after z-stack processing in FIJI (ImageJ2) (Joyce et al, 2024).

**Table 2.** Sequences of CRISPR single guide (sg) RNAs and primers used for PCR and quantitative PCR.

| | Sequence |
| --- | --- |
| OH_Wnt9a#1_Fw | GTGACCTATGAACTCAGGAGTCCTATCCTCCCTCTGACCCTGG |
| OH_Wnt9a#1_Rv | CTGAGACTTGCACATCGCAGCTGCACACATGACAGGGTGAG |
| OH_Wnt9a#2_Fw | GTGACCTATGAACTCAGGAGTCCTCTCTGACTTGGCTCAGTGC |
| OH_Wnt9a#2_Rv | CTGAGACTTGCACATCGCAGCCACTCATGCTTACGGCCTCC |
| sgWnt9a#1 | GCAGTTCCAGCGCTCAAAGCGG |
| sgWnt9a#2 | GGTCAGAGGGAGGATAGTCAGG |
| Gapdh_Fw | CTTCACCACCATGGAGGAGGC |
| Gapdh_Rv | GGCATGGACTGTGGTCATGAG |
| mGli1_Fw | GCAACCTTCTTGCTCACACA |
| mGli1_Rv | GAAGGAATTCGTGTGCCATT |
| mGli2_Fw | CCAATGAGAAACCCTACATCTG |
| mGli2_Rv | TTCACATGCTTGCGGAGT |
| mPtch1_Fw | TGACAAAGCCGACTACATGC |
| mPtch1_Rv | AGCGTACTCGATGGGCTCT |
| mPtch2_Fw | TCCGAGTGGCTGTAATTG |
| mPtch2_Rv | GCTTCTCCTTGGTGTAGT |
| mCol15a1_Fw | ACACCCACAGTGACTCCCAAGA |
| mCol15a1_Rv | TCCTCATTGCCCACGATGTCTC |
| mCd34_Fw | CTGGGTAGCTCTCTGCCTGA |
| mCd34_Rv | AGAAGTCTCCGTGGTAGCAG |

## CRISPR-Cas9 editing

CD34⁻ CAFs were isolated from raGFP-Wnt1 tumors by FACS and cultured in DMEM (Gibco) supplemented with 10% (v/v) FCS (Thermo Fisher, #15140122). At passage three, cells were transduced with third-generation lentivirus produced in HEK293T cells. Plasmids for lentivirus production included FUCas9-Cherry and U6-gRNA/PGK-Puro-2A-BFP (Sanger arrayed whole genome lentiviral CRISPR library, Sigma) for control and Wnt9a sgRNAs. The guide sequences are listed in Table 2. Viral titers were defined by titration curves in HEK293T cells. Double-positive cells were sorted and used for downstream assays. The efficiency of guides was validated by next generation sequencing (MiSeq, Illumina) using PCR primers with overhang sequences for each sgRNA (Table 2), as previously described (Dekkers et al, 2020). DNA was isolated using the Allprep DNA/RNA Mini Kit (Qiagen, #80204).

## β-galactosidase assay

Fibroblasts were seeded at a density of 25,000 cells per well, or 12,500 cells per well, of a 6- or 12-well plate, respectively. Twenty-four hours after plating, cells were treated with etoposide (12.5 μM, Sapphire Bioscience #12092) or vehicle (H$_2$O) control for 24 h and allowed to recover for 4 days. Cells were then rinsed with DPBS (Gibco, #14190-144) and fixed with 1X fixative solution from the senescence β-galactosidase staining kit (CST, #9860) for 15 min. Cells were washed twice with PBS before incubation in the β-galactosidase staining solution (pH 6.0) for 17 h at 37 °C, in a dry

incubator without CO$_2$. The staining solution was removed, and cells were washed before imaging with a Nikon Eclipse TE200 microscope using the 10x/0.3 objective. Cells, either positive or negative for β-galactosidase, were manually counted in at least five randomly selected fields for each well (a minimum of 60 cells per well were counted). Senescence was quantified as the percentage of β-galactosidase-positive cells. Four independent experiments using two independent sets of cultured primary fibroblasts were performed in duplicate.

## Western blot analysis

Fibroblasts were treated with etoposide (20 μM, Sapphire Bioscience #12092) or vehicle and collected 24 h after treatment. Cells were lysed in RIPA buffer containing 1X complete mini protease inhibitor cocktail (Roche, #11836153001) and 1X Phos-STOP phosphatase inhibitor cocktail (Roche, #4906845001). Protein was quantified using the Pierce™ BCA Protein Assay kit (Thermo Fisher, #23225) and fractionated on NuPage 4–12% bis-tris polyacrylamide gels (Invitrogen) followed by transfer onto PVDF membranes using the iBlot™ 2 Dry Blotting system (Invitrogen). Membranes were probed with the following primary antibodies: anti-p21 (CST, #64016), anti-p16 (Proteintech, #10883-1-AP), anti-alpha-tubulin (Merck, #T6199), and anti-vinculin (Merck, #V9131). Membranes were then probed using HRP-conjugated anti-IgG secondary antibodies, developed in ECL (GE Healthcare Life Sciences), and imaged using the ChemiDoc™ Touch Imaging System. The intensity of the bands was quantified using FIJI (ImageJ2) and normalized to vinculin (loading control).

## Quantitative real-time PCR

RNA was purified from sorted or cultured fibroblasts using the RNeasy kit (Qiagen, #74004) with on-column DNase digestion (Qiagen, #79256) and used to generate cDNA with the SuperScript™ IV First-Strand Synthesis System (Thermo Fisher, #18091050). Quantitative RT-PCR was then carried out with SensiMix™ Hi-ROX (Meridian Bioscience, #QT605-05) on a Corbett Rotor-Gene 300 or QuantStudio 12 K Flex system. Forward and Reverse primer pairs for each gene can be found in Table 2.

## 2D colony-forming assays

For colony-formation assays, 200 freshly sorted basal/myoepithelial epithelial cells (CD24$^{lo}$CD29$^{hi}$) were plated together with irradiated primary fibroblasts in 24-well plates (Corning, #353047) in mammary growth medium (DMEM/F12 with glutamax, 10 ng/ml EGF, 5 µg/ml insulin, 0.5 µg/ml hydrocortisone, 20 ng/ml cholera toxin) with 5% (v/v) fetal calf serum (FCS) and changed to the same medium containing 1% (v/v) FCS the next day. FACS-sorted primary fibroblasts were obtained from mammary glands and cultured at 37 °C, 5% O$_2$, 5% CO$_2$ in DMEM Glutamax™ media (Gibco, #10569-010) supplemented with 10% (v/v) FCS (Thermo Fisher, #15140122). After two or three passages, primary fibroblasts were irradiated at 50 Gray before using them as feeders. Colonies were visualized using Giemsa staining (Sigma, #1092040) and imaged on a Zeiss Stemi 2000-C stereomicroscope.

## CellTiter-Glo® (CTG) luminescent cell viability assay

Matched HAFs and CAFs from three biological replicates or CRISPR-edited CD34$^-$ CAFs were seeded at a density of 500 cells per well of 96-well plates in triplicate. Cells were treated with CTG reagent from the CellTiter-Glo® Luminescent Cell Viability Assay kit (Promega, #G7570) according to the manufacturer's instructions, and luminescence was measured using a CLARIOstar Plus microplate reader (BMG Labtech). Luminescence was measured at 1, 3, 4, and 7 days after plating, and values were normalized to day 1.

## Mass spectrometry-based proteomics sample preparation

Cell supernatants (secretome) were collected from CD34$^{hi}$ and CD34$^{lo}$ fibroblasts (passage 5, $n = 3$ mice) after 3 h of serum starvation, filtered through a 0.22 µm filter, and prepared for mass spectrometry analysis using the FASP (filter-aided sample preparation) method (Wiśniewski et al, 2009), with the following modifications. Proteins were reduced with 10 mM Tris-(2-carboxyethyl) phosphine (TCEP), alkylated with 50 mM iodoacetamide, then digested with 1 µg sequence-grade modified trypsin gold (Promega) in 50 mM NH4HCO3 and incubated overnight at 37 °C. Peptides were eluted with 50 mM NH4HCO3 in two 40 µl sequential washes and acidified in 1% formic acid (FA, final concentration). Peptides were lyophilized to dryness using a CentriVap (Labconco) before being reconstituted in 30 µl of 0.1% FA/2% ACN ready for mass spectrometry analysis. Peptide samples (1 µl) were separated by reverse-phase chromatography (IonOpticks Aurora 75 µm ID, OD 360 µm × 15 cm length, 1.6-µm C18 beads) using a custom nano-flow HPLC system (Thermo Ultimate

300 RSLC Nano-LC, PAL systems CTC autosampler). The HPLC was coupled to a timsTOF Pro (Bruker) equipped with a CaptiveSpray source. Peptides were loaded directly onto the column at a constant flow rate of 400 nl/min with buffer A (99.9% Milli-Q water, 0.1% FA) and eluted with a 30-min linear gradient from 2 to 34% buffer B (90% ACN, 0.1% FA). The timsTOF Pro (Bruker) was operated in diaPASEF mode using Compass Hystar 5.1. The settings on the thermal ionization mass spectrometry (TIMS) analyzer were as follows: Lock Duty Cycle to 100% with equal accumulation and ramp times of 100 ms, and 1/K0 start 0.6 V·s/cm2 end 1.6 V·s/cm2, capillary voltage 1,400 V, Dry Gas 3 L/min, dry temp 180 °C. diaPASEF acquisition was performed using methods previously described (Meier et al, 2020). Briefly, 16× 25 m/z precursor isolation scans (resulting in 32 windows) were aligned across the m/z (400–1200) and ion mobility (0.8–1.4), with 1 Da overlap, and CID collision energy ramped stepwise from 20 eV at 0.8 V·s/cm$^2$ to 59 eV at 1.3 Vs/cm$^2$.

## Proteomics data processing and statistical analysis

DIA data were analyzed using DIA-NN 1.8 in the library-free model (Demichev et al, 2022). diaPASEF d. files were searched against reviewed sequences from mouse Uniprot Reference Proteome (downloaded November 2021) with the following settings: trypsin specificity, peptide length of 7–30 residues, cysteine carbidomethylation as a fixed modification, variable modifications set to n-terminal protein acetylation and oxidation of methionine, the maximum number of missed cleavages at 2. Mass accuracy was set to 10 ppm for both MS1 and MS2 spectra, a match between runs (MBR) was enabled, and filtering outputs were set at a precursor q-value <1%. Data processing and statistical analyses were performed using R software (version 4.2.1). Proteins lacking proteotypic precursors and those with a q-value greater than 0.01 were excluded from the analysis. Additionally, only proteins quantified in at least 50% of the replicates in any given condition were retained. This resulted in a total of 3243 proteins, which were included in the subsequent analysis. Protein intensities were log2-transformed. Missing values were imputed by drawing random numbers from a normal distribution (width = 0.3 and downshift = 1.8). The RUV-IIIC package (v.1.0.19) was used for data normalization. Negative controls were selected empirically, specifically proteins that remained invariant across all conditions (P value >0.5). Multivariate analysis, principal component analysis (PCA), was employed to identify any potential outliers. Pair-wise differential expression analysis was then carried out using limma (v. 3.50.1). A protein was considered significantly differentially expressed if it had an absolute log$_2$ fold change >0.5 and false discovery rate (FDR) of ≤5% following adjustment for multiple comparisons using the Benjamini–Hochberg (BH) correction.

## Statistics

No statistical methods were used to predetermine sample sizes, and no randomization was used. Data collection and analysis were not performed blinded to the experimental conditions. Statistical analyses of mean and variance were performed with Prism 9.5.1 (GraphPad Software,) and the statistical tests are indicated. All statistical comparisons were "two-sided" or "two-tailed". No data points were excluded.

## Data availability

Sequencing data have been deposited at GEO: scRNA-seq data, Superseries accession number GSE289925. The mass spectrometry proteomics data have been deposited to the ProteomeXchange Consortium via the PRIDE partner repository with the dataset identifier PXD060415. The data supporting the findings of this study are available within this paper, expanded view data, source data and supplementary information.

The source data of this paper are collected in the following database record: biostudies:S-SCDT-10_1038-S44318-025-00422-3.

## Peer review information

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

## Acknowledgements

We wish to thank the following WEHI facilities: Bioservices, Flow Cytometry Lab, Centre for Dynamic Imaging, Advanced Histotechnology, Genomics and Information Technology. We thank C Biben for the *Pdgfra*[H2B-GFP] mice, L Whitehead for image analysis, and M Samuel for discussions. This work was supported by the NBCF (IIRS-20-022), NHMRC IRIISS; the Victorian State Government Operational Infrastructure Support; the Breast Cancer Research Foundation (BCRF, USA). YC was supported by MRFF Investigator Grant (#1176199). GKS was supported by NHMRC Fellowship #1154970 and Investigator Grant #2025645. GJL and JEV by NHMRC Fellowships (GJL #1078730 and 1175960; JEV #1037230 and 1102742).

## Author contributions

**Rosa Pascual**: Conceptualization; Data curation; Formal analysis; Validation; Investigation; Methodology; Writing—original draft; Writing—review and editing. **Jinming Cheng**: Resources; Data curation; Software; Formal analysis; Investigation; Methodology. **Amelia H De Smet**: Formal analysis; Validation; Investigation; Methodology. **Bianca D Capaldo**: Investigation; Visualization; Methodology. **Minhsuang Tsai**: Investigation. **Somayeh Kordafshari**: Investigation. **François Vaillant**: Investigation. **Xiaoyu Song**: Investigation. **Göknur Giner**: Formal analysis. **Michael J G Milevskiy**: Investigation. **Felicity C Jackling**: Investigation. **Bhupinder Pal**: Investigation. **Toby Dite**: Formal analysis; Investigation. **Jumana Yousef**: Formal analysis; Investigation. **Laura F Dagley**: Formal analysis; Gordon K Smyth: Formal analysis; Supervision. Supervision. **Naiyang Fu**: Conceptualization; Investigation. **Geoffrey J Lindeman**: Conceptualization; Supervision; Funding acquisition. **Yunshun Chen**: Data curation; Formal analysis; Supervision; Funding acquisition; Writing—original draft. **Jane E Visvader**: Conceptualization; Supervision; Funding acquisition; Visualization; Writing—original draft; Project administration; Writing—review and editing.

Source data underlying figure panels in this paper may have individual authorship assigned. Where available, figure panel/source data authorship is listed in the following database record: biostudies:S-SCDT-10_1038-S44318-025-00422-3.

## Disclosure and competing interests statement

The authors declare no competing interests.

# Expanded View Figures

**Figure EV1.  Normal mammary fibroblast clusters.**

(**A**) UMAP plots showing cell types for individual stages before removing contaminant cells (see Table 1). (**B**) UMAP plot of integrated data from all developmental stages colored by *Bmp5* expression. (**C**) Violin plot showing enrichment of the C2 normal signature (lobular-like fibroblasts) in fibroblast clusters found in other tissues (Buechler et al, 2021). Upregulated genes in C2 were used to generate the lobular-like fibroblast signature. (**D**) Bar plot of top KEGG upregulated pathways in one cluster vs the rest. Down- or upregulated genes for each cluster were obtained by pseudo-bulk differential gene expression analysis. (**E**) UMAP plots of integrated data from all developmental stages colored by expression of selected top marker genes for C3. (**F**) Heatmap of gene expression showing the top 15 markers genes for clusters in C1 subclusters (C0$^{C1}$–C4$^{C1}$). (**G**) UMAP plot for the C1 sub-clustering analysis colored by *Fabp4* expression (**H**) UMAP plot colored by *Dlk1* expression (integrated data for all developmental stages). (**I**) Boxplots for enrichment score of C1 or C2 normal fibroblast transcriptional signatures (Sign.) in human breast tissue with high versus low mammographic density (MD) (Kumar et al, 2023). Box plots show quartiles, minimum and maximum. ***$p < 0.001$, Wilcoxon rank-sum test. (**J**) UMAP plots for an independent integrated scRNA-seq dataset for puberty and adult stages colored by cluster identity (clusters 0–4). (**K**) Dot plot visualization of expression of selected marker genes in each cluster (from Fig. EV1J). The orange square highlights highly expressed genes in cluster 1. The size of the dot encodes the percentage of cells within a cluster, while the color encodes the average expression levels across all cells within a cluster. (**L**) UMAP plots of the sub-clustering of cluster 1 (from Fig. 1EVJ) colored by cluster identity. (**M**) Violin plot for *Crabp1* expression in the cluster 1 subclusters (from Fig. 1EVI).

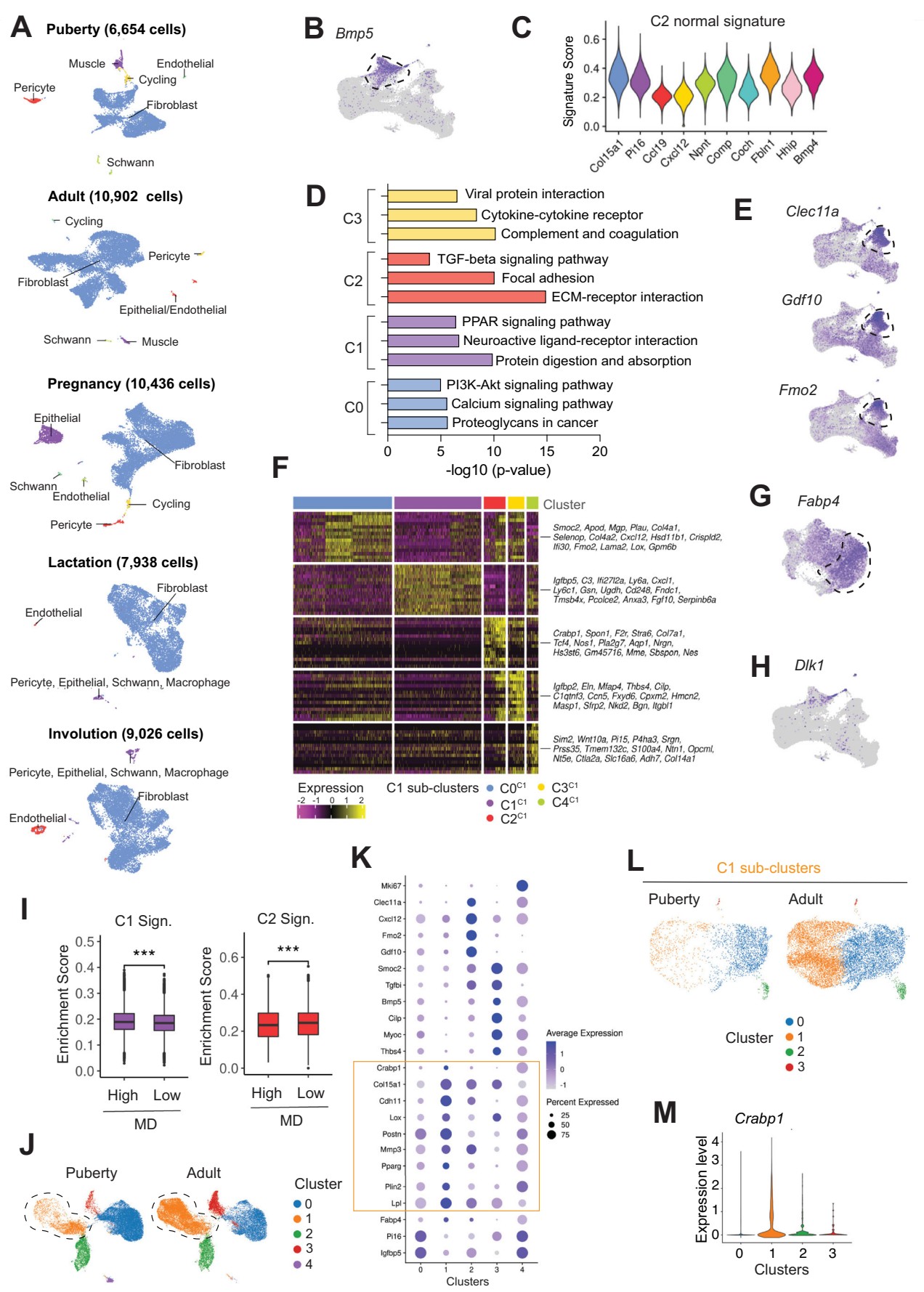

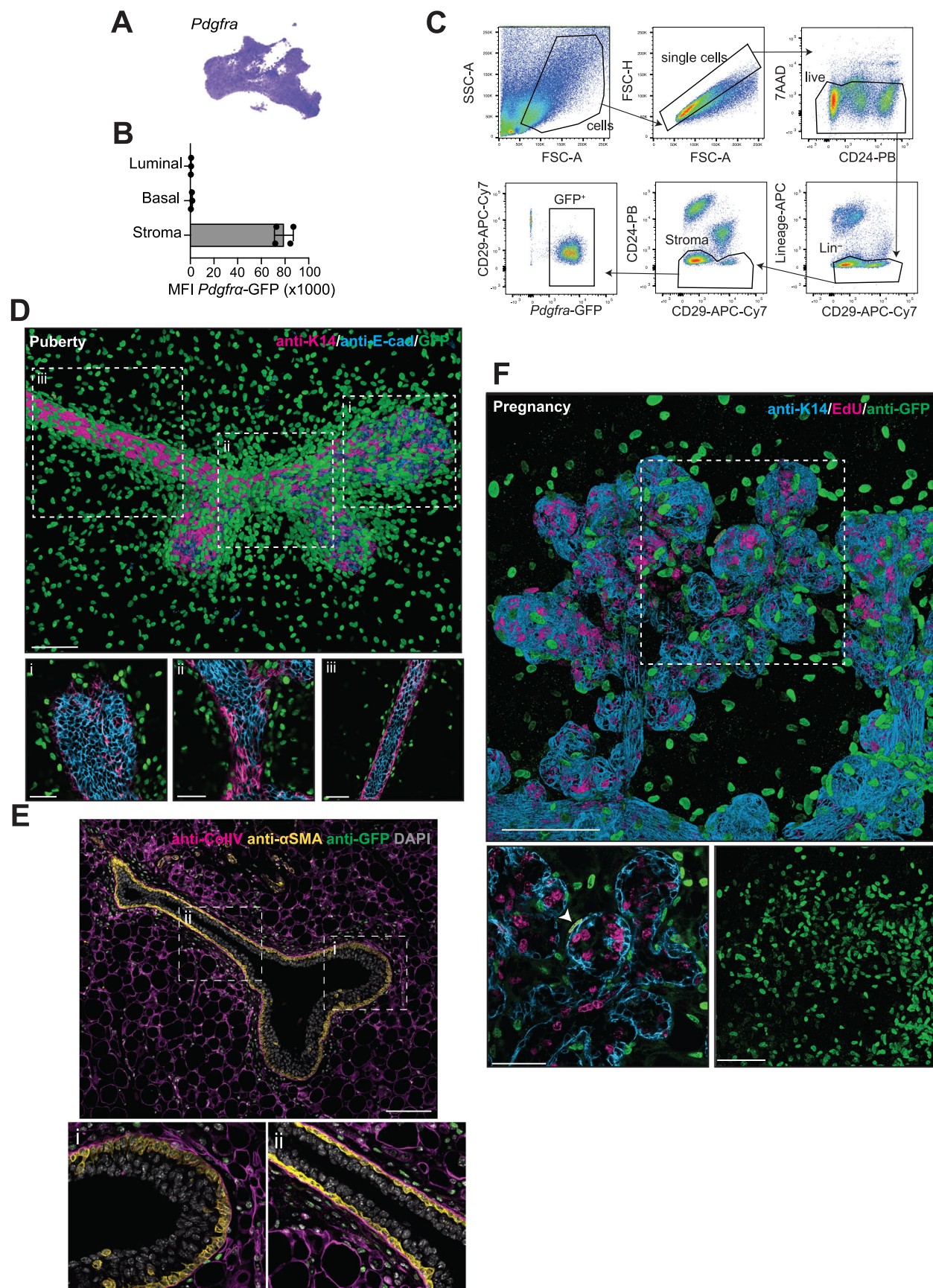

**Figure EV2. Characterization of fibroblasts using the *Pdgfra*[H2B-GFP] model.**

(A) UMAP plot colored by *Pdgfra* expression. (B) Mean fluorescence Intensity (MFI) of *Pdgfra*-GFP levels in the luminal, basal and stromal compartments. Each dot represents an individual mouse ($n = 4$). Error bars, mean ± s.e.m. (C) Gating strategy in the *Pdgfra*[H2B-GFP] reporter mouse model. (D) Representative 3D confocal image and sections of a pubertal *Pdgfra*[H2B-GFP] mammary gland (6-week old, $n = 3$). Keratin 14 (K14), E-cadherin and GFP shown in magenta, cyan and green, respectively. Scale bar, 100 μm for wholemounts and 50 μm for optical sections. (E) Representative confocal image showing a 6-week-old *Pdgfra*[H2B-GFP] mammary gland stained with anti-Collagen IV (ColIV, basement membrane, magenta), anti-α-SMA (myoepithelial, yellow), anti-GFP (fibroblasts, green) and DAPI (gray) ($n = 3$). Scale bar, 100 and 50 μm for zoomed-in images. (F) Representative 3D confocal image and optical sections from 12.5-day pregnant *Pdgfra*[H2B-GFP] mice injected with EdU 2 h prior to collection ($n = 2$). Keratin 14 (K14), EdU and GFP shown in cyan, magenta and green, respectively. Scale bar, 100 μm for wholemounts and 50 μm for optical sections.

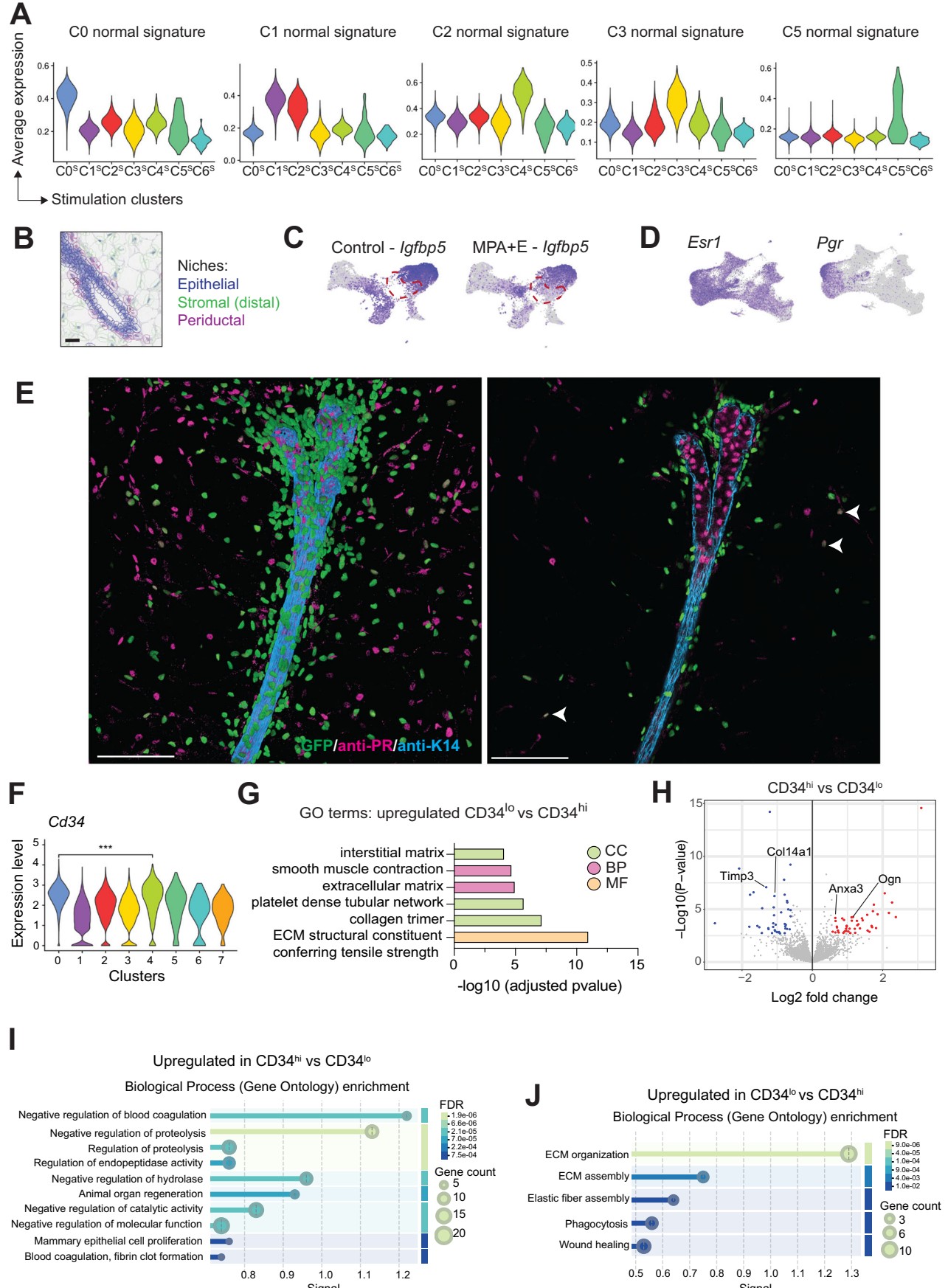

**A** C0 normal signature — C1 normal signature — C2 normal signature — C3 normal signature — C5 normal signature

Average expression / Stimulation clusters

**B** Niches:
Epithelial
Stromal (distal)
Periductal

**C** Control - *Igfbp5* / MPA+E - *Igfbp5*

**D** *Esr1* / *Pgr*

**E** GFP/anti-PR/anti-K14

**F** *Cd34* — Expression level / Clusters — ***

**G** GO terms: upregulated CD34^lo vs CD34^hi
interstitial matrix
smooth muscle contraction
extracellular matrix
platelet dense tubular network
collagen trimer
ECM structural constituent
conferring tensile strength
-log10 (adjusted pvalue)
CC / BP / MF

**H** CD34^hi vs CD34^lo
Col14a1 / Timp3 / Anxa3 / Ogn
-Log10(P-value) / Log2 fold change

**I** Upregulated in CD34^hi vs CD34^lo
Biological Process (Gene Ontology) enrichment
Negative regulation of blood coagulation
Negative regulation of proteolysis
Regulation of proteolysis
Regulation of endopeptidase activity
Negative regulation of hydrolase
Animal organ regeneration
Negative regulation of catalytic activity
Negative regulation of molecular function
Mammary epithelial cell proliferation
Blood coagulation, fibrin clot formation
Signal
FDR / Gene count

**J** Upregulated in CD34^lo vs CD34^hi
Biological Process (Gene Ontology) enrichment
ECM organization
ECM assembly
Elastic fiber assembly
Phagocytosis
Wound healing
Signal
FDR / Gene count

◀

**Figure EV3.  Fibroblast subsets during acute hormonal stimulation and characterization of CD34$^{hi/lo}$ fibroblasts.**

(A) Violin plots showing enrichment of signatures from normal fibroblast clusters in each stimulation cluster (C0$^S$-C6$^S$). Upregulated genes in each normal cluster (post-natal development) were used to generate cluster-specific signatures. (B) Snapshot of QuPath classification of mammary niches used in Fig. 3F. Scale bar, 20 μm. (C) Separate UMAP plots for control or MPA + E conditions colored by expression of *Igfbp5*. (D) UMAP plots of integrated data across post-natal development colored by expression of *Esr1* (left) and *Pgr* (right). (E) Representative 3D confocal image and optical section from a *Pdgfra*$^{H2B-GFP}$ mammary gland showing PR (progesterone receptor, magenta), K14 (Keratin 14, cyan), and GFP (fibroblasts, green) ($n = 2$). Scale bars, 100 μm. Double GFP$^+$PR$^+$ cells are highlighted with white arrowheads. (F) Violin plots of *Cd34* expression in each cluster. \*\*\*$p < 0.001$, Wilcoxon rank-sum test. (G) Bar plot of top Gene Ontology (GO) upregulated pathways in CD34$^{lo}$ vs CD34$^{hi}$ fibroblasts by bulk RNA-seq ($n = 3$). (H) Volcano plot illustrating the statistical significance ($-\log_{10} p$ value) versus the magnitude of proteomic changes ($\log_2$ fold change) in the secretomes of CD34$^{hi}$ versus CD34$^{lo}$ cultured fibroblasts by mass spectrometry analysis ($n = 3$). Proteins were deemed differentially regulated when the $\log_2$ fold change in protein expression was ≥1-fold and exhibited an adjusted $p$ value ≤0.05. (I) Gene ontology pathway analysis of significantly enriched local network clusters (STRING) for proteins upregulated in the CD34$^{hi}$ vs CD34$^{lo}$ secretomes ($n = 3$). For STRING network, active interaction sources include experiments, databases, and co-expression; and minimum required interaction score was 0.700 (medium). FDR, false discovery rate. (J) Gene ontology pathway analysis of significantly enriched local network clusters (STRING) for proteins upregulated in CD34$^{lo}$ vs CD34$^{hi}$ fibroblast secretomes ($n = 3$). For STRING network active interaction sources include experiment, databases and co-expression; and minimum required interaction score was 0.700 (medium). FDR, false discovery rate.

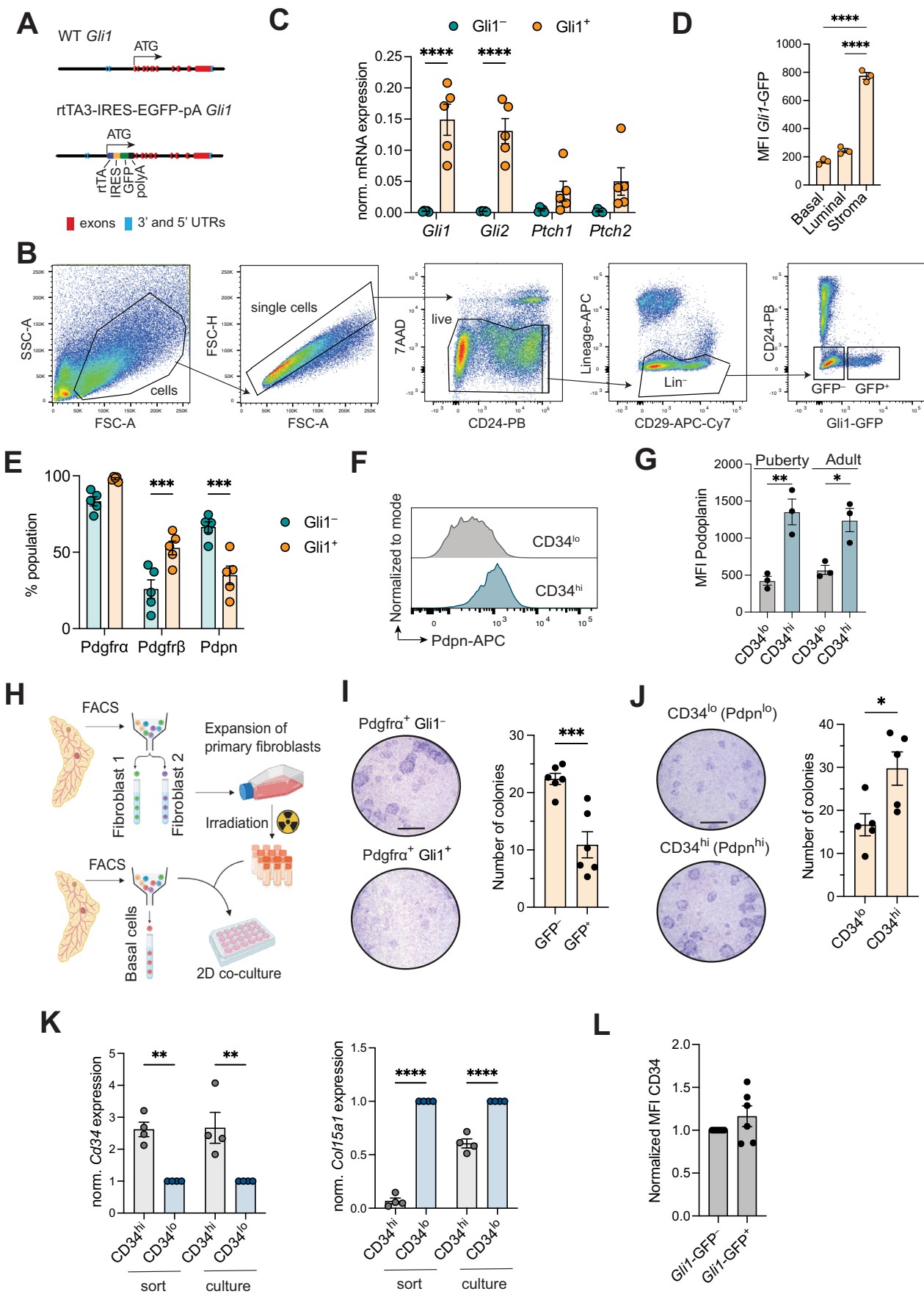

◄ **Figure EV4.   Characterization of Gli1⁺ specialized fibroblasts.**

(A) Targeting strategy to generate Gli1 rtTA3-IRES-EGFP-pA reporter mice. WT, wild-type. rtTA, reverse tetracycline-controlled trans-activator. IRES, internal ribosome entry sites. UTR, untranslated region. (B) Gating strategy for the Gli1-rtTA-IRES-GFP KI mouse model ($n = 8$). (C) mRNA expression of *Gli1*, *Gli2*, *Ptch1*, and *Ptch2* normalized to *Gapdh* expression in *Gli1*-GFP⁻ and *Gli1*-GFP⁺ mammary fibroblasts sorted from adult female mice. Each dot represents an individual mouse ($n = 5$). Error bars, mean ± s.e.m., ****$p < 0.0001$, two-way ANOVA. (D) Mean Fluorescence Intensity (MFI) of *Gli1*-GFP expression in basal (CD29^hi CD24^lo), luminal (CD29^lo CD24^hi) and stromal (CD24⁻) populations. Each dot represents an individual mouse ($n = 3$). Error bars, mean ± s.e.m., ****$p < 0.0001$, ordinary one-way ANOVA. (E) Percentage of Pdgfrα-, Pdgfrβ- or Pdpn (Podoplanin)-positive cells in *Gli1*-GFP^+/- stroma assessed by flow cytometry. Each dot represents an individual mouse ($n = 5$). Error bars, mean ± s.e.m., ***$p < 0.001$, two-way ANOVA. (F) Representative histograms for Pdpn expression in Pdgfrα⁺CD34^hi/lo fibroblasts from 6-week-old FVB/NJ mice by flow cytometry ($n = 3$). (G) Bar plot of MFI for Pdpn levels in Pdgfrα⁺CD34^hi/lo populations in pubertal (6-week old, $n = 3$) and adult (9-week old, $n = 3$) FVB/NJ mice. Error bars, mean ± s.e.m., *$p < 0.05$, **$P < 0.01$, unpaired *t*-test. MFI mean fluorescence intensity. (H) Workflow for 2D co-culture assays with primary irradiated fibroblasts and freshly sorted basal epithelial cells. Created with BioRender.com. (I) Representative images (left) and quantification (right) of basal colonies seeded with Pdgfrα⁺ *Gli1*-GFP⁺ or Pdgfrα⁺ *Gli1*-GFP⁻ cells ($n = 6$). Each dot represents the average of three replicates, each condition includes two independent sets of fibroblasts. Error bars, mean ± s.e.m., ***$p < 0.001$, unpaired *t*-test. Scale bar, 5 mm. (J) Representative images (left) and quantification (right) of basal/myoepithelial colonies seeded with Pdgfrα⁺CD34^hi/lo fibroblasts ($n = 5$). Each dot represents the average of three replicates, each condition includes two independent sets of fibroblasts. Error bars, mean ± s.e.m., *$p < 0.05$, unpaired *t*-test. Scale bar, 5 mm. (K) mRNA expression of *Cd34* (left) or *Col1a51a1* (right) genes normalized to *Gapdh* expression (housekeeping gene) in CD34^hi and CD34^lo mammary fibroblasts freshly sorted or after one passage in culture. Each expression value for CD34^hi cells was normalized to their CD34^lo counterpart. Each dot represents an individual mouse ($n = 4$). Error bars, mean ± s.e.m., **$p < 0.01$, ****$p < 0.0001$, ordinary one-way ANOVA. (L) Mean fluorescence intensity (MFI) of CD34 expression in *Gli1*-GFP^+/- CAFs from Gli1-Wnt1 tumors, normalized by the *Gli1*-GFP⁻ population for each tumor. Each dot represents a tumor ($n = 6$). Error bars, mean ± s.e.m. Source data are available online for this figure.

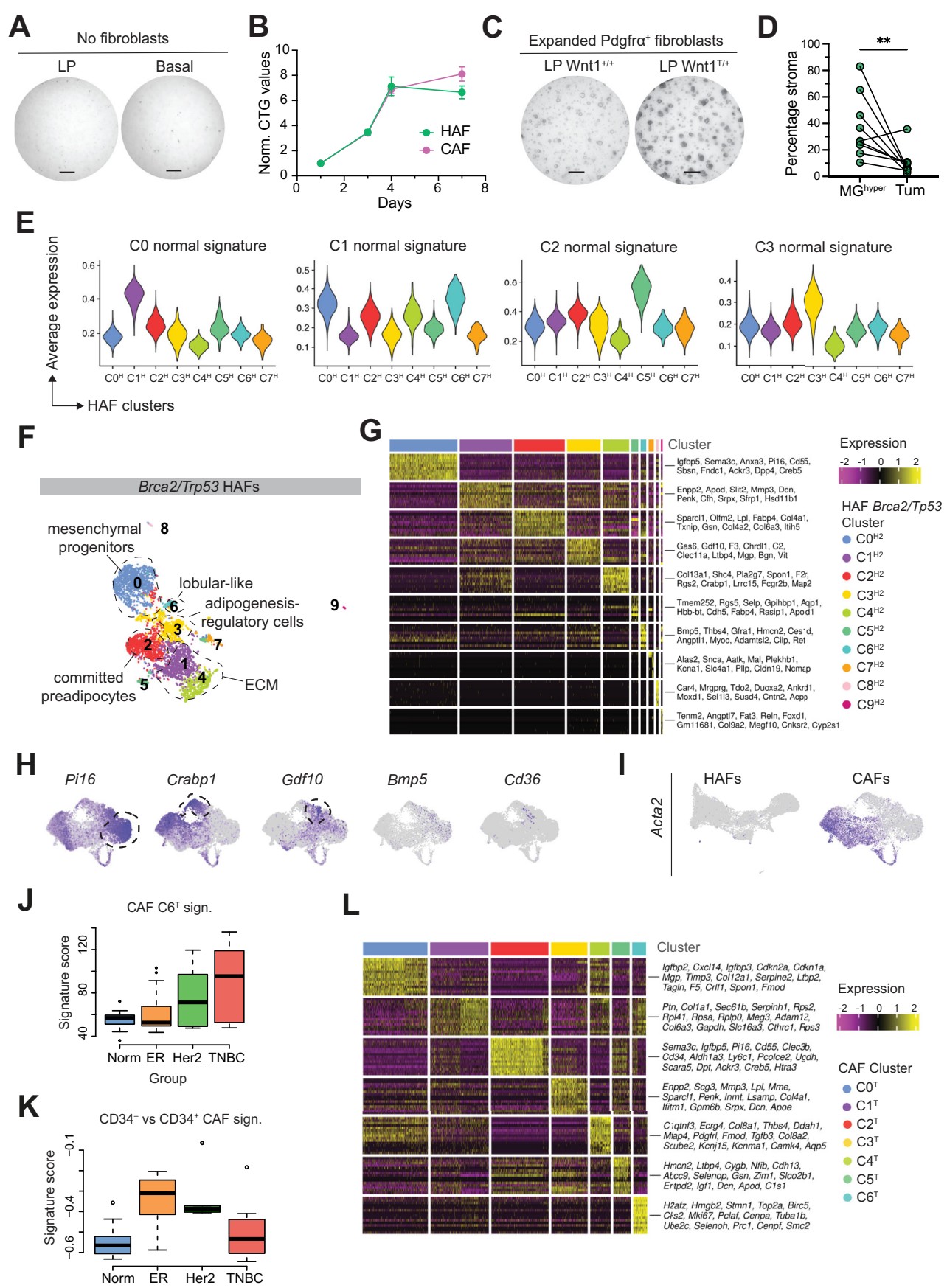

◄  **Figure EV5.   Characterization of hyperplasia- and cancer-associated fibroblasts.**

(A) Representative images of 3D colony-forming assays with basal and luminal progenitor (LP) cells without fibroblasts. Scale bars, 200 μm. (B) CellTiter-Glo (CTG) values, normalized to day 1, were used to assess the growth of HAFs and CAFs. $n = 3$ sets of matched raGFP-Wnt1 HAFs and CAFs. Error bars, mean ± s.e.m. (C) Representative images of 3D colony-forming assays with luminal progenitor (LP) cells from hyperplastic glands (16-week-old FVB-Wnt1$^{T/+}$) or littermate controls (FVB-Wnt1$^{+/+}$) co-cultured with Pdgfrα$^+$ fibroblasts. Scale bars, 200 μm. (D) Quantification of the percentage of stroma (Lin$^-$CD24$^-$) in raGFP-Wnt1 hyperplastic mammary glands and tumors. Each dot represents an individual mouse ($n = 9$). Error bars, mean ± s.e.m., **$p < 0.01$, unpaired $t$-test. (E) Violin plots for the enrichment of signatures from normal fibroblast clusters in each raGFP-Wnt1 HAF cluster (C0$^H$-C7$^H$). Upregulated genes in each normal cluster were used to generate cluster-specific signatures. HAF clusters are indicated by color. (F) UMAP plot of HAFs in *Brca2/Trp53*-deficient hyperplastic tissue showing ten clusters (C0$^{H2}$-C9$^{H2}$) indicated by color. Clusters were annotated according to their expression profiles. (G) Heatmap of gene expression for the top 10 marker genes for each HAF cluster in *Brca2/Trp53*-deficient glands (C0$^{H2}$-C9$^{H2}$). (H) UMAP plot (raGFP-Wnt1 CAFs) colored by expression of selected marker genes. (I) UMAP plots (raGFP-Wnt1 HAF and CAFs) colored by expression of *Acta2*. (J) Box plots showing enrichment of the mouse C6$^T$ CAF cluster transcriptional signature in normal human fibroblasts ($n = 13$) or CAFs in different breast cancer subtypes. ER, estrogen receptor ($n = 13$). HER2, human epidermal growth factor receptor 2 ($n = 6$). TNBC, triple-negative breast cancer ($n = 8$). Box plots show quartiles, minimum and maximum. (K) Box plots showing enrichment of the raGFP-Wnt1 CD34$^-$ CAF signature generated by bulk differential gene expression analysis ($n = 3$) in normal human fibroblasts or CAFs in different breast cancer subtypes. Box plots show quartiles, minimum and maximum. (L) Heatmap showing expression of the top 15 marker genes for each cluster in raGFP-Wnt1 CAFs (C0$^T$–C6$^T$).

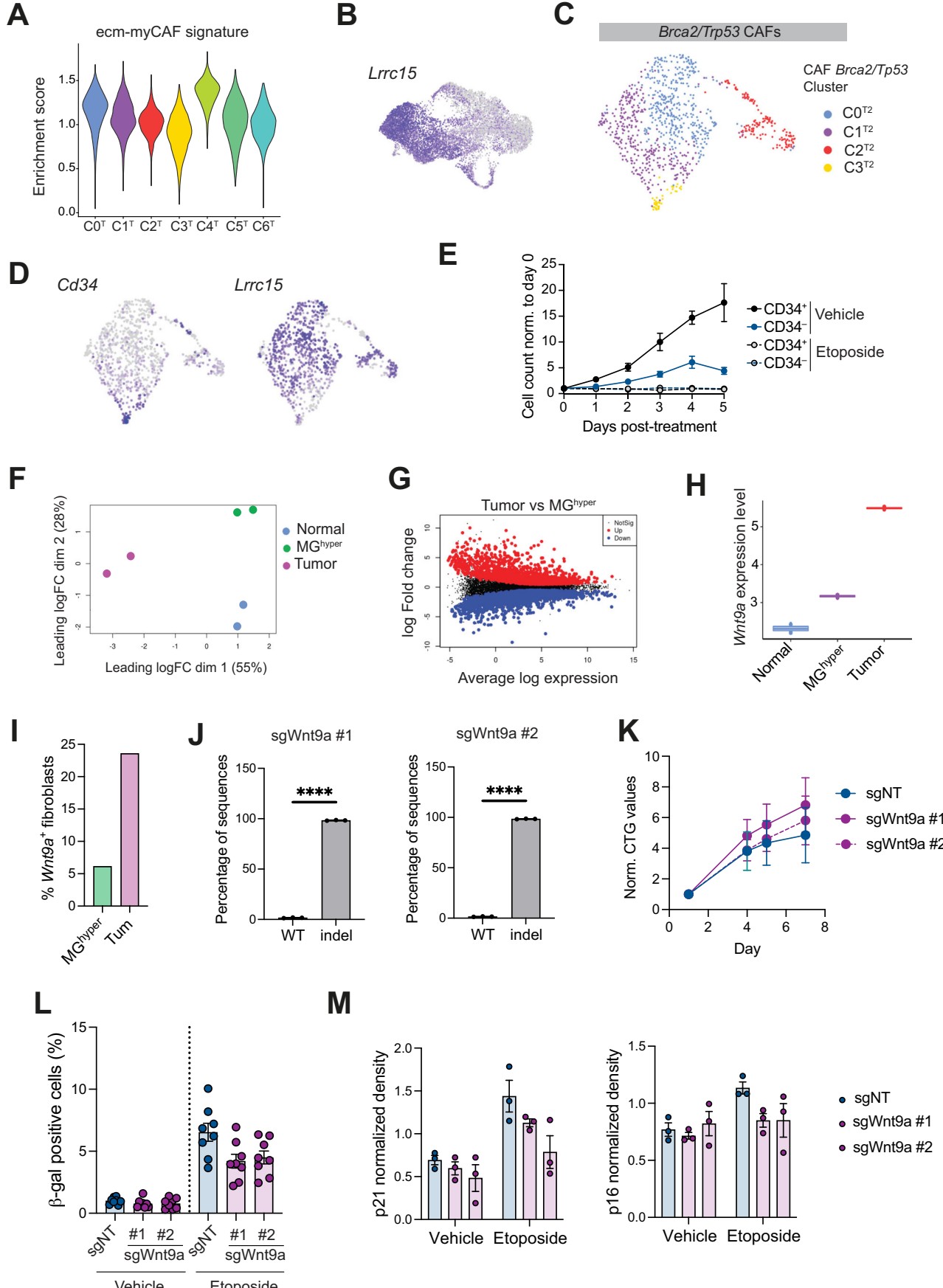

◀ **Figure EV6. Molecular features of CD34⁻ myCAFs.**

(A) Violin plots for enrichment of the ecm-myCAF signature (Kieffer et al, 2020) in each tumor cluster (C0$^T$–C6$^T$). (B) UMAP plot (raGFP-Wnt1 CAFs) colored by *Lrrc15* expression. (C) UMAP plot (CAFs, *Brca2/Trp53*-deficient model) showing four clusters (C0$^{T2}$–C3$^{T2}$) indicated by color. (D) UMAP plot (CAFs, *Brca2/Trp53*-deficient model) colored by expression of *Cd34* (left) and *Lrrc15* (right). (E) Cell counts for CD34$^+$ or CD34$^-$ raGFP-Wnt1 CAFs treated with vehicle or etoposide normalized to day 0 ($n = 4$, two independent sets of primary fibroblasts). Error bars, mean ± s.e.m. (F) Multidimensional scaling (MDS) plot for pseudo-bulk gene expression analysis of all fibroblasts in normal tissue (puberty + adult), hyperplastic tissue (MG$^{hyper}$, $n = 2$), and tumors ($n = 2$). (G) Mean-difference plot showing differentially expressed genes between fibroblasts in tumors vs hyperplastic glands MG$^{hyper}$ ($n = 2$). Significantly upregulated and downregulated genes are shown as red and blue dots, respectively. (H) *Wnt9a* expression in pseudo-bulk data for normal, hyperplastic (raGFP-Wnt1 MG$^{hyper}$, $n = 2$) and raGFP-Wnt1 tumor states ($n = 2$). (I) Percentage of *Wnt9a*$^+$ fibroblasts in hyperplastic tissue and tumors in *Brca2/Trp53*-deficient mice assessed by scRNA-seq ($n = 2$ mice per timepoint). (J) Percentage of WT (wild-type) or mutated (indels) sequences in CD34$^-$ myCAFs transduced with sgRNAs targeting the *Wnt9a* locus (sgWnt9a). Each dot represents a replicate ($n = 2$). ****$p < 0.0001$, unpaired *t*-test. Error bars, mean ± s.e.m. (K) CellTiter-Glo (CTG) values, normalized to day 1, to assess growth of control (sgNT) and Wnt9a-KO CD34$^-$ myCAFs. $n = 2$ independent sets of CRISPR-edited CD34$^-$ myCAFs performed in triplicate. Error bars, mean ± s.e.m. (L) Quantification of the percentage of β-galactosidase$^+$ CD34$^-$ myCAFs, either Wnt9a-KO or control (sgNT) cells treated with vehicle or etoposide. Values were normalized to the average of the untreated control for each experiment. Raw values for the etoposide-treated conditions are the same as in Fig. 6I. Each dot represents a technical duplicate, $n = 4$ independent experiments with two independent sets of primary CAFs. Error bars, mean ± s.e.m. (M) Quantification of p21 (left) and p16 (right) protein band density, normalized to Vinculin, evaluated by western blot analysis of control (sgNT) or Wnt9a-KO raGFP-Wnt1 CD34$^-$ myCAFs treated with vehicle or etoposide ($n = 3$). Error bars, mean ± s.e.m.

