## [Peer Review File · The EMBO Journal]

Fibroblast hierarchy dynamics during mammary gland morphogenesis and tumorigenesis

Rosa Pascual, Jinming Cheng, Amelia H. De Smet, Bianca D. Capaldo, Minhsuang Tsai, Somayeh Kordafshari, François Vaillant, Xiaoyu Song, Göknur Giner, Michael J.G. Milevskiy, Felicity C. Jackling, Bhupinder Pal, Toby Dite, Jumana Yousef, Laura F. Dagley, Gordon K. Smyth, Naiyang Fu, Geoffrey J. Lindeman, Yunshun Chen and Jane E. Visvader

Corresponding author: Jane Visvader (visvader@wehi.edu.au)

Review Timeline:

Submission Date:	16th Sep 24
Editorial Decision:	13th Nov 24
Revision Received:	11th Feb 25
Editorial Decision:	4th Mar 25
Revision Received:	11th Mar 25
Accepted:	14th Mar 25

Editor: Daniel Klimmeck

Transaction Report:

Dear Dr Visvader,

Thank you again for the submission of your manuscript (EMBOJ-2024-119042) to The EMBO Journal. Please accept my apologies for getting back to you with unusual delay due to protracted referee input and detailed discussion in the editorial team. As mentioned earlier, your study was assessed by three reviewers with expertise in breast cancer heterogeneity, fibroblast biology, and single-cell profiling analyses, whose comments are enclosed below.

As you will see from the experts' reports, the referees acknowledge the analyses combining longitudinal single-cell profiling on a relevant context with complementary *in vitro* and *in vivo* work and state the potential interest of your results. However, they also express major concerns, which need to be addressed thoroughly to make them supportive of publication in the EMBO Journal. In more detail, referee #2 points to overlap with previous single-cell results and a limited mechanistic insight into the fibroblast-dependent changes induced during mammary gland development and tumorigenesis (ref#1, standfirst). All reviewers raise issues with the level of conclusive support provided for a functional role of Wnt9a in regulating senescence, requesting complementary experiments to corroborate this claim (ref#1, pt.3; ref#2, pt.6; ref#3, pt.6-8). Further, referees #1 and #2 request integration of human patient data to put the current findings into context (ref#1, pt.5; ref#2, pt.7). Referee #3 in addition points to the need for you to revisit the annotation of clustering methods and scoring paradigms applied (ref#3, pts. 1-3). Finally, the reviewers point to concerns related to the data presentation, annotation of the newly introduced models, additional controls required and overall discussion of the findings and related literature, that would need to be conclusively addressed to achieve the level of robustness and clarity needed for The EMBO Journal.

Given the overall interest stated and broader angle of your findings, we are able to invite you to revise your manuscript experimentally to address the referees' comments. I need to stress though that we do require strong support from the referees on a revised version of the study in order to move on to publication of the work.

In light of the extensive experimentation requested, I would appreciate if you could contact me during the next weeks for exchange e.g. a video call to discuss your perspective on the comments and potential plan for revisions.

Please feel free to contact me if you have any questions or need further input on the referee comments.

When submitting your revised manuscript, please carefully review the instructions below.

Please feel free to approach me any time should you have additional questions related to this.

Thank you for the opportunity to consider your work for publication.

I look forward to your revision.

Best regards,

Daniel Klimmeck

Daniel Klimmeck, PhD
Senior Editor
The EMBO Journal

Instruction for the preparation of your revised manuscript:

2) individual production quality figure files as .eps, .tif, .jpg (one file per figure).

3) a .docx formatted letter INCLUDING the reviewers' reports and your detailed point-by-point response to their comments. As part of the EMBO Press transparent editorial process, the point-by-point response is part of the Review Process File (RPF), which will be published alongside your paper.

4) a complete author checklist, which you can download from our author guidelines ([https://wol-prod-cdn.literatumonline.com/pb-assets/embo-site/Author Checklist%20-%20EMBO%20J-1561436015657.xlsx](https://wol-prod-cdn.literatumonline.com/pb-assets/embo-site/Author%20Checklist%20-%20EMBO%20J-1561436015657.xlsx)). Please insert information in the checklist that is also reflected in the manuscript. The completed author checklist will also be part of the RPF.

6) It is mandatory to include a 'Data Availability' section after the Materials and Methods. Before submitting your revision, primary datasets produced in this study need to be deposited in an appropriate public database, and the accession numbers and database listed under 'Data Availability'. Please remember to provide a reviewer password if the datasets are not yet public (see <https://www.embopress.org/page/journal/14602075/authorguide#datadeposition>).

7) Our journal encourages inclusion of *data citations in the reference list* to directly cite datasets that were re-used and obtained from public databases. Data citations in the article text are distinct from normal bibliographical citations and should directly link to the database records from which the data can be accessed. In the main text, data citations are formatted as follows: "Data ref: Smith et al, 2001" or "Data ref: NCBI Sequence Read Archive PRJNA342805, 2017". In the Reference list, data citations must be labeled with "[DATASET]". A data reference must provide the database name, accession number/identifiers and a resolvable link to the landing page from which the data can be accessed at the end of the reference. Further instructions are available at .

8) At EMBO Press we ask authors to provide source data for the main and EV figures. Our source data coordinator will contact you to discuss which figure panels we would need source data for and will also provide you with helpful tips on how to upload and organize the files.

Numerical data can be provided as individual .xls or .csv files (including a tab describing the data). For 'blots' or microscopy, uncropped images should be submitted (using a zip archive or a single pdf per main figure if multiple images need to be supplied for one panel). Additional information on source data and instruction on how to label the files are available at .

9) We replaced Supplementary Information with Expanded View (EV) Figures and Tables that are collapsible/expandable online (see examples in <https://www.embopress.org/doi/10.15252/embo.201695874>). A maximum of 5 EV Figures can be typeset. EV Figures should be cited as 'Figure EV1, Figure EV2' etc. in the text and their respective legends should be included in the main text after the legends of regular figures.

11) For data quantification: please specify the name of the statistical test used to generate error bars and P values, the number (n) of independent experiments (specify technical or biological replicates) underlying each data point and the test used to calculate p-values in each figure legend. The figure legends should contain a basic description of n, P and the test applied. Graphs must include a description of the bars and the error bars (s.d., s.e.m.).

We realize that it is difficult to revise to a specific deadline. In the interest of protecting the conceptual advance provided by the work, we recommend a revision within 3 months (11th Feb 2025). Please discuss the revision progress ahead of this time with the editor if you require more time to complete the revisions.

Referee #1:

In this manuscript, Pascual et al. provide a comprehensive analysis of mammary-associated fibroblasts using single-cell RNA sequencing across various developmental stages and during oncogenesis in mice. The study reveals several unique fibroblast subtypes in normal mammary tissue, during puberty and pregnancy, which appear to originate from a CD34 high progenitor population. In late-stage cancer, the authors identify a variety of cancer-associated fibroblast populations, notably a Wnt9a-regulated CD34- subtype that they report as exhibiting traits associated with senescence or senCAFs. This work significantly advances our understanding of the diverse and dynamic fibroblast populations in the mammary gland.

While the study is well-executed and presents relevant data, additional functional assays could enhance the robustness of the findings.

Specific Suggestions:

1. Fig 1 . The authors should clarify whether littermate controls were used to ensure genetic and environmental consistency, especially for developmental comparisons. Given that female mice undergo estrous cycling, which may impact fibroblast populations in the puberty/adult group, were mice in these groups synchronized? Additionally, please specify if the pregnancy analyzed was a first pregnancy and discuss the potential impact of multiple pregnancies on these fibroblast clusters.
2. The manuscript lacks additional protein-level validation. Assessing cytokine production or protein expression related to the fibroblast secretome using high-throughput flow cytometry, proteomics-based cytokine arrays, or immunoblotting assays would enhance the study.
3. Fig 6. Additional functional assays to validate Wnt9a's role in regulating senescence would strengthen this claim. β -galactosidase is used as the sole readout, and the signal appears weak in Fig 6h. The authors should include western blots for senescence markers or assays using SASP-conditioned medium.
4. With recent but conflicting evidence on reversible senescence, the authors should specify the cell culture duration, and expand the discussion on senCAFs plasticity, addressing whether senCAFs remain in this state during oncogenesis or revert to an activated state.
5. The authors should explore whether specific fibroblast clusters or signatures uncovered throughout the study correlate with patient outcomes, such as overall survival, response to treatment, or specific breast cancer subtype. This would add clinical relevance to the manuscript.
6. Drawing clearer connections between fibroblast organization and clusters in development and of CAFs during oncogenesis, would improve the manuscript's flow and impact.

Referee #2:

In this manuscript, the authors perform single cell RNA sequencing (scRNA-seq) of fibroblasts derived from different stages of mammary gland development, hyperplasia, and cancer. Pseudotime trajectory analysis identified CD34 high mesenchymal cells with increased Hedgehog signaling as potential progenitors of fibroblasts. CD34- fibroblasts that have a transcriptional profile similar to senescent CAFs were identified in mammary cancer, with Wnt9a promoting CAF senescence. The mechanisms driving the changes in fibroblasts during mammary gland morphogenesis and cancer are largely not addressed; thus, the major impact this manuscript has is as a data resource. A similar dataset was previously reported (profiling of CD45- cells, PMID: 29474909), whereas this manuscript has better resolution of fibroblasts and includes puberty and hyperplasia states, somewhat limiting the overall impact of the findings. Several points should be addressed, as outlined below.

1. The transcriptional profiles of fibroblasts are evaluated in this manuscript through negative selection and the authors draw conclusions about abundance of specific fibroblasts subsets, but since the same number of fibroblasts are captured for scRNA-seq for each condition, it should be clarified within the manuscript that this is relative abundance as opposed to overall

- abundance. Similarly, does the overall number of fibroblasts change under different development stages and in tumorigenesis?
2. The Gli1-GFP model appears to be a newly developed model in this manuscript; however, characterization of the mouse model is not included. Additional validation that the insertion of the allele does not impact overall expression of Gli1 and the specificity of GFP labeling to Gli1-expressing cells should be included.
 3. Co-culture experiments are performed to determine the role of CD34+/CD34- and Gli1+/Gli1- fibroblasts in promoting the colony formation of basal cells. It is not clear what the rationale was for evaluating such functions of fibroblasts or how this is connected to the differentiation status of fibroblasts, as suggested by the authors.
 4. In addition, the authors should address whether co-culture with basal cells impacts fibroblast markers, e.g., whether CD34 and Gli1 expression are maintained in culture.
 5. CAFs compared to HAFs are shown to support the expansion of luminal progenitor cells. Differences in the proliferation rates of each fibroblast type could impact luminal cell progenitor proliferation and should be evaluated.
 6. Wnt9a is identified as a potential mediator of CAF senescence. The impact of Wnt9a loss on cell function beyond senescence should be included, such as alterations in the expression of fibroblast markers.
 7. The authors could analyze human fibroblast data from breast cancer, adjacent normal and pre-invasive lesions from public datasets to dissect similarities and differences between their mammary models and human data head-to-head. This analysis would further strengthen the relevance of this manuscript.
 8. In the discussion section, it is mentioned that since α SMA+ fibroblasts are only detected in tumors that myofibroblast conversion only occurs in late stages of tumor development; however, this conclusion should be tempered since other markers of myofibroblasts are not evaluated and lineage tracing is not performed. In addition, the authors draw conclusions about the function of Lrrc15+ myCAF s by comparing them to previously published studies which evaluate Lrrc15+ CAF s in pancreatic cancer. It is possible that CAF s have tissue-specific functions and extrapolations across cancer types should be tempered or the caveats discussed. Similarly, the authors assign pro-tumorigenic roles for senCAF s in Fig.6 and EV5. The authors' use of functional terms in the absence of experimental validation should be avoided.
 9. While lab mice undergo a polyestrous cycle, the differences in hormonal changes between mice and other primates with a menstrual cycle could impact breast development and oncogenesis differently. The authors could discuss such differences and their translational impact on human disease/physiology.
 10. Do the authors observe benign breast pathologies/ANDI in their mouse models?

Minor comments

1. In Fig. S1A, the genes defining the 'contaminating' cell populations should be included.
2. The authors mention the number of cells captured by scRNA-seq, but it would help to also clarify the number of mice evaluated per group within the main text (results section).

Referee #3:

Study by Pascal et al. examines heterogeneity of fibroblast populations in mammary glands across five developmental stages (puberty, adulthood, pregnancy, lactation and early involution) and during oncogenesis. Based on the clustering analysis of the single cell RNA-seq (scRNA-seq), the study characterises seven different clusters (population) of fibroblasts during postnatal development. Characterization of the populations is primarily focused on analysis of which genes and pathways are enriched in individual clusters. This analysis confirms findings of other studies in other organs (e.g. Buechler et al., 2021). More interesting result of the study is the role of Hedgehog signalling in fibroblast differentiation in normal mammary glands. Hedgehog signalling was earlier shown to regulate fibroblast activation and differentiation during development in other organs such as lung (Kugler et al, 2017). This study now shows that Hedgehog signalling is upregulated in fibroblasts during mammary gland development. Most interesting finding is the role of Wnt9a in regulation of the senescence in cancer-associated fibroblasts (CAF s). Although, the significance of Wnt9a on the main fibroblast function, i.e. regulation of extracellular matrix production, as well as Wnt9a role in tumour promotion and development is not investigated in the study. This study would be of interest mostly to the researchers in the breast cancer field. Methodologically, the study is mostly sound although I have some concerns.

Major comments

1. The main analysis in the study is based on the clustering of the scRNA-seq. It is unclear from the method section how the optimal cluster resolution (i.e. number of clusters) was assessed (resolution argument in the FindClusters function). Higher resolution will result in a higher number of smaller clusters. Optimal cluster resolution should be chosen in a way that maximises intercluster variability and minimises intracluster variability (e.g. by using silhouette score).
2. Authors state: "As C1 showed the most notable changes during ductal morphogenesis, we further interrogated heterogeneity within this cluster (Figs. 1i, EV1g)." It is unclear to me how it was determined that the C1 cluster showed the most notable changes during morphogenesis. Figures 1i and EV1g show the results of C1 subclustering.
3. According to the Method section: "Average expression of each signature gene list of interest was used to indicate the signature score." As I understand this, a signature score was calculated by averaging the expression of genes which were listed in signature score gene sets in each cell. If that is the case, this would mean that a small subset of genes which have high expression and are differentially expressed across clusters could result in substantial changes in the signature score. In addition, choice of gene count normalisation could influence the result. How does this signature score compare to scores calculated by method which use gene ranking, like AUCell (Aibar et al, 2017).
4. In the hormonal stimulation treatment section, authors state "The response to acute hormonal treatment may be directly mediated via progesterone and estrogen receptor expression on stromal cells (Fig. EV2h)." Could authors explain this in more detail? Figure EV2h shows expression of Esr1 and Pgr across clusters. Is this statement based exclusively on the expression of these receptors?
5. In the mammary tumour development section: "Within tumors, however, new Acta2+ myCAF populations lacking CD34 expression emerged, forming at least three subsets." In scRNA-seq data based on the overlap of the signature score of myCAF and CD34- CAF populations, it could be inferred that myCAF population (comprising of three clusters) does not express CD34. However, there is no UMAP plot showing directly CD34 expression in scRNA-seq data. To confirm that CD34 can distinguish between myCAF and iCAF population in scRNA-seq data, in the same manner as in bulk RNA-seq please provide the UMAP shown in figure 5l coloured by CD34 expression. Alternatively, Violin plot of CD34 expression across clusters.
6. Wnt9a regulation section: "This analysis revealed that C1T myCAFs were highly metabolically active (upregulation of Glycolysis and Ribosome)." This is an overstatement. It is very hard to interpret activity of metabolic pathways for RNA-seq data (Huang et al, 2023).
7. It would be interesting to determine the significance of Wnt9a on regulation of extracellular matrix production, and its role in tumour promotion and development. For example, does the expression of the components of ECM (e.g. Col1a1) or modifiers of ECM (e.g. Lox) differs between sgNT and sgWNT9a fibroblasts? Does the downregulation of Wnt9a in CAFs act as a tumour suppressor?
8. Is Wnt9a RNA and/or protein expression lower in sgWNT9a fibroblasts in comparison to the control?

Minor comments

1. Abbreviation CAF appears in the abstract for the first time, but it's only defined in the introduction.
2. In the heatmap shown in Figure 1d (and in other similar heatmaps) please define more explicitly what is plotted. Term "expression" is very vague. Are these log2FC of normalised counts between clusters or z scores of normalised counts?
3. Same for the x-axis in Figure 4c. "Statistic" is not very descriptive.
4. To be able to reproduce the scRNA-seq analysis, markers that were used to identify fibroblasts and other contaminating cells should be listed.
5. Buechler et al. publication is cited twice.

Additional suggestion

I would strongly encourage authors to make their code publicly available.

Citations

Aibar S, Bravo Gonzalez-Blas C, Moerman T, Huynh-Thu V, Imrichova H, Hulselmans G, Rambow F, Marine J, Geurts P, Aerts J, van den Oord J, Kalender Atak Z, Wouters J, Aerts S (2017). "SCENIC: Single-Cell Regulatory Network Inference And Clustering." *Nature Methods*, 14, 1083-1086.

Buechler, M.B., Pradhan, R.N., Krishnamurthy, A.T. et al. Cross-tissue organization of the fibroblast lineage. *Nature* 593, 575-579 (2021).

Huang, Y., Mohanty, V., Dede, M. et al. Characterizing cancer metabolism from bulk and single-cell RNA-seq data using METAFflux. *Nat Commun* 14, 4883 (2023).

Kugler, Matthias C., et al. "Sonic hedgehog signaling regulates myofibroblast function during alveolar septum formation in murine postnatal lung." *American Journal of Respiratory Cell and Molecular Biology* 57.3 (2017): 280-293.

We thank the Reviewers for their constructive comments, which we have addressed through further experiments and clarifications, as described below.

Referee #1:

In this manuscript, Pascual et al. provide a comprehensive analysis of mammary-associated fibroblasts using single-cell RNA sequencing across various developmental stages and during oncogenesis in mice. The study reveals several unique fibroblast subtypes in normal mammary tissue, during puberty and pregnancy, which appear to originate from a CD34^{hi} progenitor population. In late-stage cancer, the authors identify a variety of cancer-associated fibroblast populations, notably a Wnt9a-regulated CD34⁻ subtype that they report as exhibiting traits associated with senescence or senCAFs. This work significantly advances our understanding of the diverse and dynamic fibroblast populations in the mammary gland. While the study is well-executed and presents relevant data, additional functional assays could enhance the robustness of the findings.

1. Fig 1. The authors should clarify whether littermate controls were used to ensure genetic and environmental consistency, especially for developmental comparisons. Given that female mice undergo estrous cycling, which may impact fibroblast populations in the puberty/adult group, were mice in these groups synchronized? Additionally, please specify if the pregnancy analyzed was a first pregnancy and discuss the potential impact of multiple pregnancies on these fibroblast clusters.

All scRNA-seq experiments were performed using C57BL/6 mice from our Animal facility (originally from JAX). For these experiments, littermate controls could not be used for puberty and adult timepoints since the captures for different developmental stages were done on the same day to minimize variability in preparation of libraries. However, littermate controls were used where possible and pools of mice per timepoint, to minimize the effect of estrous cycling. The first pregnancy was always analyzed. Methods and legends have been modified accordingly to make these points clearer.

2. The manuscript lacks additional protein-level validation. Assessing cytokine production or protein expression related to the fibroblast secretome using high-throughput flow cytometry, proteomics-based cytokine arrays, or immunoblotting assays would enhance the study.

To address this point, we now include a mass spectrometry-based secretome analysis of cultured CD34^{hi} and CD34^{lo} fibroblasts (Fig. EV3h,i,j). While secreted factors from CD34^{hi} cells were related to proteolysis and endopeptidase activity, the CD34^{lo} secretome was enriched for genes involved with ECM organization, in accordance with their roles as Pi16⁺ mesenchymal progenitors and specialized fibroblasts, respectively.

The revised version of the manuscript also includes the following protein-level validation:

- CRABP1 immunohistochemistry and immunofluorescence. We validated the increase in CRABP1⁺ cells during acute hormonal stimulation (Fig. 3e,f) and hyperplasia preceding mammary tumor formation (Fig. 5h,i) and resolved their restricted location close to mammary epithelial ducts.
- Immunoblot for senescent markers in CRISPR-edited CD34⁻ myCAFs (Fig. 7e).
- CD34 protein expression was validated by FACS as a marker of different fibroblast populations in normal tissue (CD34^{hi} and CD34^{lo} as mesenchymal progenitors and specialized fibroblasts, respectively) as well as in tumors (CD34⁺ and CD34⁻ as

iCAFs and myCAFs, respectively); please see Figure 4. Characterization of Gli⁺ fibroblasts also included FACS profiling for Pdgfra, Pdgfr β and Podoplanin expression (Fig. EV4e).

3. Fig 6. Additional functional assays to validate Wnt9a's role in regulating senescence would strengthen this claim. β -galactosidase is used as the sole readout, and the signal appears weak in Fig 6h. The authors should include western blots for senescence markers or assays using SASP-conditioned medium.

To strengthen the role of Wnt9a as a regulator of senescence in CD34-negative myCAFs we have performed western blots for p21 and p16, markers of senescence. In accordance with the β -galactosidase results, Figure 7e shows that non-target control cells after etoposide treatment displayed higher levels of these senescent markers compared to Wnt9a-KO cells (Fig. EV6m).

We now include clearer contrast images for β -galactosidase (Fig. 7c). Please note that even after etoposide treatment there are only between 5-10% of β -galactosidase⁺ cells in the sgNT conditions and this percentage is even lower in Wnt9a-KO cells (Fig. EV6l).

4. With recent but conflicting evidence on reversible senescence, the authors should specify the cell culture duration, and expand the discussion on senCAFs plasticity, addressing whether senCAFs remain in this state during oncogenesis or revert to an activated state.

Cells were treated for 24 hours and allowed to recover for 4 days prior to the β -galactosidase assay (specified in Methods). For immunoblotting of senescent markers, cells were taken 24h after etoposide treatment. As these are short-term treatments, one would not expect to see reversibility of the senescent phenotype. It is plausible that senCAFs could revert to an activated state *in vivo* although this is an inefficient process (Ramponi et al, Cancer Res, 2025). It is likely that senescence-reverted cells would harbor epigenetic alterations compared to the pre-senescence state, but this is beyond the scope of this resource manuscript. We have added a sentence in the discussion related to this matter and further clarified details.

5. The authors should explore whether specific fibroblast clusters or signatures uncovered throughout the study correlate with patient outcomes, such as overall survival, response to treatment, or specific breast cancer subtype. This would add clinical relevance to the manuscript.

To address this question, we have analyzed fibroblast gene expression signatures from the single cell RNAseq data spanning 13 normal specimens and 27 breast tumors (ER+, HER2+ and TNBC) (Pal et al., EMBO J, 2021). The revised version of this manuscript includes interrogation of this dataset with the CAF transcriptional signatures uncovered in the present work. Similar to our mouse data, we found that myCAF clusters (C0^T, C1^T, C4^T) were enriched in human breast tumors compared to normal breast tissue, while iCAF clusters (C2^T, C3^T, C5^T) were not (Fig. 6e). No correlation between specific mouse CAF signatures and breast cancer subtype was found. However, a trend towards increased proliferative CAFs was seen in triple-negative breast cancer and enrichment of the CD34⁻ transcriptional signature was seen in ER+ tumors (Fig. EV5j,k).

Previous work had explored the prognostic value of human breast CAF signatures in response to immunotherapy in melanoma and lung cancer patients (Kieffer et al, 2020). We interrogated these signatures in our mouse CAF dataset and found that the ecm-myCAF signature, which is associated with an immunosuppressive environment and poor response to immune checkpoint blockade, was enriched in the C4^T myCAF cluster (Rebuttal Fig. 1, Fig. EV6a). Conversely, the detox-iCAF signature (that included *PII6* and *CXCL12* genes) was upregulated in the C2^T and C5^T iCAF clusters (Rebuttal Fig. 1). Detox-iCAF were predominant in triple-negative breast cancer and correlated with increased CD8⁺ T-cell infiltration (Kieffer et al, 2020). No clear enrichment of TGFβ-myCAF or IL-iCAF signatures was observed in any given CAF cluster.

Rebuttal Figure 1. Violin plots for enrichment of human breast CAF signatures (Kieffer et al, 2020) in each raGFP-Wnt1 tumor cluster (C0^T-C6^T).

6. Drawing clearer connections between fibroblast organization and clusters in development and of CAFs during oncogenesis, would improve the manuscript's flow and impact.

We thank the Reviewer for this suggestion. We have now clarified the connections between normal, hyperplastic and cancer states by adding new panels (eg. Figs. 11, 3e,f, 5h,i, 6e) and modifications to the text (P10, P11, P13).

In the revised version of this manuscript, we include new data that reveals an increased proportion of CRABP1⁺ cells upon acute hormonal upregulation and during hyperplasia (Figs. 3e,f, 5h,i), thus further characterizing the impact of hormones in the stromal compartment and changes in the stromal population before tumor development.

The interrogation of tumor fibroblast cluster signatures in human normal and cancerous fibroblasts (Fig. 6e) strengthens the findings of transcriptional similarity between normal fibroblasts and iCAFs, while myCAFs appear tumor-specific. In addition, we show that fibroblasts in normal tissue are CD34-positive (with CD34^{hi} being distinct from CD34^{lo} cells), while a new population of CD34-negative fibroblasts emerge in cancer. Accordingly, our bulk RNA-seq signature for CD34+ CAFs is enriched in normal human fibroblasts (Fig. EV5k).

To create a better flow from normal to cancer tissue, we now include characterization of the predicted fibroblast hierarchy based on CD34 expression in the normal mammary gland and its disruption during oncogenesis in the same figure (Figure 4).

Referee #2:

In this manuscript, the authors perform single cell RNA sequencing (scRNA-seq) of fibroblasts derived from different stages of mammary gland development, hyperplasia, and cancer. Pseudotime trajectory analysis identified CD34 high mesenchymal cells with increased Hedgehog signaling as potential progenitors of fibroblasts. CD34- fibroblasts that have a transcriptional profile similar to senescent CAFs were identified in mammary cancer, with Wnt9a promoting CAF senescence. The mechanisms driving the changes in fibroblasts during mammary gland morphogenesis and cancer are largely not addressed; thus, the major impact this manuscript has is as a data resource. A similar dataset was previously reported (profiling of CD45- cells, PMID: 29474909), whereas this manuscript has better resolution of fibroblasts and includes puberty and hyperplasia states, somewhat limiting the overall impact of the findings. Several points should be addressed, as outlined below.

1. The transcriptional profiles of fibroblasts are evaluated in this manuscript through negative selection and the authors draw conclusions about abundance of specific fibroblasts subsets, but since the same number of fibroblasts are captured for scRNA-seq for each condition, it should be clarified within the manuscript that this is relative abundance as opposed to overall abundance. Similarly, does the overall number of fibroblasts change under different development stages and in tumorigenesis?

For the scRNA-seq clusters, we include graphs showing the cell proportion as a percentage of each cluster, indicating their relative abundance in the total population of fibroblasts. To highlight this further we have added the term “relative” to the axis title of these graphs (Figs. 1c, 3d, 5d). As shown in Fig. EV1, the number of cells that were captured for scRNA-seq was slightly different for each condition.

The relative changes in fibroblast proportions at different stages of mammary gland development and tumorigenesis are indicated by the percentage of fibroblast-enriched stroma (CD24⁻) within the lineage negative fraction (CD31⁻CD45⁻Ter119⁻) (Rebuttal Fig. 2 and Fig. EV5d). In physiological conditions, the decreased percentage of fibroblasts within the Lin⁻ population during pregnancy is likely influenced by the expansion of basal and luminal cells. The lactation timepoint has been excluded since milk-containing luminal cells often

Rebuttal Figure 2. Percentage of stroma in the Lineage negative fraction evaluated by flow cytometry through several stages of post-natal development. $n=3$ for all developmental timepoints except for adult ($n=5$). Error bars, mean \pm s.e.m.

burst during cell preparation for flow cytometry analysis, therefore affecting quantification. Although not quantitative, immunofluorescence images using the *Pdgfra*^{H2B-GFP} model also show variable levels of GFP⁺ cells through post-natal development. Total GFP⁺ cells appeared to decrease in pregnancy and lactation, presumably due to the expansion of alveolar luminal cells (Rebuttal Fig. 3, n=2). However, within the fibroblast-enriched stroma, the vast majority of fibroblasts are Pdgfra⁺ at all developmental stages, as shown in Fig. 2b.

Rebuttal Figure 3. Representative confocal images showing puberty, pregnancy, lactation and involution timepoints in the *Pdgfra*^{H2B-GFP} mice stained with anti- α -SMA (white), anti-GFP (green) and DAPI (blue) (n=2). Scale bar, 100 μ m.

2. The Gli1-GFP model appears to be a newly developed model in this manuscript; however, characterization of the mouse model is not included. Additional validation that the insertion of the allele does not impact overall expression of Gli1 and the specificity of GFP labeling to Gli1-expressing cells should be included.

The correlation between GFP and Gli1 expression is now included in the revised manuscript (Fig. EV4c) by analysis of RNA expression of *Gli1* and Gli1 target genes (*Gli2*, *Ptch1* and *Ptch2*) by qPCR analysis of sorted GFP⁺ and GFP⁻ mammary fibroblasts. Please note that endogenous Gli1 gene expression is destroyed in the Gli-GFP reporter mouse and

heterozygous $KI/+$ mice are used through this study. This has now been made clear in the Methods section. It is known that *Gli1*-deficient mice appear normal in many tissues, where it has not been possible to identify a distinct non-redundant function for *Gli1* (Park et al., 2000; Bai et al., 2002).

3. Co-culture experiments are performed to determine the role of $CD34^+/CD34^-$ and $Gli1^+/Gli1^-$ fibroblasts in promoting the colony formation of basal cells. It is not clear what the rationale was for evaluating such functions of fibroblasts or how this is connected to the differentiation status of fibroblasts, as suggested by the authors.

Basal and luminal progenitor cells can form colonies in 2D cultures when co-cultured with irradiated NIH3T3 cells. Differing from NIH3T3 cells, primary normal fibroblasts (irradiated) can support basal cell colonies but not the growth from luminal progenitor cells (Rebuttal Fig. 4).

Rebuttal Figure 4. Representative images of basal and Luminal Progenitor (LP) colonies seeded with irradiated 3T3 or primary fibroblasts ($n=5$). Scale bar, 5 mm.

We then investigated whether distinct sub-populations of fibroblasts differentially promote the growth of basal cells in this assay, which could indicate whether, even when fibroblasts change upon culture (Baranyi et al, Cells, 2019; Antero et al, Ageing Res. Rev., 2020), some intrinsic characteristics of subpopulations can be retained. In these 2D colony assays, in line with our transcriptomic and *in vivo* data, we found interesting and opposing data for $CD34^{hi}$ vs $CD34^{lo}$ and $Gli1^+$ vs $Gli1^-$ cells, suggesting that primary fibroblast subpopulations indeed retain some characteristics in culture (see point below). We agree with the Reviewer that it is not clear whether there is crosstalk *in vivo* between fibroblasts and epithelial cells, and whether this is connected to the differentiation status of fibroblasts. Therefore, we have moved these data to Extended view data and modified the text accordingly, since this is not a main finding of our work.

In an attempt to further tackle this question, we used the 2D assay to evaluate the growth of normal basal or luminal progenitor (LP) cells together with primary fibroblasts from *Brca2/Tp53*-deficient mammary glands (hyperplasia-associated fibroblasts) or tumors (cancer-associated fibroblasts). We found that CAFs, as opposed to normal fibroblasts or HAFs, supported the growth of LP cells and further promoted basal cell growth compared to co-culture with HAFs (Rebuttal Fig. 5). Since CAFs include a differentiated population, this suggests that enhanced epithelial cell growth in these assays may be independent of fibroblast differentiation status or that cancer-educated fibroblasts have acquired new features *in vivo* that eliminates that dependency. We have included the data for the Reviewers (Rebuttal Fig. 5).

Rebuttal Figure 5. Representative brightfield images of colonies from normal FVB/N basal or luminal progenitor (LP) cells seeded with irradiated hyperplasia-associated fibroblasts (HAFs) or cancer-associated fibroblasts (CAFs) from Brca2/Tp53 mice (n=4). Error bars, mean \pm s.e.m

4. In addition, the authors should address whether co-culture with basal cells impacts fibroblast markers, e.g., whether CD34 and Gli1 expression are maintained in culture.

We agree that the culture of fibroblasts and expansion *in vitro* will impact the expression of some fibroblast genes. In accordance with previous literature (Baranyi et al, Cells, 2019; Antero et al, Ageing Res. Rev., 2020), we found that fibroblasts acquire a more myofibroblast-like phenotype upon culture, as shown by the downregulation of *Cd34* and upregulation of *Acta2* expression (Rebuttal Fig. 6). However, as shown in Fig. EV4k, CD34^{hi} and CD34^{lo} fibroblasts in culture can retain *in vivo* features such as lower *Cd34* and higher *Coll5a1* expression in specialized fibroblasts compared to mesenchymal progenitor cells.

Rebuttal Figure 6. mRNA expression of *Acta2* (left, $n=3$) or *Cd34* (right, $n=4$) genes normalized by *Gapdh* expression in $CD34^{hi}$ and $CD34^{lo}$ mammary fibroblasts freshly sorted or in culture. Error bars, mean \pm s.e.m.

5. CAFs compared to HAFs are shown to support the expansion of luminal progenitor cells. Differences in the proliferation rates of each fibroblast type could impact luminal cell progenitor proliferation and should be evaluated.

We thank the Reviewer for raising this point. Although proliferative fibroblasts are more common in tumors than hyperplasia *in vivo*, we did not observe any statistical difference in the proliferation of CAFs vs HAFs *in vitro*. Cell proliferation was measured by the CellTiter-Glo assay, and these results are included in Fig. EV5b.

6. Wnt9a is identified as a potential mediator of CAF senescence. The impact of Wnt9a loss on cell function beyond senescence should be included, such as alterations in the expression of fibroblast markers.

Fibroblast markers can dramatically change *in vitro*, please see point 4. We therefore checked *Cxcl12* (iCAF) and *Acta2* (myCAF) expression as well as proliferation rates in Wnt9a-KO fibroblasts after transduction/passaging and did not observe any differences (Rebuttal Fig. 7, EV6k). To more precisely address this question in the future, a conditional Wnt9a KO model could be used, ideally using an *Acta2*-driven inducible Cre in a breast cancer mouse model (since Wnt9a is mainly expressed in myCAFs).

Rebuttal Figure 7. mRNA expression of Acta2 (left) or Cxcl12 (right) genes normalized by Gapdh expression and to the sgNon-target control values. n=6, error bars, mean \pm s.e.m

7. The authors could analyze human fibroblast data from breast cancer, adjacent normal and pre-invasive lesions from public datasets to dissect similarities and differences between their mammary models and human data head-to head. This analysis would further strengthen the relevance of this manuscript.

To address this question, we have analyzed fibroblast gene expression signatures from the single cell RNAseq data spanning 13 normal specimens and 27 breast tumors (ER+, HER2+ and TNBC) (Pal et al., EMBO J, 2021). The revised version of this manuscript includes interrogation of this dataset with the CAF transcriptional signatures uncovered in the present work. Similar to our mouse data, we found that myCAF clusters (C0^T, C1^T, C4^T) were enriched in human breast tumors compared to normal breast tissue, while iCAF clusters (C2^T, C3^T, C5^T) were not (Fig. 6e). No correlation between specific mouse CAF signatures and breast cancer subtype was found. However, a trend towards increased proliferative CAFs was seen in triple-negative breast cancer and enrichment of the CD34⁻ transcriptional signature was seen in ER+ tumors (Fig. EV5j,k).

8. In the discussion section, it is mentioned that since aSMA+ fibroblasts are only detected in tumors that myofibroblast conversion only occurs in late stages of tumor development; however, this conclusion should be tempered since other markers of myofibroblasts are not evaluated and lineage tracing is not performed. In addition, the authors draw conclusions about the function of Lrrc15+ myCAFs by comparing them to previously published studies which evaluate Lrrc15+ CAFs in pancreatic cancer. It is possible that CAFs have tissue-specific functions and extrapolations across cancer types should be tempered or the caveats discussed. Similarly, the authors assign pro-tumorigenic roles for senCAFs in Fig.6 and EV5. The authors' use of functional terms in the absence of experimental validation should be avoided.

We have toned down the text to address this point (P14). Although Lrrc15 upregulation in CAFs has been shown in multiple cancers including breast (Dominguez et al., 2020), their immunosuppressive function has not been demonstrated in breast cancer. However, senescent CAFs with an immunosuppressive function have been shown to express high levels of Lrrc15 (Ye at al., Cancer Discovery, 2024). We have modified the text in the discussion to address this comment.

9. While lab mice undergo a polyestrous cycle, the differences in hormonal changes between mice and other primates with a menstrual cycle could impact breast development and oncogenesis differently. The authors could discuss such differences and their translational impact on human disease/physiology.

We have now added a sentence in the Discussion regarding the possible effect of the estrous cycle on fibroblast populations: “Fibroblasts might also respond to endogenous hormonal fluctuations during the estrous cycle and breast epithelial-associated fibroblasts have been observed to change with menopausal status (Pal et al EMBO J 2021).”

10. Do the authors observe benign breast pathologies/ANDI in their mouse models?

No benign breast pathologies were detected in these mouse models.

Minor comments

1. In Fig. S1A, the genes defining the 'contaminating' cell populations should be included. These are now included in Supplementary Table 1.

2. The authors mention the number of cells captured by scRNA-seq, but it would help to also clarify the number of mice evaluated per group within the main text (results section).

Methods now include clearer information on the number of mice evaluated per group for scRNA-seq: “Freshly sorted stromal fractions (Lineage⁻CD24⁻) were obtained from five post-natal developmental stages (2 C57BL/6 female mice per condition), control or hormone-stimulated (placebo control or MPA+E, n=2, pool of two C57BL/6 females per replicate) as well as freshly sorted GFP⁺ cells from raGFP-Wnt1 mammary tumors and paired hyperplastic mammary glands from the same mouse (n=2)”.

Referee #3:

1. The main analysis in the study is based on the clustering of the scRNA-seq. It is unclear from the method section how the optimal cluster resolution (i.e. number of clusters) was assessed (resolution argument in the FindClusters function). Higher resolution will result in a higher number of smaller clusters. Optimal cluster resolution should be chosen in a way that maximises intercluster variability and minimises intracluster variability (e.g by using silhouette score).

We thank the Reviewer for this suggestion. Indeed, metrics like silhouette score can help determine the cluster resolution that is technically the ‘optimal’, but it is crucial to also consider the biological context. Some people will be satisfied with resolution of the major cell types (which might be technically the optimal resolution), whereas others may want resolution of subtypes or different states within those subtypes. In this exploratory study, our goal was to identify heterogeneity amongst fibroblast subsets. Achieving this might not be possible by simply maximising the intercluster and minimising the intracluster variability. Instead, we took an iterative approach, which is suitable for the nature of our study. Initially, we chose clustering resolutions within the range of 0.2 to 0.3 and then evaluated the

outcomes through UMAP visualization and marker gene analysis for each cell cluster. By incorporating prior knowledge from the literature and our understanding of the biology, we fine-tuned the clustering resolution to ensure the final results were biologically meaningful.

2. Authors state: "As C1 showed the most notable changes during ductal morphogenesis, we further interrogated heterogeneity within this cluster (Figs. 1i, EV1g)." It is unclear to me how it was determined that the C1 cluster showed the most notable changes during morphogenesis. Figures 1i and EV1g show the results of C1 subclustering.

Notable changes in C1 were evident in Fig. 1b,c while heterogeneity within the cluster was investigated in Figs. 1j-l, EV1f. We apologize for the confusion and have added further references to figures in the text to make clearer. To further strengthen this point, we performed an independent capture of puberty and adult fibroblasts, which showed the emergence of new subpopulations within the C1 cluster in the adult (Fig. EV1j-m).

3. According to the Method section: "Average expression of each signature gene list of interest was used to indicate the signature score." As I understand this, a signature score was calculated by averaging the expression of genes which were listed in signature score gene sets in each cell. If that is the case, this would mean that a small subset of genes which have high expression and are differentially expressed across clusters could result in substantial changes in the signature score. In addition, choice of gene count normalisation could influence the result. How does this signature score compare to scores calculated by method which use gene ranking, like AUCell (Aibar et al, 2017).

Indeed, the signature scores were calculated by averaging the expression levels of a set of genes. As we are comparing these scores across different cell clusters, the absolute expression levels of the signature genes do not matter. They effectively cancel out in comparative analyses. What is important is the fold change, which can be considered as a weight in the calculation of the signature scores. Assigning higher weights to genes that show significant differences between cell clusters is a logical approach. This methodology has been successfully applied in several studies (Pal et al., Nat. Commun., 2017; Pal et al., EMBO J, 2021; Joyce & Pascual, Nat. Cell Biol. 2024). We did not explore alternative methods that use gene ranking, such as AUCell, as we did not find it necessary.

4. In the hormonal stimulation treatment section, authors state "The response to acute hormonal treatment may be directly mediated via progesterone and estrogen receptor expression on stromal cells (Fig. EV2h)." Could authors explain this in more detail? Figure EV2h shows expression of *Esr1* and *Pgr* across clusters. Is this statement based exclusively on the expression of these receptors?

We have now clarified this point in the text and added a new panel in Fig. EV3e showing colocalization of Progesterone receptor and *Pdgfra*-GFP in the stroma based on 3D confocal imaging.

5. In the mammary tumour development section: "Within tumors, however, new Acta2+ myCAF populations lacking CD34 expression emerged, forming at least three subsets." In scRNA-seq data based on the overlap of the signature score of myCAF and CD34- CAF populations, it could be inferred that myCAF population (comprising of three clusters) does not express CD34. However, there is no UMAP plot showing directly CD34 expression in scRNA-seq data. To confirm that CD34 can distinguish between myCAF and iCAF

population in scRNA-seq data, in the same manner as in bulk RNA-seq please provide the UMAP shown in figure 5l coloured by CD34 expression. Alternatively, Violin plot of CD34 expression across clusters.

We thank the Reviewer for the suggestion and now include a UMAP colored by CD34 expression in Fig. 6c.

6. Wnt9a regulation section: "This analysis revealed that C1T myCAFs were highly metabolically active (upregulation of Glycolysis and Ribosome)." This is an overstatement. It is very hard to interpret activity of metabolic pathways for RNA-seq data (Huang et al, 2023).

We have modified the text to correct this.

7. It would be interesting to determine the significance of Wnt9a on regulation of extracellular matrix production, and its role in tumour promotion and development. For example, does the expression of the components of ECM (e.g. *Colla1*) or modifiers of ECM (e.g. *Lox*) differs between sgNT and sgWNT9a fibroblasts? Does the downregulation of Wnt9a in CAFs act as a tumour suppressor?

Wnt9a as a target for reprogramming CAFs into a more tumor suppressive state is an interesting question that requires further investigation that goes beyond the scope of this resource paper. The culture of fibroblasts *in vitro* can greatly impact the expression of fibroblast markers (Baranyi et al, *Cells*, 2019; Antero et al, *Ageing Res. Rev.*, 2020), therefore characterization of the expression of components or modifiers of the ECM might be misleading after CRISPR-Cas9 editing *in vitro*. To more precisely address this question, a conditional Wnt9a knockout model should be used, ideally using an *Acta2*-driven inducible Cre in a mammary tumor mouse model (since Wnt9a is mainly expressed in myCAFs).

We checked proliferation rates as well as the expression of several fibroblast markers including *Acta2* (myCAF), *Cxcl12* (iCAF), *Colla2* (ECM component), *Lox* (ECM modifier) and *Cdh11* (inflammatory) and did not observe any differences (Rebuttal Figure 8, EV6k).

Rebuttal Figure 8. mRNA expression of selected genes normalized by *Gapdh* expression and to the sgNon-target (sgNT) control values. $n=6$, error bars, mean \pm s.e.m.

8. Is Wnt9a RNA and/or protein expression lower in sgWNT9a fibroblasts in comparison to the control?

Unfortunately, there are no reliable Wnt9a antibodies that could be used to detect Wnt9a protein. We attempted to assess differences in *Wnt9a* RNA expression by qPCR, but *Wnt9a* is so lowly expressed that a reduction in *Wnt9a* upon KO did not reach statistical significance for both guides (Rebuttal Figure 9). However, given the clear results by DNA sequencing we are confident that the Wnt9a locus is targeted efficiently by these guides.

Rebuttal Figure 9. mRNA expression of *Wnt9a* normalized by *Gapdh* expression. $n=8$, error bars, mean *s.e.m.*

Minor comments

1. Abbreviation CAF appears in the abstract for the first time, but it's only defined in the introduction.

We have now added the abbreviation CAFs after cancer-associated fibroblasts in the abstract.

2. In the heatmap shown in Figure 1d (and in other similar heatmaps) please define more explicitly what is plotted. Term "expression" is very vague. Are these log₂FC of normalised counts between clusters or z scores of normalised counts?

"Expression" is the row-scaled z-scores of log-normalized counts, now added to the Methods section.

3. Same for the x- axis in Figure 4c. "Statistic" is not very descriptive.

"Statistics" are the log₂-FC of all the genes between "Cluster 0" and "Other" clusters. We have modified the panel to reflect the correct term.

4. To be able to reproduce the scRNA-seq analysis, markers that were used to identify fibroblasts and other contaminating cells should be listed.

This information is now included in Supplementary Table 1

5. Buechler et al. publication is cited twice.

We have amended this, thank you.

Dear Dr Visvader,

Thank you for submitting your revised manuscript (EMBOJ-2024-119042R) to The EMBO Journal, as well for your patience with our response. Your amended study was sent back to the three referees for their scientific re-evaluation, and we have received detailed comments from all of them, which I enclose below. As you will see, the experts state that the work has been substantially enhanced by the revisions and they are now broadly in favour of publication, pending minor revision.

Thus, we are pleased to inform you that your manuscript has been accepted in principle for publication in The EMBO Journal.

Please carefully consider the remaining minor points raised by the reviewers regarding comprehensive discussion of the findings, data and experimental annotation, additional literature references and bioinformatic analysis by revisiting the analysis or adjusting the statements and wording of the manuscript where appropriate.

We also now need you to take care of a number of issues related to formatting and data presentation as detailed below, which should be addressed at re-submission.

Please contact me at any time if you have additional questions related to below points.

As you might remember from previous experience, every paper at the EMBO Journal now includes a 'Synopsis', displayed on the html and freely accessible to all readers. Besides a 'model' figure the synopsis includes 2-5 one-short-sentence bullet points that summarize the article. I would appreciate if you could provide these bullet points.

Thank you for giving us the chance to consider your manuscript for The EMBO Journal. I look forward to your final revision.

Again, please contact me at any time if you need any help or have further questions.

Best regards,

Daniel Klimmeck

>> Please add up to five keywords to your study.

>> Authors: please revisit name discrepancy in the manuscript and our online system for co-author S.K. .

>> Author Contributions: Remove the author contributions information from the manuscript text. Note that CRediT has replaced the traditional author contributions section as of now because it offers a systematic machine-readable author contributions format that allows for more effective research assessment. and use the free text boxes beneath each contributing author's name to add specific details on the author's contribution.

More information is available in our guide to authors.
<https://www.embopress.org/page/journal/14602075/authorguide>

>> Provide a completed Author Checklist.

>> Adjust the title of the 'Declaration of Interest' section to 'Disclosure and Competing Interests Statement'.

>> Correct order of manuscript sections: Abstract / Keywords / Introduction / Results / Discussion / Methods / Data Availability / Acknowledgements / Disclosure and competing interests statement // References / Figure legends / Tables and their legends / Expanded View Figure legends

>> Figures in separate files: main and EV figures should be uploaded as individual, high-resolution figure files.

>> Figure callouts: Please ensure that the figure panel Fig 7F is called out in sequential order.

>> References: Please recheck the ones with DOI as DOIs should only be used for preprints and datasets that have not been published yet.

>> Funding: please enter the following funding information in the list of funders in our online system: 'IIRS-20-022 (NBCF); the Victorian State Government Operational Infrastructure Support; MRFF Investigator Grant (#1176199GJL); NHMRC Fellowship 1154970; Investigator Grant 2025645; NHMRC Fellowships (GJL #1078730, 1175960; JEV #1037230, 1102742)'. Note that the Comments box should not be used but rather the 'More Funders' option.

>> Dataset EV legends: From the currently seven Tables uploaded as one Excel file (each table one sheet); only Table 1 and Table 7 can stay with those labels but each should be uploaded as a separate Table file (Table 1 and Table 2); Tables 2-6 should be uploaded as a separate Dataset file (Dataset EV1-EV5); please update the nomenclature for both tables and datasets in: source file names, legends in the files, titles in our online system and manuscript callouts ("Supplementary" should not be used).

>> Add a Reagents and Tools table to the Methods section, as a separate file using the existing template in the Guide For Authors, listing key reagents, experimental models, software and relevant equipment.

>> Data availability section: change the header of the section from 'Data and code availability' to 'Data availability section'. Remove the referee tokens and make sure GEO and PRIDE datasets are made publicly accessible.

>> Source data: source data should be uploaded as one (zipped) file per figure - inside each folder there should be separate files/folders, one file/folder per panel.

>> Add a Reagents and Tools table to the Methods section, as a separate file using the existing template in the Guide For Authors, listing key reagents, experimental models, software and relevant equipment.

>> The nomenclature of the EV figure legends in the manuscript needs correction: it should be Figure EV1, etc. instead of Expanded View Fig. 1. etc. .

>> Consider additional changes and comments from our production team as indicated below:

- Please note that the box plots need to be defined in terms of minima, maxima, centre, bounds of box and whiskers, and percentile in the legends of figures 6E, EV1 I; EV5 J, K
- Please note that information related to n is missing in the legends of figures 1E, 3I, 6E, G; 7B; EV1 C, I, M; EV3 A, F, H; EV5 E, J; EV6 A, J
- Please note that n=2 in figures 2D, EV6 H, K
- Please note that the error bars are not defined in the legends of figures 2D, EV6 J, M"

*Please note that the specific URLs for GSE277228, PXD060415 datasets are not provided in the data availability statement.

Figure Legends - Comments

- Please indicate what */ **/ ***/ **** represents; if this represents p value(s), please indicate the statistical test used and where appropriate, and the exact p value in the legend(s) of figure(s) 6D
- Please note that the exact p values are not provided in the legends of figures 1E, 2D, 3F, I; 4I, K, M, O; 5B, H, I; 6G, K; 7D, EV1 I; EV3 F, EV4 C, D, E, G, I, J, K; EV5 D, EV6 J.
- Please indicate the statistical test used for data analysis in the legends of figures EV1 D, EV3 G, H

Referee #1:

The authors have addressed most of my concerns. A few points to consider:

1. As mentioned previously, speculating on the potential effects of multiple pregnancies on fibroblasts/CAF clusters would be a valuable addition to the discussion.
2. Page 7: Are the observed gene expression changes upon hormonal treatment due to ER and PR transcriptionally regulating Crabp1, Fabp4, Gdf10, and Bmp5?

Referee #2:

Ref.1:

1. Comment 1: The changes the authors have been made are well done. What would be even more precise is if the authors could define the actual timespan of the each period in days. What was the timespan in days the authors were using mice in puberty? What in Adulthood etc etc.. this would help authors researchers to potentially reproduce the results.

2. Comment 2: The additional performed analysis is good.

3. Comment 3: The full blots should be provided in the supplementary material!
"We now include clearer contrast images for β -galactosidase (Fig. 7c)."
This has to be marked in the actual figure because it is not well visible for the reader.

4. Comment 4: The arguments listed of the authors that it is beyond the purpose of this manuscript are ok. But further literature supporting "the one sentence in the discussion" should be implemented.

5. Comment 5: The additional analysis is good. Can the authors clarify in the investigated cohort what the actual pathology of the human tissues looks like? Is all breast cancers included in the cohort or do the clusters found vary in different subtypes? The issue is that there is a huge literature existing for different markers and it needs to be clearly specified in a common nomenclature for further investigations.

Ref 2:

6. Comment 1: Did the authors perform statistical analysis on the graphs shown?

7. Comment 2: Is there more recent literature than 2002?

8. Comment 5: The authors cannot see more proliferation of CAFs vs HAFs in vitro. Did the authors perform stainings for the named proliferation of their mouse models?

9. Comment 6: Also here the arguments of the authors towards a future knock out model are valid. But before that Wnt9 stainings on their mouse model in different age stages could be performed.

10. Comment 7: Again please define the human samples towards the cancer pathology more precise.

Ref 3:

11. Comment 7: Here again a staining for Wnt might be additional helpful as in vivo confirmation is tissues of these mice were kept.

Additionally to these in vitro experiments.

12. Comment 8: Is there is really no direct Wnt9 antibody available another close antibody of the similar family/ pathway could be used.

Recommend accept and I suggest an accompanying commentary to align this work with what is known the field

Referee #3:

While the authors have addressed some of my concerns, certain issues, in my opinion, still need to be resolved before publication:

1. The issue of clustering resolution (original point 1) has not been adequately addressed. The proposed iterative approach presents several issues. First, the resolution range between 0.2 and 0.3 is already quite narrow. Additionally, interpreting clustering results using UMAP visualizations requires caution, as UMAP is highly sensitive to hyperparameter choices, and the distances between clusters may not be meaningful. As the authors themselves demonstrate with their subclustering approach, determining the final number of clusters is inherently difficult, as clusters can be further subdivided. Relying on the technically most optimal solution helps maintain an unbiased approach, whereas biological interpretation remains largely subjective. In my opinion, the authors have not convincingly shown that their chosen resolution is superior or more justified than the technically optimal alternative.

2. From figures 1b and c I can't assess that cluster C1 has the most notable changes. Majority of the clusters show changes as these are relative proportions. In addition, changes in relative proportions do not necessarily have to lead to changes in population heterogeneity. While I don't have an issue with the subclustering analysis I am still unclear why exactly C1 was chosen.

Referee #1:

The authors have addressed most of my concerns. A few points to consider:

1. As mentioned previously, speculating on the potential effects of multiple pregnancies on fibroblasts/CAF clusters would be a valuable addition to the discussion.

We have now added one sentence on pregnancy in the Discussion.

2. Page 7: Are the observed gene expression changes upon hormonal treatment due to ER and PR transcriptionally regulating Crabp1, Fabp4, Gdf10, and Bmp5?

This is an interesting point but remains to be proven in these cells.

Referee #2:

Ref.1:

1. Comment 1: The changes the authors have been made are well done. What would be even more precise is if the authors could define the actual timespan of the each period in days. What was the timespan in days the authors were using mice in puberty? What in Adulthood etc etc.. this would help authors researchers to potentially reproduce the results.

We have specified 4.5 weeks, 9-10 weeks, day 10 lactation, and so forth in the Figure legends for each experiment.

2. Comment 2: The additional performed analysis is good.

3. Comment 3: The full blots should be provided in the supplementary material! Full blots are included in Source Data.

"We now include clearer contrast images for β -galactosidase (Fig. 7c)."

This has to be marked in the actual figure because it is not well visible for the reader.

Arrows have now been added to the panels.

4. Comment 4: The arguments listed of the authors that it is beyond the purpose of this manuscript are ok. But further literature supporting "the one sentence in the discussion" should be implemented.

We have modified the sentence in the discussion and added a reference.

5. Comment 5: The additional analysis is good. Can the authors clarify in the investigated cohort what the actual pathology of the human tissues looks like? Is all breast cancers included in the cohort or do the clusters found vary in different subtypes?

The issue is that there is a huge literature existing for different markers and it needs to be clearly specified in a common nomen clature for further investigations.

These breast cancers correspond to the TCGA dataset, which is well-characterized as ER+ (luminal A and luminal B), HER2+, TNBC.

Ref 2:

6. Comment 1: Did the authors perform statistical analysis on the graphs shown?

We did not perform any statistical analysis on this graph because the graph showed the most obvious difference during pregnancy.

7. Comment 2: Is there more recent literature than 2002?

These appear to be the most cited papers that describe Gli1, Gli2 and Gli3 KO mice and their redundant functions *in vivo*.

8. Comment 5: The authors cannot see more proliferation of CAFs vs HAFs *in vitro*.

Did the authors perform stainings for the named proliferation of their mouse models?

We have not performed EdU incorporation experiments in the raGFP-Wnt1 model to assess proliferation of fibroblasts in hyperplasia versus tumors.

9. Comment 6: Also here the arguments of the authors towards a future knock out model are valid. But before that Wnt9 stainings on their mouse model in different age stages could be performed.

We agree but unfortunately there are no good Wnt9a antibodies available.

10. Comment 7: Again please define the human samples towards the cancer pathology more precise.

Please see above.

11. Comment 7: Here again a staining for Wnt might be additional helpful as *in vivo* confirmation is tissues of these mice were kept.

Additionally to these *in vitro* experiments.

We agree but unfortunately there are no good Wnt9a antibodies available.

12. Comment 8: Is there really no direct Wnt9 antibody available another close antibody of the similar family/ pathway could be used.

This is correct. The function of this Wnt ligand is largely unknown and hence the pathways utilised have not been defined as yet.

Recommend accept and I suggest an accompanying commentary to align this work with what is known the field

Referee #3:

While the authors have addressed some of my concerns, certain issues, in my opinion, still need to be resolved before publication:

1. The issue of clustering resolution (original point 1) has not been adequately addressed. The proposed iterative approach presents several issues. First, the resolution range between 0.2 and 0.3 is already quite narrow.

In our response, we mentioned that '*Initially, we chose clustering resolutions within the range of 0.2 to 0.3...*'. This indicates that our starting resolution was within this range, but it does not imply that we limited our analysis to these values. For each clustering analysis, we also tested resolutions outside this range to explore the data more comprehensively.

Additionally, interpreting clustering results using UMAP visualizations requires caution, as UMAP is highly sensitive to hyperparameter choices, and the distances between clusters may not be meaningful.

We thank the reviewer for the comment. We fully acknowledge that UMAP visualizations can vary depending on parameter settings. However, our interpretation of the clustering results did not rely solely on UMAP. Instead, we primarily examined and interpreted the clusters based on their marker genes and signature scores—an approach entirely independent of UMAP visualization—alongside prior knowledge from the literature and our understanding of the underlying biology.

As the authors themselves demonstrate with their subclustering approach, determining the final number of clusters is inherently difficult, as clusters can be further subdivided. Relying on the technically most optimal solution helps maintain an unbiased approach, whereas biological interpretation remains largely subjective. In my opinion, the authors have not convincingly shown that their chosen resolution is superior or more justified than the technically optimal alternative.

There is no universally optimal clustering solution that works for all single-cell datasets, across all purposes, and in every biological context. The reviewer suggested using the silhouette score — which maximizes inter-cluster variability and minimizes intra-cluster variability — to select the optimal clustering resolution. However, this metric is not the most suitable for our specific goal: identifying heterogeneity, including subtypes and distinct states, within fibroblast subsets.

To illustrate why the silhouette score may not always align with particular research objectives, we present a simple example using a well-characterized public dataset (code used for generating the plots can be provided upon request). This dataset, featured in Seurat's vignette (https://satijalab.org/seurat/articles/pbmc3k_tutorial), contains known ground truth for cell identities. Following the exact analysis described in the vignette, the silhouette score would suggest an optimal clustering resolution of 0.15. However, at this resolution, critical distinctions between biologically distinct populations are lost — such as the separation of CD8 T cells from NK cells, naïve CD4 T cells from memory CD4 T cells, and CD14+ monocytes from FCGR3A+ monocytes. These distinctions only become apparent at a higher resolution of 0.5 – a resolution that was manually chosen in the vignette.

Similarly, in our study, where we aim to explore subclusters within a specific cell population, relying on a technically optimal metric like the silhouette score would not adequately capture the biological heterogeneity that we are investigating.

2. From figures 1b and c I can't assess that cluster C1 has the most notable changes. Majority of the clusters show changes as these are relative proportions. In addition, changes in relative proportions do not necessarily have to lead to changes in population heterogeneity. While I don't have an issue with the subclustering analysis I am still unclear why exactly C1 was chosen.

Among all the clusters, C1 is the only one in which the cell proportion increases from approximately 15% during puberty to nearly 50% during involution (Fig. 1c). It is also interesting that about half of the C1 cluster is absent in the puberty sample, as shown in Fig.1b.

Dear Dr Visvader,

Thank you for submitting the revised version of your manuscript. I have now evaluated your amended manuscript and concluded that the remaining minor concerns have been sufficiently addressed.

I am thus pleased to inform you that your manuscript has been accepted for publication in the EMBO Journal.

Related, I would like to hereby ask your consent on keeping the rebuttal figures included in this file.

On a different note, I would like to alert you that EMBO Press offers a format for a video-synopsis of work published with us, which essentially is a short, author-generated film explaining the core findings in hand drawings, and, as we believe, can be very useful to increase visibility of the work. Please see the following link for representative examples and their integration into the article web page:

<https://www.embopress.org/doi/full/10.15252/emj.2019103932>

Best regards,

Daniel Klimmeck

Daniel Klimmeck, PhD
Senior Editor
The EMBO Journal
EMBO
Postfach 1022-40
Meyerhofstrasse 1

D-69117 Heidelberg
contact@embojournal.org
